# Continuous strain tuning of oxygen evolution catalysts with anisotropic thermal expansion

Yu Du[1], Fakang Xie[1], Mengfei Lu[1,2], Rongxian Lv[3], Wangxi Liu[1,2], Yuandong Yan[1], Shicheng Yan [1] ✉ & Zhigang Zou [1,2]

Compressive strain, downshifting the $d$-band center of transition metal oxides, is an effective way to accelerate the sluggish kinetics of oxygen evolution reaction (OER) for water electrolysis. Here, we find that anisotropic thermal expansion can produce compressive strains of the $IrO_6$ octahedron in $Sr_2IrO_4$ catalyst, thus downshifting its $d$-band center. Different from the previous strategies to create constant strains in the crystals, the thermal-triggered compressive strains can be real-timely tuned by varying temperature. As a result of the thermal strain accelerating OER kinetics, the $Sr_2IrO_4$ exhibits the nonlinear $\ln j_o$ - $T^{-1}$ ($j_o$, exchange current density; $T$, absolute temperature) Arrhenius relationship, resulting from the thermally induced low-barrier electron transfer in the presence of thermal compressive strains. Our results verify that the thermal field can be utilized to manipulate the electronic states of $Sr_2IrO_4$ via thermal compressive strains downshifting the $d$-band center, significantly accelerating the OER kinetics, beyond the traditional thermal diffusion effects.

Electrochemical water splitting driven by renewable electricity provides an ideal strategy for sustainable production of clean hydrogen fuel from water. However, the high overpotentials for oxygen evolution reaction (OER) in the water electrolysis pose a bottleneck for large-scale hydrogen production. Therefore, it is urgent to develop efficient and stable OER catalysts to overcome this challenge. Electronic states of materials, determining the energetics during the reagent adsorption, the evolution of intermediates, and the product desorption, are directly related to the OER kinetics. Strain engineering is especially effective in optimizing the $d$-band center of transition metal-based catalysts via compressive or tensile strain in crystals[1–5]. Generally, tensile strain reduces the overlap of the wavefunctions and thus gives rise to the narrowing of the metal $d$ band and an upshift of $d$-band center[6]. In contrast, compressive strain has the opposite effect, causing a broadening of the $d$ band and a downshift of the $d$-band center. For OER, the $d$-band center is closely related to the interaction energy between adsorbate states and the metal $d$ states,

thus determining the OER kinetics. Conventionally, various straining methods, such as lattice mismatch[7–10], doping heteroatoms[11,12], morphology controlling[13,14], and introducing defects[15–17], were proposed to produce the constant strains in the crystals. In these methods, the constant strain is mainly determined by the compositions, microstructures and synthesis conditions of materials. Additionally, the variable strains can be in situ generated during the service of materials, in which the materials directly grow or are coated onto the flexible substrates to form a flexible film which can be subjected to tensile or compressive loading from the external forces[18–21]. Obviously, although the variable strain is beneficial to optimize the electronic states, the dependence of complex preparation processes and state-of-the-art equipment limits its practical applications.

In order to promote the applications of strain engineering in OER, there is an urgent need to find a simple method to generate variable strains in materials. Heat is a common source of energy

[1]Collaborative Innovation Center of Advanced Microstructures, National Laboratory of Solid State Microstructures, Eco-materials and Renewable Energy Research Center (ERERC), College of Engineering and Applied Sciences, Nanjing University, No. 22 Hankou Road, Nanjing 210093 Jiangsu, PR China. [2]Jiangsu Key Laboratory for Nano Technology, Nanjing University, No. 22 Hankou Road, Nanjing 210093 Jiangsu, PR China. [3]Industrial Center, Nanjing Institute of Technology, No. 1 Hongjing Avenue, Nanjing 211167 Jiangsu, PR China. ✉e-mail: yscfei@nju.edu.cn

that can be easily utilized and modulated to produce various thermophysical effects, such as thermal expansion[22], thermal diffusion[23], and thermal phase transition[24,25], which are useful in energy conversion fields. In particular, it was demonstrated that the anisotropic thermal expansion typically originates from the thermal-induced elementary microstructural deformation caused by transverse vibrations of the bridging atoms, probably producing strains[26]. Accordingly, it is possible to utilize the thermal expansion effects as a convenient route to generate variable strains in OER catalysts. Here, we heat the $Sr_2IrO_4$, an anisotropic thermal expansion material, to generate thermal strains for verifying the effectiveness of this idea in regulating electronic states of materials. Under thermal stimulation, the compressive strains of the $IrO_6$ octahedra in $Sr_2IrO_4$ catalyst can be created, consequently downshifting the $d$-band center. As a result, the binding strength between OER intermediates and the Ir active species was optimized to meet the low-barrier catalytic reaction described by Sabatier principle[27], thus accelerating OER kinetics following the nonlinear Arrhenius relationship. Our findings prove that thermal stimulation producing compressive strains is an effective route to tune the electronic states of $Sr_2IrO_4$ with anisotropic thermal expansion, significantly increasing OER performances, beyond the traditional thermal diffusion effects.

# Results

## Thermal strains in $Sr_2IrO_4$

In order to confirm the structures and compositions of $Sr_2IrO_4$ during temperature-dependent OER, we carried out an OER $i$-$t$ test at 1.45 V at 90 °C for 1 h. Afterwards, the X-ray diffraction (XRD) and scanning electron microscope (SEM) analyses indicated that the electrode is electrochemically stable $Sr_2IrO_4$ particles with tetragonal structure (JCPDS Card No. 09-0099) and particle size of 200–300 nm (Supplementary Fig. 1a, b). The high-resolution transmission electron microscope (HRTEM) lattice image confirmed that the $Sr_2IrO_4$ particle is highly crystalline and exhibits the continuous lattice fringes of 0.32 nm for (004) facet extending through the whole particle, evidencing that the particle with irregular shape is an undeveloped single crystal, as confirmed by the fast Fourier transform image (Fig. 1a). To disclose the growth mechanism of the $Sr_2IrO_4$ single crystals, the HRTEM lattice image and the selected area electron diffraction (SAED) were observed on a relatively perfect $Sr_2IrO_4$ crystal (Fig. 1b). The results suggested that the $Sr_2IrO_4$ particle tends to stack the facet along [001] axis, thus exhibiting a possibility to expose (001) facet (Supplementary Fig. 1). The compositions of the particle were determined by the high-angle annular dark field scanning transmission electron microscopy (HAADF-STEM) image and elemental mapping. The particle comprises of uniformly dispersed Sr, Ir, and O elements (Supplementary Fig. 2).

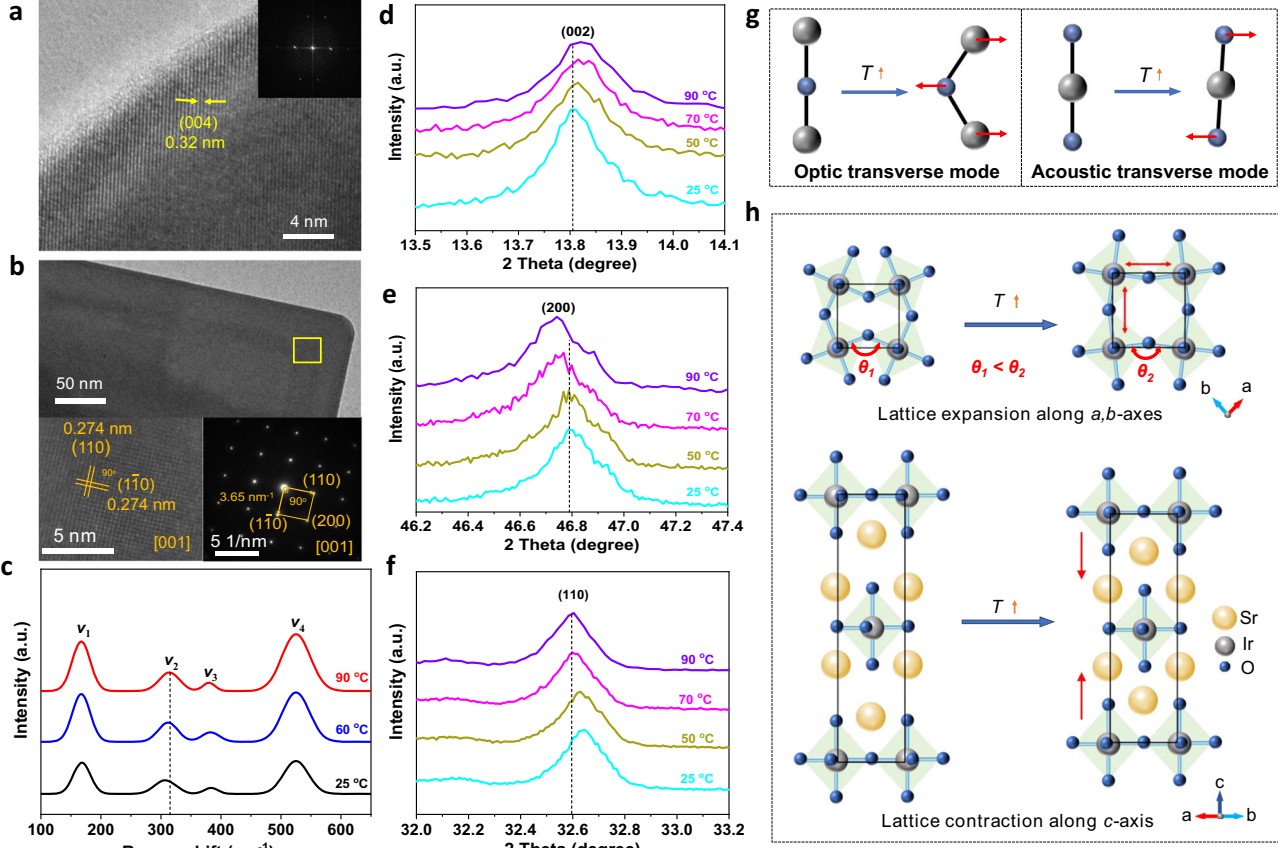

**Fig. 1 | The thermal properties of $Sr_2IrO_4$. a** HRTEM lattice image for an undeveloped $Sr_2IrO_4$ single crystal with irregular shape. Inset shows the fast Fourier transform image. **b** TEM image, HRTEM lattice image, and SAED pattern for a relatively perfect $Sr_2IrO_4$ single crystal to show the possible growth mechanism and exposed facet. The HRTEM lattice image and SAED pattern were collected on the yellow box area. **c** Raman spectra of $Sr_2IrO_4$ under different temperatures. The mode $v_1$ (a superposition of an $A_{1g}$ and a $B_{2g}$) is a rotation of the $IrO_6$ octahedra along the c-axis combined with an in-phase Sr displacement along c, $v_2$ ($A_{1g}$ mode) is a rotation of the $IrO_6$ octahedra along c, $v_3$ ($B_{2g}$ mode) is an in-plane bending of the $IrO_6$ octahedra, and $v_4$ ($A_{1g}$ mode) is a stretching mode involving a modulation of the Ir-O (apical) distance. **d**–**f** XRD peak shifts for (002), (200), and (110) facets of the $Sr_2IrO_4$ when varied temperatures from 25 to 90 °C. **g** A scheme to show the transverse vibrations of bridging atoms under thermal stimulation, optic transverse mode (left) and acoustic transverse mode (right). **h** Cooperative rocking or tilting motions of $IrO_6$ octahedra in $Sr_2IrO_4$ resulting from the optic and acoustic transverse modes, leading to the expansion along $a,b$-axes and the contraction along $c$-axis.

The thermal response of $Sr_2IrO_4$ particles is first checked by temperature-dependent Raman spectra. As shown in Fig. 1c, there are four Raman peaks at 168.20, 307.31, 383.59, and 525.05 $cm^{-1}$, labeled as $v_1$ - $v_4$. The Lorentzian function was used to fit the data and get the peak parameters (Supplementary Table 1). The mode $v_1$ (a superposition of an $A_{1g}$ and a $B_{2g}$) is a rotation of the $IrO_6$ octahedra along the $c$-axis combined with an in-phase Sr displacement along $c$, $v_2$ ($A_{1g}$ mode) is a pure rotation of the $IrO_6$ octahedra along $c$, $v_3$ ($B_{2g}$ mode) is an in-plane bending of the $IrO_6$ octahedra, and $v_4$ ($A_{1g}$ mode) is a stretching mode involving a modulation of the Ir-O (apical) distance. Obviously, the $v_1$, $v_2$, and $v_4$, vibration modes result from the transverse vibrations of the bridging oxygens among $IrO_6$ units[28,29]. With increasing temperatures, $v_2$ for rotation of $IrO_6$ units shifts from 307.31 $cm^{-1}$ under 25 °C to 314.12 $cm^{-1}$ under 90 °C, indicating the reduced tilting of $IrO_6$ octahedra. The thermally increased intensity of peaks $v_1$ and $v_4$ for the mixed motion is indicative of lattice distortions. Temperature-dependent XRD patterns indicated that, while heating, the (002) diffraction peak shifts towards higher angles, whereas the (200) and (110) diffraction peaks move towards lower angles (Fig. 1d–f), suggesting that the lattice contracts along $c$-axis and the lattice expands along $a,b$-axes. Indeed, Rietveld refinements of XRD patterns further revealed that the $Sr_2IrO_4$ with $I4_1/acd$ symmetry displays the contraction of the unit cell parameter $c$, from 25.74 Å under 25 °C to 25.72 Å under 90 °C, and the expansion of the unit cell parameter $a$, from 5.50 Å under 25 °C to 5.51 Å under 90 °C (Supplementary Fig. 3). The $Sr_2IrO_4$ is an anisotropic thermal expansion material, following the lattice anharmonic vibrations dominated low-frequency phonon mechanism[30]. Under thermal stimulation, the $IrO_6$ octahedra will undergo cooperative rocking motions resulting from the optic and acoustic transverse modes of bridging atoms (Fig. 1g). As a result, with elevating temperatures, the reduced tilting of $IrO_6$ octahedra of $Sr_2IrO_4$ will trigger obvious lattice distortion, the expansion along $a,b$-axes and the contraction along $c$-axis (Fig. 1h)[22,26,31,32].

## The Ir-O bond and $d$ band in $Sr_2IrO_4$ under heating

To gain insight into thermal-induced variations in the local electronic and geometric structure in $Sr_2IrO_4$, we collected X-ray absorption near-edge structure (XANES), extended X-ray absorption fine structure (EXAFS), and X-ray photoelectron spectroscopy (XPS) valence band spectra (Fig. 2). In XANES spectra (Fig. 2a), the white line of Ir $L_3$-edge originates from the electron transition from the occupied Ir $2p_{3/2}$ orbital to the partially occupied Ir $5d$ orbital[11]. The peak intensity of white line is related to the number of unoccupied $5d$ states. The white-line intensity increases when temperatures rise from 25 to 90 °C, which reveals the higher valence state of Ir when thermally increased the number of unoccupied $5d$ states. The EXAFS provides details about the distance of the back-scattering atom surrounding the central absorbing atom. The peak at 1.0 − 2.0 Å is assigned to scattering of the nearest 6-coordinated Ir-O atoms for the $IrO_6$ units in $Sr_2IrO_4$ (Fig. 2b and Supplementary Fig. 4). Considering the thermal distortions, the $IrO_6$ unit can be split into 4-coordinated Ir-O bond with bond length of 1.989 Å in the $ab$-plane (Ir-O$_{ab\text{-plane}}$) and 2-coordinated Ir-O bond with bond length of 1.993 Å in the $c$-axis (Ir-O$_{c\text{-axis}}$) under 25 °C (Supplementary Table 2). Heating up to 90 °C, the Ir-O$_{ab\text{-plane}}$ and Ir-O$_{c\text{-axis}}$ bond lengths are shortened to 1.938 Å and 1.937 Å, respectively. This is a solid evidence of compressive strain generation in the $IrO_6$ octahedron as heating $Sr_2IrO_4$. The peak at 2.0–3.0 Å is attributed to the scattering of neighboring Ir-Ir atoms. The Ir-Ir bond is elongated from 2.730 Å under 25 °C to 2.744 Å under 90 °C. The elongated Ir-Ir bond is indicative of the reduced titling among $IrO_6$ octahedra. This means that the compressive strain generation in the $IrO_6$ octahedron is a result of the transverse vibrations of bridging oxygens among $IrO_6$ octahedra. Wavelet transform (WT) analysis of EXAFS spectra, which provides both **R**- and **k**-space information and discriminates the back-scattering atoms, was undertaken to visualize and confirm the Ir-O bond in $IrO_6$ units and Ir-Ir bond among $IrO_6$ units (Fig. 2c, d).

The Ir $4f_{5/2}$ core-level XPS analysis indicated that the binding energy at 61.81 eV under 25 °C is assigned to the $Ir^{4+}$ (Fig. 2e and Supplementary Table 3)[33,34]. Heating $Sr_2IrO_4$ up to 90 °C, the binding energy of $Ir^{4+}$ shifted positively about 0.17 eV, indicating the increased valence state of Ir, in line with the EXAFS result[9]. This result suggests that the compressive strain in $IrO_6$ units is beneficial to stabilize the high valence of Ir species. Due to strong spin-orbital coupling in $5d$ transition-metal oxides[35], the low-spin $Ir^{4+}$ ($5d^5$, $t_{2g}^5 e_g^0$) 5d orbital in $Sr_2IrO_4$ splits the five electrons in the $t_{2g}$ band into four electrons occupied as the $J_{eff}=3/2$ sub-band and to one electron occupied as the $J_{eff}=1/2$ sub-band. The XPS valence band analysis was carried out to visualize the $d$-band electronic structures near the Fermi level ($E_f$). The valence-band XPS spectra under 25 °C show three peaks (Fig. 2f and Supplementary Table 4): the $J_{eff}=1/2$ sub-band at 0.99 eV, the $J_{eff}=3/2$ sub-band at 3.81 eV, and the metal-oxygen π bands at 8.17 eV[36,37]. Under 90 °C, they move to 1.03, 3.92, and 8.17 eV, respectively. The increase in binding energies with increasing temperatures implies that the thermal strains in $IrO_6$ units make the downshift of $d$-band away from the $E_f$. Indeed, the thermal-induced compressive strains shorten the Ir-O bond, thus contributing to the downshift of the $d$ band due to the enlarged Ir-O covalency[38].

## OER on $Sr_2IrO_4$ under heating

Next, we investigate the effects of thermal strains on OER performances. The linear sweep voltammetry (LSV) polarization curves of the $Sr_2IrO_4$ electrode were recorded under heating 1.0 M KOH electrolyte from 20 to 90 °C with inner resistance ($iR$) compensation of 90%, and the OER currents were normalized by the electrochemical active surface areas (ECSA) (Supplementary Fig. 5). As shown in Fig. 3a, heating from 20 to 90 °C, the potential requirement to reach 10 mA $cm^{-2}$ decreased from 1.55 to 1.40 V. Considering the temperature dependence of OER thermodynamic potentials (Supplementary Table 5)[39], the OER overpotentials reduced from 312 mV under 20 °C to 235 mV under 90 °C. The OER performances of the $Sr_2IrO_4$ under room temperature is comparable with the known reported Ir-based catalysts (Supplementary Table 6). Notably, under heating, the OER performances of the $Sr_2IrO_4$ increase significantly. Correspondingly, the Tafel slope decreased from 90.32 mV $dec^{-1}$ under 20 °C to 45.14 mV $dec^{-1}$ under 90 °C (Fig. 3b). The significantly reduced Tafel slopes under higher temperatures indicate that the thermal strains, beyond the thermal-induced mass transfer, greatly accelerated OER kinetics. The positive correlation between thermal strain and OER activity was further verified by monitoring the thermal response of LSV curves (Supplementary Fig. 6). During heating and cooling electrolyte, the completely reversible LSV polarization curves witnessed that the OER performances of $Sr_2IrO_4$ are highly sensitive to the temperature.

As is well-known, the thermal diffusion can promote the electrochemical OER rate through accelerating mass transfer, obeying a linear Arrhenius relationship of $\ln j_o = -Q/(RT) + \ln A$ ($j_o$, exchange current density; $Q$, activation energy; $A$, pre-exponential factor; $T$, absolute temperature; $R$, universal gas constant)[23]. This means that the increase in OER reaction rate by thermal diffusion is strictly following a linear $\ln j_o \cdot T^{-1}$ relationship if heating does not change the electronic states of materials. Usually, according to this linear relationship, the $Q$ of a catalytic reaction can be calculated with a basic assumption of electronic structures of materials without dependency of temperatures. We accordingly calculated the $\ln j_o \cdot T^{-1}$ relationship for OER of $Sr_2IrO_4$. As shown in Fig. 3c, a nonlinear $\ln j_o \cdot T^{-1}$ relationship revealed that the $Q$ for OER of $Sr_2IrO_4$ is a function of temperatures. This nonlinear relationship would originate from that the real-time generation of thermal strains changes the electronic states of the $Sr_2IrO_4$, thus producing a variable OER rate with the dependence of temperatures. To clearly distinguish the contributions of thermal diffusion and thermal strain

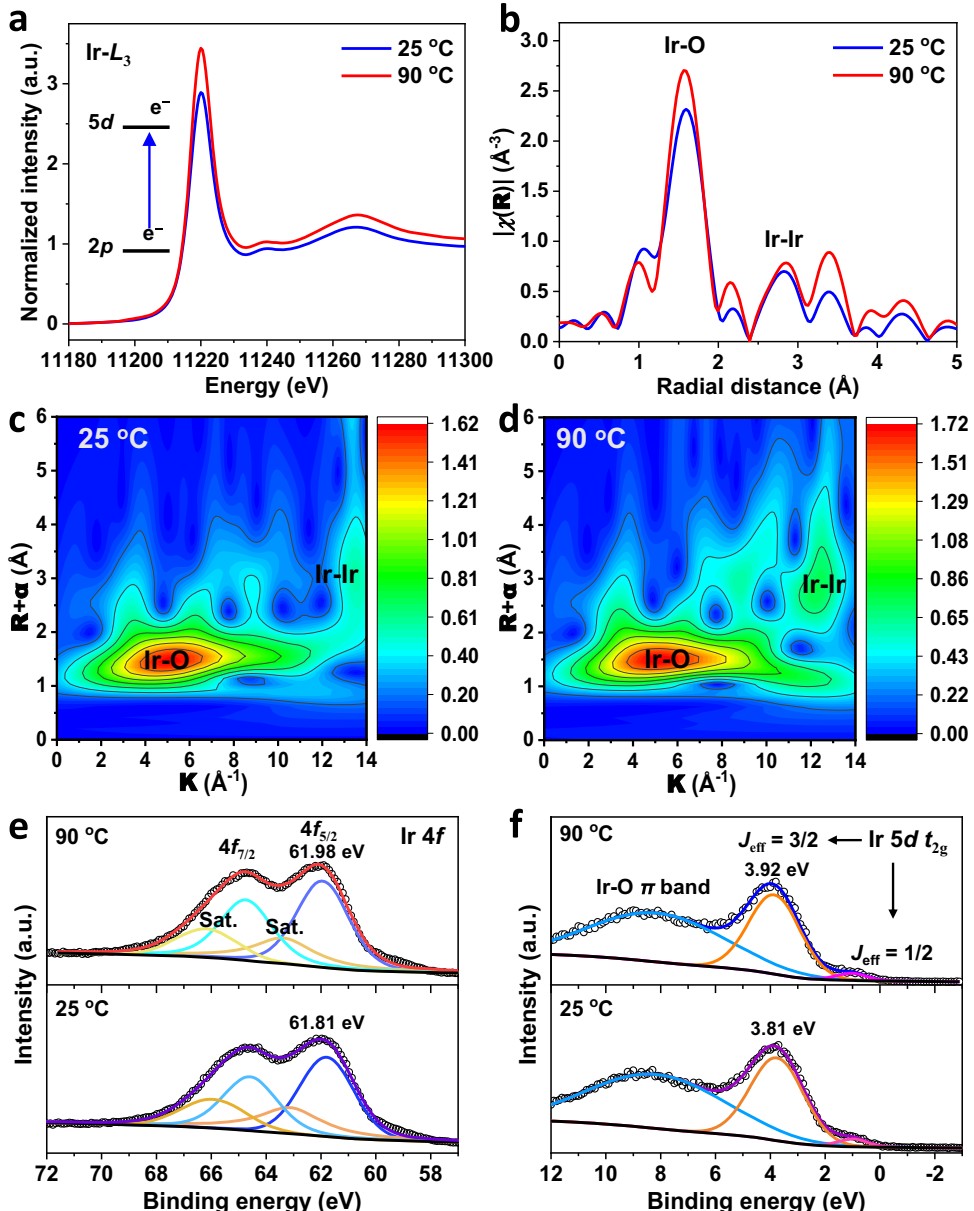

**Fig. 2 | Electronic structure changes in Sr$_2$IrO$_4$ under thermal stimulation. a** Ir $L_3$-edge XANES region of Sr$_2$IrO$_4$ measured under different temperatures. **b** k$^3$-weighted FT curves of the EXAFS data at the Ir $L_3$-edge. Wavelet transform plots of the EXAFS data of Sr$_2$IrO$_4$ under 20 °C (**c**) and 90 °C (**d**). Ir $4f$ XPS spectra (**e**) and valence band spectra (**f**) of Sr$_2$IrO$_4$ under different temperatures.

during OER process, we studied heating water oxidation on SrIrO$_3$ (6H phase) with negligible thermal strains under 20–90 °C, with just thermal diffusion to accelerate chemical reaction kinetics. Totally different from the Sr$_2$IrO$_4$, the SrIrO$_3$ exhibits no obvious lattice distortions in XRD patterns when varying temperatures from 25 to 90 °C (Supplementary Fig. 7). This result suggests that the crystal structures of the SrIrO$_3$ are not sensitive to the temperature. Evidently, as heating, the OER performances on SrIrO$_3$ are thermally accelerated, following a linear ln$j_o$ - $T^{-1}$ Arrhenius relationship (Supplementary Figs. 8–10). The heat-induced negligible changes in binding energies for Ir, Sr, and O in the SrIrO$_3$ confirm that heating does not affect the electronic states of the SrIrO$_3$ (Supplementary Fig. 11). As a result, the thermally enhanced OER kinetics on the SrIrO$_3$ electrode arise from the thermal diffusion.

The in situ electrochemical impedance spectroscopy (EIS) was performed to verify the thermally triggered strain effects on OER electron transfer. A single phase peak was detected in Bode plots and can be assigned to the electron transfer at the interface between active center and OER intermediates (˙OH, ˙O, ˙OOH) (Fig. 3d, e). Indeed, the phase angle starts at the potentials above 1.5 V under 20 °C and above 1.4 V under 90 °C, in good agreement with the OER onset potentials of LSV curves (Fig. 3a). Increasing temperatures of the electrolyte, the phase peaks moved to the higher frequencies and the phase angles sharply decreased, confirming that the thermal strains accelerate the OER kinetics. The open circuit potential ($V_{ocp}$) describes the quasi-Fermi level of the electrode against reference electrode when the electrode is at open-circuit state. In our case, the quasi-Fermi level of Sr$_2$IrO$_4$ electrode is mainly determined by the electron filling of the $d$ band, thus providing a route to monitor the electron transfer kinetics. After polarizing the electrode at 1.55 V under different temperatures, the $V_{ocp}$ decay monitors the real-time evolution of quasi-Fermi level of the electrode, which is mainly dominated by the speed for electron transfer from electric double layer to $d$ band (Fig. 3f). The V$_{OCP}$ decay originates from the reaction between high-valence Ir species and the reducing species from electrolyte to upshift the Fermi level of the

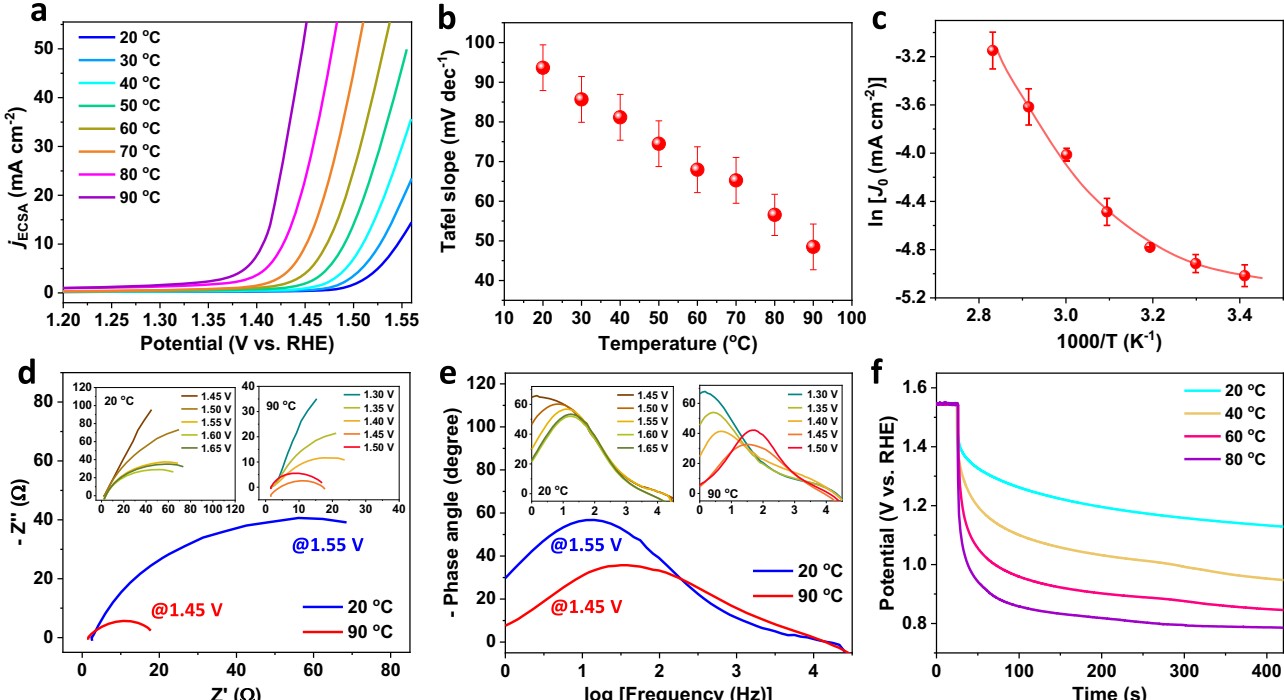

**Fig. 3 | OER performances of Sr$_2$IrO$_4$ with thermal strains. a** LSV curves normalized by ECSA for Sr$_2$IrO$_4$ in 1.0 M KOH when varied temperatures from 20 °C to 90 °C (scan rate, 5 mV/s; mass loading, 0.4 mg cm$^{-2}$; 90% *iR*-drop compensation is utilized; the pH of the electrolyte under different temperatures are shown in Supplementary Table 7.) **b** Temperature-dependent Tafel slopes of Sr$_2$IrO$_4$. **c** Calculated ln$j_0 \cdot T^{-1}$ plot with nonlinear Arrhenius relationship for OER on Sr$_2$IrO$_4$. **d** Potential-dependent Nyquist plots at 1.55 V under 20 °C and at 1.45 V under 90 °C.

Insets show the Nyquist plots when varied potentials from 1.45 to 1.65 V under 20 °C and the Nyquist plots when varied potentials from 1.3 to 1.5 V under 90 °C. **e** Potential-dependent Bode plots at 1.55 V under 20 °C and at 1.45 V under 90 °C. Insets show the Bode plots when varied potentials from 1.45 to 1.65 V under 20 °C and the Bode plots when varied potentials from 1.3 to 1.5 V under 90 °C. **f** OCP decay after polarization of Sr$_2$IrO$_4$ electrode at 1.55 V under different temperatures.

electrode and is obviously temperature-dependent. The $V_{ocp}$ decay stopped at a quasi-steady potential, 1.1 V for Ir$^{4+}$ species under 20 °C and 0.77 V for Ir$^{\delta+}$ ($\delta < 4$) under 90 °C, as demonstrated by the XPS analysis to show low-valence Ir for electrode after OER under 90 °C (Supplementary Fig. 12)[40]. This fact suggests that the electron transfer and reactivity of the surface active Ir species are temperature-dependent. Higher temperatures will benefit the valence-state variation of active Ir species, indicative of the higher OER activity for Sr$_2$IrO$_4$ with thermal strains. Enlarging the thermal strains under higher temperatures, the rapid $V_{ocp}$ decay and the lower quasi-Fermi level are a result of low-barrier electron transfer.

Next, we established the plausibility of the mechanism of thermal strains promoted OER on the likely exposed facets by density functional theory (DFT) calculations. In modeling thermal-induced strains, we should ensure that the compressive strains was produced in the IrO$_6$ units of Sr$_2$IrO$_4$. We firstly created the structural model of Sr$_2$IrO$_4$ with the thermal strains at 20 °C and 90 °C by setting the unit cell parameters with a thermal expansion coefficient of $2.8 \times 10^{-5}$ K$^{-1}$ for *a*- and *b*-axes and $-1.2 \times 10^{-5}$ K$^{-1}$ for *c*-axis, which is obtained by fitting XRD data. However, the structural model of Sr$_2$IrO$_4$ created by setting the thermal expansion coefficient is underperformance in describing the compressive strains in IrO$_6$ units (Supplementary Fig. 13). Therefore, as a compromise, the structural model of Sr$_2$IrO$_4$ with $I4_1/acd$ symmetry[41] was built by setting the unit cell parameters from the temperature-dependent XRD refinement and the atomic fractional coordinates from Ir-O bonding length in IrO$_6$ units from EXAFS analysis at 20 and 90 °C. The lattice parameters of Sr$_2$IrO$_4$ are $a = b = 5.50$ Å, $c = 25.74$ Å, for Sr$_2$IrO$_4$ under 25 °C, and $a = b = 5.51$ Å, $c = 25.72$ Å for Sr$_2$IrO$_4$ under 90 °C. Under 25 °C, the Ir-O$_{ab\text{-plane}}$ and Ir-O$_{c\text{-axis}}$ bond lengths are 1.989 Å and 1.993 Å, respectively. Under 90 °C, the Ir-O$_{ab\text{-plane}}$ and Ir-O$_{c\text{-axis}}$ bond lengths are 1.938 Å and 1.937 Å, respectively. Considering

that the Sr$_2$IrO$_4$ particle is not a perfect crystal with nearly irregular profile, we give comprehensive consideration on selecting the likely exposed (001) and (110) facets to carry out the DFT calculations. Firstly, the (001) facet is a thermodynamically stable plane in the Sr$_2$IrO$_4$[42], which is able to reflect at least partially the bulk properties of Sr$_2$IrO$_4$. In addition, the layered Sr$_2$IrO$_4$ is easy to cleave along the (001) facet due to the weak interlayer interactions[42], increasing the probability to expose it as terminal facet. Indeed, as shown in Supplementary Fig. 1b, although the single-crystal Sr$_2$IrO$_4$ particles are nearly irregular, the stacking growth of (001) facets is visible as traced by dotted lines. Therefore, the DFT calculations were conducted to investigate the adsorption properties of oxygen-containing intermediates on Ir sites of Sr$_2$IrO$_4$ (001) and (110) facets, two likely exposed facets according to the possible crystal growth mechanism and the crystallographic symmetry of Sr$_2$IrO$_4$.

According to the four-step OER mechanism, the Sr$_2$IrO$_4$ underwent the four consecutive proton-coupled electron transfer steps with *OH, *O, and *OOH intermediates[9,43]. Considering the satisfied stability of Sr$_2$IrO$_4$ during OER (No surface reconstruction after OER *i-t* test at 1.45 V at 90 °C for 1 h is observed by the HRTEM in Fig. 1a) and the single Ir coordination environment on both (001) and (110) facets (Supplementary Fig. 14), we adopted the single-site adsorbate evolution mechanism (AEM) model with 100% coverage of adsorbates (100% coverage is one absorbate per one active site) on Ir active site to calculate the Gibbs free energy changes of the elementary reaction steps[41]. In this model, the energy requirement for adsorbing the OER adsorbates onto the single active Ir site can reflect their evolution kinetics. The surface (001) and (110) slab models of Sr$_2$IrO$_4$ were created and stabilized by hydroxyl group passivating the unsaturated Ir sites except the Ir active site because the hydroxylation of oxides is a thermodynamically favorite process in the 1.0 M

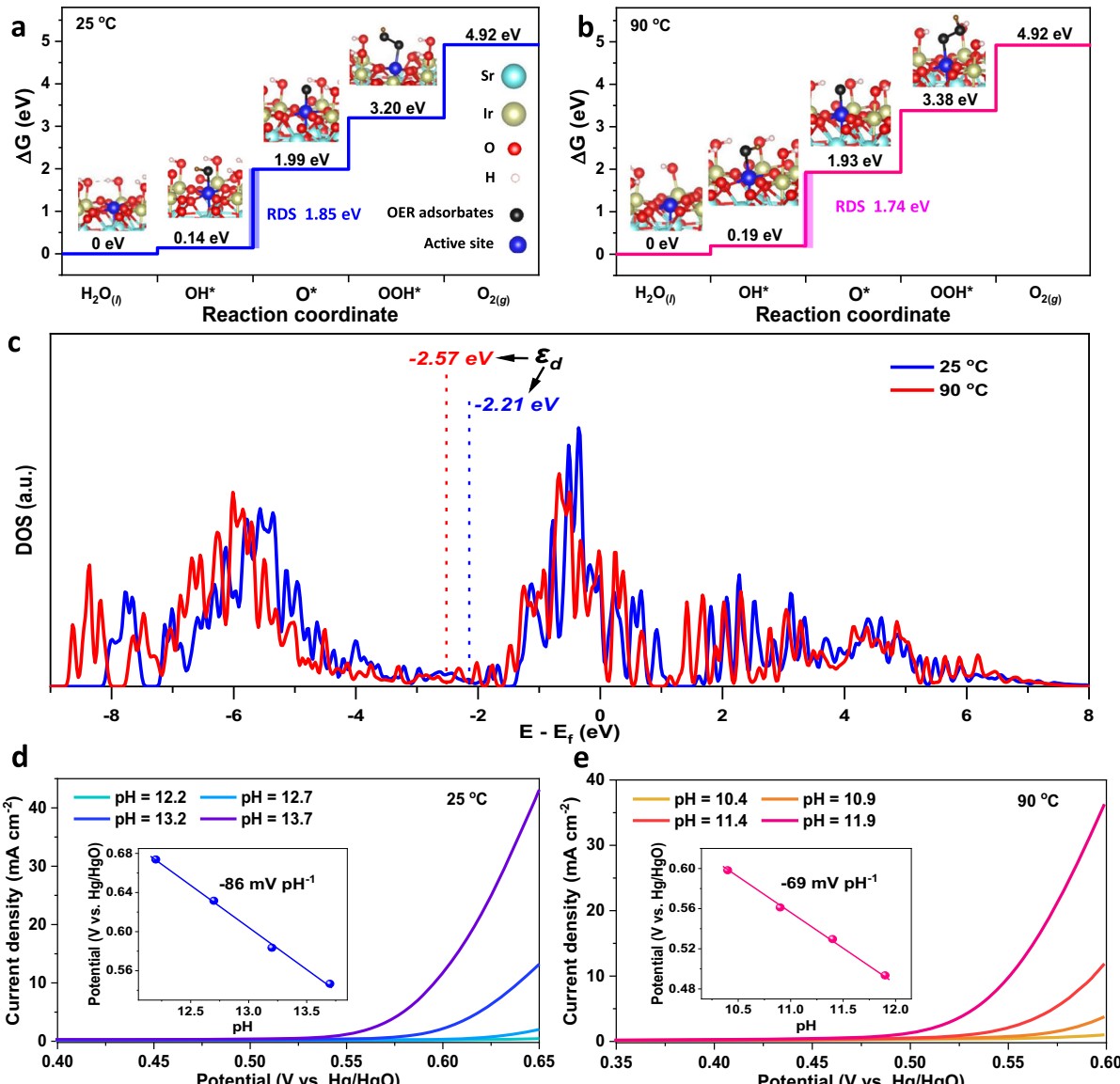

**Fig. 4 | DFT calculations to show the plausibility of the thermal strains-accelerated OER kinetics. a**, **b** Gibbs free energy diagrams for OER intermediates (˙OH, ˙O, ˙OOH) adsorbing onto the single Ir active site (the 100% coverage of OER intermediates was achieved by hydroxyl group passivating the coordinately unsaturated Ir sites except the single Ir active site) of the likely exposed Sr$_2$IrO$_4$ (001) surface with thermal strains under 25 °C (**a**) and 90 °C (**b**), with calculated structures and rate-determining steps. Insets are the corresponding structures of the ˙OH, ˙O, and ˙OOH adsorptions on Ir sites. The Gibbs free energy changes were calculated without considering the effects of temperature, pH, and solvation on OER. **c** The projected DOS of Ir 5d of Sr$_2$IrO$_4$ under 25 °C and 90 °C. The ε$_d$ indicates the d-band center calculated as the first statistical moment of the d-projected DOS. **d**, **e** The pH dependence of OER onset potentials to confirm the rate-determining step, the deprotonation of ˙OH on Sr$_2$IrO$_4$ with thermal strains under 25 °C (**d**) and 90 °C (**e**). Onset potentials were determined by the applied potentials at a constant current of 1 mA cm$^{-2}$. Insets show the linear relationship of onset potential vs. pH.

KOH electrolyte with abundant hydroxyls[44,45], thus avoiding the undesired surface reconstruction during structure optimization. In the single-site AEM model, there is no evolution of adsorbates between the adjacent sites or the simultaneous evolution of adsorbates on all Ir sites of the surface at the same time. Obviously, the single-site AEM model avoids the possible interactions between adjacent sites, thus benefiting to reflect the intrinsic catalytic activity (Supplementary Fig. 15).

We employed the PBE exchange-correlation functional with corrections of on-site Coulomb U and spin-orbit coupling (SOC) effect to describe the electronic states of Sr$_2$IrO$_4$ (details see the Methods). The corrections of U and SOC effect aim at compensating the underperformances in PBE describing localized 5d states. For describing the interactions between catalyst and adsorbates, Nørskov has proposed

that the RPBE (a revised PBE functional for accurately describing the chemisorption energies, generating by improving the mathematical form for the exchange energy enhancement factor[46]) is more accurate than PBE. However, the two functionals, PBE and RPBE, share the same construction logic and therefore contain the same physics and fulfill the same physical criteria, and therefore the most limitations of PBE still remain in RPBE and the RPBE exhibits obvious improvement in describing interactions between catalyst and adsorbates[46]. It is worth noting that the efficacy of RPBE in describing catalyst/adsorbates interactions depends on transition metals and RPBE usually reduces an overestimation of PBE describing catalyst/adsorbates interactions[47]. To validate the plausibility of chemisorption energies calculated by PBE functional, we compared the energy profiles obtained with PBE + U + SOC (Fig. 4a, b) and RPBE + U + SOC (Supplementary Fig. 16) and

found that they exhibit the similar efforts in describing the interactions between $Sr_2IrO_4$ and OER adsorbates, in good agreement with the previous OER calculations on $IrO_2$[48,49], suggesting that the PBE + U + SOC approach is acceptable for describing the electronic states of $Sr_2IrO_4$ and its interactions with the OER adsorbates. The small difference in chemisorption energies calculated by PBE and RPBE probably stems from that the RPBE is mainly an advantage for the early and mid transition metals and not very much for the late transition metals[47]. Here, it is worth noting that, our DFT calculations were based on the assumptions including the direct use of experimental crystal structure, the likely exposed (001) and (110) facets, and the single-site AEM model. In this situation, the DFT calculations just provide the information of electronic states of the $Sr_2IrO_4$ structure with strains, which do not reflect the ground-state properties of $Sr_2IrO_4$. And also the Gibbs free energy is a result of the interactions between the valence states of OER intermediates and the electronic states of the $Sr_2IrO_4$ structure with strains.

The free energy calculations on (001) facet of the $Sr_2IrO_4$ with the small thermal strains under 25 °C exhibit that the free energy difference between $\Delta G_{*OH}$ and $\Delta G_{*O}$ is maximum to be 1.85 eV, indicating that a minimum potential of 1.85 V has to be applied to make each step downhill in free energy. This means that the deprotonation of $*OH$ to form $*O$ is the rate-determining step (RDS) due to the strong adsorption of $*OH$ (Fig. 4a). However, the potential requirement at OER RDS on the $Sr_2IrO_4$ with big thermal strains under 90 °C was significantly reduced to 1.74 V ($\Delta G_{*OH} - \Delta G_{*O} = 1.74$ eV) due to the weakened adsorption of $*OH$ (Fig. 4b), resulting in a reduction in the OER overpotential. Indeed, recent theoretical studies on the OER mechanism for transition metal oxides have demonstrated that the $\Delta G_{*O} - \Delta G_{*OH}$ difference can effectively describe their OER activities[50,51]. Actually, this difference reflects the metal-oxygen binding strength[51]. The optimal catalysts are required to have a moderate metal-oxygen bond strength and thus a neither too big nor too small $\Delta G_{*O} - \Delta G_{*OH}$ difference. Our DFT calculations reveal that the thermal strains in $Sr_2IrO_4$ lead to a smaller $\Delta G_{*O} - \Delta G_{*OH}$, i.e., a smaller overpotential. This result suggested that the thermal strains in $Sr_2IrO_4$ can optimize the interactions between oxygen-containing OER intermediates and Ir active species. We provided the Bader charge analysis to check the interactions between Ir active site and OER intermediates. As shown in Supplementary Fig. 17, Bader charge analysis revealed that about 0.39–0.54 electrons were transferred from OER intermediates into Ir active site during the adsorption to proceed, confirming that the Ir active site is able to extract electrons from OER intermediates via their interactions. The higher electron transfer numbers for $*OH$ than $*OOH$ under 25–90 °C agree well with the Sabatier principle, that is, an ideal OER site should adsorb $*OOH$ weaker than $*OH$ for initial reactant capture and product desorption. Indeed, considering the operation conditions (including pH, potential, temperature, and solvation effects), the $Sr_2IrO_4$ with thermal strains still exhibited the similar Gibbs energy profile with the RDS for deprotonation of $*OH$ (Supplementary Figs. 18 and 19). The theoretical calculations exhibited the similar OER performances on both (001) and (110) facets of $Sr_2IrO_4$ with thermal strains, proving a plausible mechanism that the OER activities on $Sr_2IrO_4$ are mainly dominated by the thermally induced strain effects (Fig. 4 and Supplementary Fig. 20). These DFT results on the likely exposed facets further confirmed that the thermal strain in $Sr_2IrO_4$ is effective to accelerate OER, beyond the thermal diffusion.

The binding strengths of adsorbed oxygen-containing OER intermediates to active metal sites are closely related to the $d$-band center of transition metal oxides. Usually, a lower $d$-band center induces a weaker metal-oxygen binding. We also carried out the DFT calculations to check the changes in $d$-band center under thermal strains, establishing the plausibility of the mechanism as a thermally induced effect. The calculated density of states (DOS) of $Sr_2IrO_4$ exhibited that the $t_{2g}$ orbital is the main electronic states near the $E_f$

(Supplementary Fig. 21). The $d$-band center was calculated as the first statistical moment of the $d$-projected DOS. The thermal strain downshifts the Ir 5$d$ band of $Sr_2IrO_4$ away from the $E_f$, from −2.21 eV under 25 °C to −2.57 eV under 90 °C (Fig. 4c). The DFT results are in good agreement with the XPS valence band analysis. After the adsorption of OER intermediates onto the Ir active sites, the downshift of $d$ band center is still dominated by the thermal strains, demonstrating that the interactions between OER intermediates and Ir active site does not change the downshift trend of $d$ band center (Supplementary Figs. 22 and 23). For OER on $Sr_2IrO_4$, the metal-oxygen bond strength depends on the interactions between the valence states of oxygen-containing OER intermediates and the Ir 5$d$ states. After adsorption of oxygen-containing OER intermediates to Ir active species, their Ir-O orbital overlap will lead to the generation of the split Ir-O bonding ($\sigma$) and antibonding ($\sigma^*$) states. Usually, the $\sigma$ states, much lower than $E_f$, are fully filled, while the filling of the $\sigma^*$ states, above $E_f$, depends the energy level of these states relative to the $E_f$. The $d$-band center level ($\varepsilon_d$) is closely relative to the energy level of $\sigma^*$ states, thus affecting the electron filling of $\sigma^*$ states. As a consequence, the lower the $\varepsilon_d$ of the $Sr_2IrO_4$ with big thermal strains, the weaker the binding of $*OH$ to the Ir active species, because more electrons will fill the $\sigma^*$ states and be occupied when adsorption occurs between the $Sr_2IrO_4$ and OER intermediates[52].

The thermal strains accelerated proton-coupled electron-transfer process in OER was further demonstrated by dependence of pH on the onset potentials of OER. According to the Nernst equation, for an electrochemical reaction with highly solvated protons under ambient conditions (25 °C, 1 atm), the thermodynamic potential of the electrode shifts −59 mV per pH unit as the pH is increased. In our case, in alkaline media, the fast reaction rate of $10^{11}$ mol $L^{-1}$ $s^{-1}$ for $H_3^+O$ and $OH^-$ recombination induces that the OER kinetics is completely limited by sluggish deprotonation of $*OH$[53]. As a result, the dependence of pH on the OER onset potentials is −86 mV $pH^{-1}$ for $Sr_2IrO_4$ with small strains under 25 °C (Fig. 4d), significantly bigger than the theoretical value of −59 mV $pH^{-1}$, confirming the frustrated deprotonation of $*OH$. As heating to 90 °C, the slope for pH dependence of OER onset potentials is −69 mV $pH^{-1}$ (Fig. 4e), which is much close to the theoretical value of −72 mV $pH^{-1}$ under 90 °C and 1 atm for completely solvated protons, indicating that the OER RDS, deprotonation of $*OH$ to $*O$, on the $Sr_2IrO_4$ is effectively accelerated when heating to produce the big thermal strains under 90 °C. The thermal compressive strains in $IrO_6$ units downshift the $d$-band center, which would induce that the proton in $*OH$ tends to be solvated due to the weakened adsorption of $*OH$ to Ir active species under larger thermal strains. Therefore, we conclude that the thermally lowered Ir 5$d$ band center will weaken the binding strength between the OER intermediates and Ir active species and gives rise to the higher intrinsic OER activity of $Sr_2IrO_4$ under thermal stimulation[54,55]. It is worth pointing out that adjusting electronic states of $Sr_2IrO_4$ by heating is different from the previous strategies, such as electronic modifications by doping, defects, or distortions. In this case, the thermal field is utilized to tune the electronic states of $Sr_2IrO_4$ via thermal strain effect. Our results imply that it is possible to apply the materials with a positive catalytic contribution of thermal strains to OER activity for creating the efficient water splitting with the simultaneous input of heat and electricity.

## Discussion

In summary, we successfully demonstrate that the $d$-band center of $Sr_2IrO_4$ can be easily tuned by heating to trigger compressive strains in its $IrO_6$ units, thus optimizing the OER kinetics by adjusting the binding strength between OER intermediates and Ir active species. Completely different from the previous routes to produce the constant strains in crystals (Fig. 5a), the thermally induced strains are easily tuned by varying temperature. As an example, the thermal compressive strains in the $Sr_2IrO_4$ crystal with anisotropic thermal

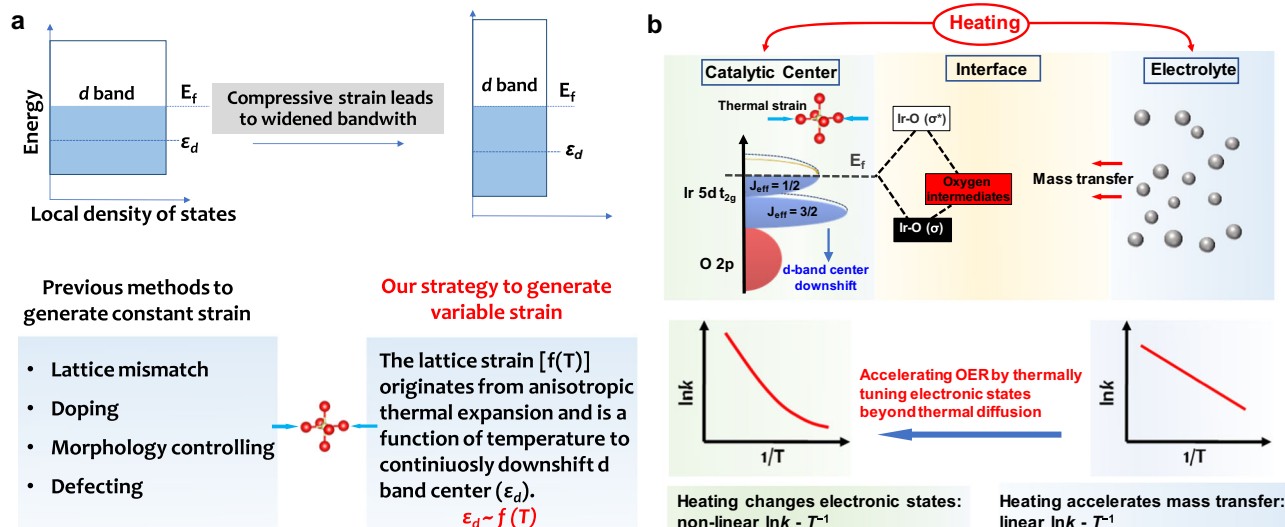

**Fig. 5 | Energetics at catalyst-electrolyte interface with thermal strain effect. a** A scheme to describe the effect of compressive strain on the d band of transition metals and our thermal stimulation strategy to generate variable lattice strain. **b** Effects of heating on the OER. The improved OER performances originate from that heating accelerates mass transfer and the thermal strains in $IrO_6$ units downshift the $d$-band center. The strong spin-orbital coupling in the low-spin $Ir^{4+}$ ($5d^5$, $t_{2g}^5 e_g^0$) $5d$ orbital in $Sr_2IrO_4$ splits the five electrons in the $t_{2g}$ band into four

electrons occupied as the $J_{eff} = 3/2$ sub-band and to one electron occupied as the $J_{eff} = 1/2$ sub-band. The thermal strains downshifting $d$-band center is real-timely sensitive to the temperature. The thermal diffusion accelerating OER follows a linear $\ln k \cdot T^{-1}$ Arrhenius relationship and the thermal strain accelerating OER follows the nonlinear $\ln k \cdot T^{-1}$ Arrhenius relationship due to that downshifting $d$-band center will adjust the Ir-O bonding ($\sigma$) and antibonding ($\sigma^*$) states to meet the Sabatier principle.

expansion will effectively regulate the electronic energetics at electrode-electrolyte interface (Fig. 5b), downshifting the $d$-band center to accelerate the OER kinetics and breaking the linear Arrhenius relationship to achieve higher energy conversion efficiency. Our findings indicated that heating is a possible route to tune the electronic states of materials, which may exhibit a positive catalytic contribution to OER kinetics, beyond the traditional thermal diffusion effect.

## Methods

### Synthesis of $Sr_2IrO_4$ and $SrIrO_3$

$IrO_2$ was purchased from Shanghai Macklin Biochemical Co., Ltd (China). $SrCO_3$, KOH, and ethanol were purchased from Shanghai Aladdin Bio-Chem Technology Co., Ltd., (China). All chemicals were analytical grade and used as obtained without further purification. $Sr_2IrO_4$ was produced by a solid-state reaction. Firstly, stoichiometric $IrO_2$ (96 mg) and $SrCO_3$ (127 mg) were dispersed in a mortar in ethanol medium by hand grinding for 40 min. The mixed powders were in turn calcined in the air at 900 °C for 24 h, 1000 °C for 24 h, and 1100 °C for 24 h with intermediate grindings. The heating rate was 2 °C/min. The as-prepared sample was finely ground using mortar for subsequent use. For comparison, $SrIrO_3$ was fabricated by a solid-state reaction. The mixed powders of $IrO_2$ (192 mg) and $SrCO_3$ (127 mg) were pretreated by the same procedure as the $Sr_2IrO_4$ and then were calcined at 900 °C for 10 h in the air with a heating rate of 2 °C min$^{-1}$ for forming the $SrIrO_3$.

### Characterizations

The crystal structures of prepared samples were characterized by X-ray diffraction (XRD, Bruker D8 Advance, Cu Kα radiation) at 40 kV and 40 mA from 10° to 70°. The morphologies and microstructures of the as-prepared sample were examined with a field-emission scanning electron microscope (SEM, Nova NanoSEM 230, FEI) and a transmission electron microscope (TEM, FEI Tecnai F20). X-ray photoelectron spectroscopy (XPS, PHI5000 Versa Probe, ULVAC-PHI, Japan) was carried out to check compositions and chemical states with monochromatized Al Kα excitation (the binding energies was corrected by

normalizing the C 1 s spectrum at 284.6 eV, and a Shirley background was used for peak fitting). Raman spectra were obtained with a Raman system (Horiba HR800) operating with a 20 mW air cooled argon-ion laser (514 nm) as the excitation light source with a heating module (Hotz, RT1000). The X-ray absorption fine structure spectroscopy (XAFS) including X-ray absorption near-edge structure (XANES) and extended X-ray absorption fine structure (EXAFS) of Ir L$_3$-edge was collected at the Beamline of TLS07A1 in National Synchrotron Radiation Research Center (NSRRC), Taiwan.

### Electrochemical measurements

All electrochemical experiments were carried out on the CHI 660E electrochemical workstation (Shanghai Chenhua Science Technology Corp., Ltd., Shanghai, China) using a standard three-electrode configuration in a 1.0 M KOH solution under 20 to 90 °C. The reference electrode was Hg/HgO (1.0 M KOH), and the counter electrode was platinum plate. To prepare the working electrode, the as-prepared powders (3 mg) were dispersed in ethanol (600 μL) by sonication for 20 min and then the carbon black (1 mg) and Nafion (60 μL, 0.25 Wt. %, DuPont) were added with another sonication for 20 min to prepare the catalyst ink. Next, the resulting ink (40 μL) was dropwise loaded on the carbon paper (0.5 cm × 1 cm) to obtain the working electrode. After drying at ambient conditions, the working electrode could be used for the electrochemical study. A MS7-H550-Pro magnetic hotplate stirrer (Dragon Lab, China) with a thermocouple was used to heat and monitor the temperatures of the electrolyte. Linear sweep voltammetry (LSV) curves were recorded by positive-going potential scan from 1.20 to 1.56 V vs. Reversible hydrogen electrode (RHE), with 90% $iR$-drop compensation at a scan rate of 5 mV/s. The $iR$ compensation was performed by on-the-fly correction, an automatic correction mode based on positive feedback[56,57]. To calculate electrochemically active surface areas (ECSA), double-layer capacitance ($C_{dl}$) values were measured by cyclic voltammetry (CV) method in the non-Faradaic region from 0.9 to 1.0 V vs. RHE at the scan rate of 20, 40, 60, 80, and 100 mV s$^{-1}$. The current density at 0.95 V vs. RHE was used for the calculations of $C_{dl}$, which equals to $(j_a - j_c)/(2\,v)$, where $j_a$ and $j_c$ are the anodic current density and cathodic current density, respectively. The

ECSA was calculated by ECSA = $C_{dl}/C_s$, where $C_s$ is the specific capacitance of an ideal flat surface of the electrode. Here, we use the average value of 0.04 mF cm$^{-2}$ for $C_s$ in alkaline solutions without taking the used material and measurement conditions into account. All measured potentials in this work were converted to RHE according to the Nernst equation of $E_{RHE} = E_{Hg/HgO} + (2.303RT/nF)pH + \Delta E_{Hg/HgO,T}$, where $E_{Hg/HgO}$ is the measured potential, $R$ is the molar gas constant, $T$ is the absolute temperature, pH is the pH value of the electrolyte, $F$ is the Faraday constant (96485 C mol$^{-1}$), $n$ is the number of electrons transferred ($n = 1$ for hydrogen), and $\Delta E_{Hg/HgO,T}$ is the temperature-dependent potential deviation of the Hg/HgO electrode. Considering the $\Delta E_{Hg/HgO,T}$ and pH values being temperature-dependent, they were experimentally corrected by detecting the open circuit voltage in two-electrode system comprising of a Hg/HgO electrode and a standard RHE at the given temperatures (Supplementary Table 7). To confirm stability of the Hg/HgO reference electrode, we checked the potential of Hg/HgO reference electrode before and after operating under different temperatures by testing the open-circuit voltage between Hg/HgO electrode and the standard RHE under room temperature (Supplementary Fig. 24). The temperature-dependent theoretical thermodynamic water splitting potential ($E^o_{H2O}$) is calculated by an empirical equation of $E^o_{H2O} = 1.229 - 0.9 \times 10^{-3} (T - 298.15)$[39]. The OER overpotential ($\eta$) was calculated according to the $\eta = E_{RHE} - E^o_{H2O}$ at the current density of 10 mA cm$^{-2}$.

## DFT calculations

All the density functional theory (DFT) calculations were performed with the Vienna Ab-initio Simulation Package (VASP)[58]. The crystal structure of the tetragonal Sr$_2$IrO$_4$ was derived from ref. 59, which belongs to a space group of $I4_1/acd$. To simulate the thermal strain in IrO$_6$ units of Sr$_2$IrO$_4$, the structural model of the Sr$_2$IrO$_4$ was built by setting the unit cell parameters from XRD refinement at 20 and 90 °C and the atomic fractional coordinates from Ir-O bonding length in IrO$_6$ units from EXAFS analysis at 20 and 90 °C. The lattice parameters of Sr$_2$IrO$_4$ are $a = b = 5.50$ Å, $c = 25.74$ Å, for Sr$_2$IrO$_4$ under 25 °C, and $a = b = 5.51$ Å, $c = 25.72$ Å for Sr$_2$IrO$_4$ under 90 °C. Under 25 °C, the Ir-O$_{ab-plane}$ and Ir-O$_{c-axis}$ bond lengths are 1.989 Å and 1.993 Å, respectively. Under 90 °C, the Ir-O$_{ab-plane}$ and Ir-O$_{c-axis}$ bond lengths are 1.938 Å and 1.937 Å, respectively. To determine the Gibbs free energy changes ($\Delta G^*$) of the OER intermediates adsorbed onto the Ir site of the (001) or (110) surface of the Sr$_2$IrO$_4$, we constructed the slab model with five atomic layers in a 1 ×1 unit cell with inversion-symmetry to avoid net dipoles in the cell. The vacuum regions of 15 Å were used to avoid periodic image interactions, that is, there is enough vacuum space so that the electron density of the material tails off to zero in the vacuum and the top of one slab has essentially no effect on the bottom of the next. The structure optimization was performed by fixing the bottom three layers to mimic the bulk region and allowing the top two layers to relax to mimic the surface[60]. Owing to the single Ir 5-coordinated environment on both (001) and (110) facets (Supplementary Fig. 14) and the satisfied stability of Sr$_2$IrO$_4$ during OER (No surface reconstruction after OER $i$-$t$ test at 1.45 V at 90 °C for 1 h is observed by the HRTEM in Fig. 1a), we calculated the OER activity on a single active site adsorbate evolution mechanism (AEM) model[41]. Owing to that the surface hydroxylation of oxides is a thermodynamically favorite process in the alkaline environment, we performed the hydroxyl group passivating the coordinately unsaturated Ir sites except the Ir active site on the slab model to stabilize the surface structure, thus avoiding the undesired surface reconstruction during structure optimization. The single-site AEM model with 100% coverage of adsorbates on Ir active site was used to check the intrinsic catalytic activity (100% coverage is one absorbate per one active site)[61], thus establishing the plausibility of the thermal strains enhanced OER on the likely exposed (001) and (110) facets under the plausible adsorbates scenery.

The exchange-correlation functional was treated by the generalized gradient approximation with the Perdew−Burke−Ernzerh (GGA-PBE) functional[62]. In describing the transition metal oxides, the PBE with inclusion of a Hubbard-type effective on-site Coulomb term ($U$), which is computationally inexpensive, often provides a computationally tractable and physically reasonable approach for transition metal oxides[63]. In particular, the PBE functional has demonstrated to be effective in solving $d$ band center position for Ir$^{4+}(5d^5)$-based oxides[1,64]. Usually, for transition-metal oxides, the localized $d$ states yield strongly correlated bands with a on-site Coulomb repulsion $U$[65]. In our case of Sr$_2$IrO$_4$, the $U$ value of 2 eV was adopted to consider the on-site $d$-electron Coulomb interaction effect for partly overcoming the DFT (GGA under the PBE functional) delocalization error[64]. In addition, the Sr$_2$IrO$_4$ with an electron configuration of $5d^5$ exhibits the strong spin-orbit coupling (SOC) effects. At this point, it is natural to consider the SOC effects in Sr$_2$IrO$_4$ since the SOC effects is much larger than that in 3$d$ and 4$d$ oxides[35,66]. The valence configurations of the pseudopotentials were $4s^2\ 4p^6\ 5s^2$ for Sr, $5d^8\ 6s^1$ for Ir, and $2s^2\ 2p^4$ for O. The plane wave cutoff energies were set to 520 eV and the threshold of self-consistent-field energy convergence was set to $10^{-6}$ eV[59] and sampled the irreducible Brillouin zone by a set of $4 \times 4 \times 1$ k-points. The density of states (DOS) projected onto the d-states can be characterized by the d-band center ($\varepsilon_d$), which can be calculated by the Eq. (1):

$$\varepsilon_d = \frac{\int n_d(\varepsilon)\varepsilon d\varepsilon}{\int n_d(\varepsilon)d\varepsilon} \tag{1}$$

where $n_d(\varepsilon)$ is the density of states.

The DOS computed within PBE + $U$ + SOC is comparable with the experimental XPS valence band spectrum (Supplementary Fig. 21a), suggesting that the PBE + $U$ + SOC is acceptable for the theoretical calculations of Sr$_2$IrO$_4$. Indeed, the metal-like electronic states of Sr$_2$IrO$_4$ with thermal strain computed within PBE + $U$ + SOC is in good agreement with an experimental fact that the Sr$_2$IrO$_4$ at 20−90 °C is paramagnetic state (Néel temperature at 240 K)[67]. This conclusion is also confirmed in the density of states of IrO$_2$ computed within PBE + $U$ + SOC[64].

The following Eqs. (2)−(4) were employed to calculate the free energy changes:

$$\Delta G_{*OH} = (E_{*OH} + 0.5 \times E_{H2} - E_{H2O} - E_*) + (ZPE_{*OH} + 0.5 \times ZPE_{H2} - ZPE_{H2O} - ZPE_*) - T \times (S_{*OH} + 0.5 \times S_{H2} - S_{H2O} - S_*) \tag{2}$$

$$\Delta G_{*O} = (E_{*O} + E_{H2} - E_{H2O} - E_*) + (ZPE_{*O} + ZPE_{H2} - ZPE_{H2O} - ZPE_*) - T \times (S_{*O} + S_{H2} - S_{H2O} - S_*) \tag{3}$$

$$\Delta G_{*OOH} = (E_{*OOH} + 1.5 \times E_{H2} - 2 \times E_{H2O} - E_*) + (ZPE_{*OOH} + 1.5 \times ZPE_{H2} - 2 \times ZPE_{H2O} - ZPE_*) - T \times (S_{*OOH} + 1.5 \times S_{H2} - 2 \times S_{H2O} - S_*) \tag{4}$$

where $E$, ZPE, and $S$ are the total energy, zero-point energy, and entropy of intermediates, respectively.

## Data availability

All data generated or analyzed during this study are included in this published article and its supplementary information files or by download from the Figshare Dataset (https://doi.org/10.6084/m9.figshare.24081492). All other data are available from the corresponding author upon request. Source data are provided with this paper.

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

## Acknowledgements

This work is supported primarily by the Scientific and Technological Innovation Project of Carbon Emission Peak and Carbon Neutrality of Jiangsu Province (No.BE2022028-1), the National Natural Science Foundation of China (Grant Nos 52272217, 22372078, 51872135, 51572121, and 21633004).

## Author contributions

Y.D. and F.X. contribute equally to this work. S.Y. and Y.D. conceived the original concept, designed the experiments and wrote the manuscript. Y.D. and F.X. performed most characterizations and analysis. R.L. finished the temperature-dependent X-ray diffraction characterization and analysis. M.L., W.L., and Y.Y. discussed characterizations and electrochemical testing results. Z.Z. discussed the experimental ideas. All authors discussed the results.

## Competing interests

The authors declare no competing interests.
