## [Peer Review File · Nature Communications]

REVIEWER COMMENTS

Reviewer #1 (Remarks to the Author):

In this work, the strain in OER catalyst was produced by a thermophysical effect, anisotropic thermal expansion. Totally different from the previous strategies to create constant strains in the catalysts, the thermal strain is real-time tuned by the temperature variation. Given that downshifting the d-band center of transition metal oxides by compressive strain is an efficient route to accelerate the sluggish OER kinetics, the temperature-adaptive compressive strain is really a big leap for development of high-activity OER materials with energy coupling properties. The authors have provided various solid experimental evidences, such as temperature-dependent XRD, XPS, Raman, and XAFS, to confirm the generation and effects of thermal strains on OER performances. In addition, the nonlinear Arrhenius relationship was proposed to be the criterion to judge the thermal contribution in controlling electronic states of materials, which is able to guide the development of materials with thermophysical effects. This study opens a new avenue to achieve water splitting by multienergy coupling and it could be recommended to publish after the authors considering the following points.

1. It is well-known that heating stimulates lattice vibration and causes thermal expansion in most materials, which may also cause lattice strain. Why did you choose this unusual material, Sr₂IrO₄, with anisotropic thermal expansion effect?
2. The novelty of this work must be further emphasized, highlighting the roles of thermophysical effect. In addition, the “high-activity OER” of Sr₂IrO₄ is the least important aspect of the paper, and should be put into better context by comparing to known reported systems.
3. The XRD pattern of Sr₂IrO₄ in supplementary Fig. 1 is different from JCPDS Card. Why is there an extra peak at about 25°.
4. The Tafel slope at each temperature point was reported in this manuscript. We recommend that the authors show the Tafel slope as a function of temperature, which is a direct comparison with the non-linear $\ln k - T^{-1}$ relationship for the readers to understand the effects of thermal strains on OER kinetics.
5. As an initial demonstration, it is expected to deeply discuss the availability and advantages of the heat-electricity coupling water splitting.

6. The intrinsic activity should play an important role in electrocatalysis, so it is more valuable to take this into account, in-depth understanding the real electrocatalytic process in this work.

Reviewer #2 (Remarks to the Author):

This manuscript demonstrate that the d-band center of Sr₂IrO₄ can be easily tuned by heating to trigger compressive strains in its IrO₆ units, thus optimizing the OER kinetics by adjusting the binding strength between OER intermediates and Ir active species. Furthermore, this manuscript indicated that thermal contributions to OER kinetics are not limited to a mass transfer effect by thermal diffusion and the heating is an effective route to continuously and real-timely tune the electronic states of materials. The manuscript is well written and the results are very useful for the scientific community. Hence, this manuscript can be suitable for publication in Nature Communications after major revision.

Additional comments:

(1) In the three-electrode configuration (1 M KOH), is the reference electrode still the Hg/HgO electrode? As far as I know, the Hg/HgO reference electrode can only stably work at a temperature below 40 °C. How did the author consider the instability of the Hg/HgO electrode at 90°C?

(2) It would be better to provide the parameters used to fit the XPS spectra (Figure 2e,f) to increase the reliability of deconvolution results.

(3) The lack of error bars is mind-boggling; especially when a nonlinear $\ln j_0 - T^{-1}$ relationship is claimed in Figure 3C. The activity values should be based on at least three separate measurements with associated standard deviations.

(4) The authors state that the activity was measured with LSV at a scan rate of 5 mV/s with 90% iR-drop compensation. However, information on potential range, sweep direction, rotations rate, number of scans etc. are not available. In addition, it should be stated how the iR-drop compensation was performed.

(5) The crystal structure of as synthesized SrIrO₃ should be claimed, given that SrIrO₃ contains multiple possible crystal phases (e.g, 3C phase, 6H phase).

Reviewer #3 (Remarks to the Author):

In the reviewed manuscript, the authors propose to use thermal energy to continuously tune internal strains in catalysts that exhibit anisotropic thermal expansion, as a route to modulate their electronic

states, with the aim of optimizing their catalytic performance. The approach is exemplified investigating the oxygen evolution reaction (OER) on the layered iridate oxide Sr₂IrO₄. It is shown that compression strain on the IrO₆ octahedron of this material can be thermally induced, which modify the electronic state of the active iridium (downshifting the d-band center), modulating (weakening) its interaction with the involved intermediates (*OH, *O and *OOH), favoring the OER reaction (reducing the rate determining step height, and thus the overpotential) beyond simply favoring thermal diffusion. It is found that, at 90° C, the Ruddelsden-Popper phase of Sr₂IrO₄, as catalyst for the OER, outperforms (in terms of overpotential) most of the known iridate catalysts for this reaction (although the optimal operation temperatures for these ones have not been explored). The reported results and mechanisms being interesting, I have doubts about the relevance of the proposal. I am concerned about the modeling. At least the presentation of the introductory part of the manuscript and the conclusions should be improved.

Understanding that mechanical strain has the potential of modifying electronic states in materials, and that temperature in materials exhibiting anisotropic thermal expansion can induce internal mechanical strain, using thermal energy to optimize the catalytic performance of materials having this property does not seem a particularly original proposal. In fact, knowing the potential relationships between mechanical strain and electronic state modulation, and between electronic states of catalysts and catalytic performance, even improving performance beyond simply favoring thermal diffusion cannot be considered as a surprise. On the one hand, given the enormous complexity and subtlety of the chain of effects involved, it does not seem that the proposed optimization approach can be easily designed from mechanisms. Moreover, in general, detailed knowledge of thermal induced effects and mechanisms on materials and reactions can hardly be transferred to other materials and reactions, which would be particularly true for complex reactions involving multidentate intermediates. On the other hand, thermally induced effects, even on complex catalysts and reactions, can be easily exploited, in practice, even without knowing them, by simply exploring experimentally the degree of freedom operation temperature of the catalytic configuration. Thus, perhaps the most valuable contribution of the manuscript would be that it provides evidence that the performance of a catalytic process involving a catalyst exhibiting anisotropic thermal expansion can be significantly increased, even beyond simply favoring thermal diffusion, by operating at a higher temperature. Therefore, the main conclusion seems to be that “it is worthwhile to explore the degree of freedom operation temperature of a catalytic configuration”, which seems obvious. This said, the reported results and mechanisms are interesting.

By means of XRD, SEM, HRTEM, and HAADF-TEM, it is concluded that single crystal nanoparticles from 200 to 300 nm of Sr₂IrO₄ were obtained. However, the shape of the nanocrystals is not identified. Therefore, the dominant crystalline plane exposed by the catalyst cannot be established, reason why the experimental and computational results cannot be reliably correlated. Citing Chem. Eng. J. 423 (2021) 130185 (Ref 41), OER on the Sr₂IrO₄ (001) surface is computationally investigated. However, the (001) surface focus in such a reference is not clearly supported. The shape of the nanoparticles in Ref. 41 is not identified. Moreover, from the comparison between Fig. S2c of Ref. 41 and Fig. S1a of the revised manuscript, similar shapes cannot be concluded. The assumed adsorbates scenery (species, coverages, sites...), under the different operation conditions (pH, potential, temperature...), should be discussed.

Solvation effects should be included. The approach of using experimental lattice parameters to accurately simulate thermally induced strain under a specific model chemistry should be validated. The applied numerical treatment (DFT + U under the PBE functional) should be validated for the application (energy profile and DOS of the OER on bimetallic earth-alkaline/transition metal oxides). Better figures displaying the investigated configurations should be provided.

The second sentence in the abstract section (lines 14-17) is too long, making it difficult to understand. Assuming that the strain is sensitive to the temperature, which can vary continuously, the constructions "...thermal-triggered continuous..." in line 19 (and 49), or "...sensitive to the temperature variation..." in line 48 (also 155 and 269), are confusing to me. "identifying" in line 20 could be not the best choice. It should be explained the sense in which strain engineering is "efficient" in optimizing bands (lines 28-30). The content from line 44 to 47 should be presented in a more concise form. I think that the last sentence in the last paragraph of the introductory section of the manuscript (lines 51 to 53) is a bit triumphalist. Significant thermally induced improvements could be limited to only a few reactions on some catalysts from among those exhibiting anisotropic thermal expansion. Sentences from line 271 onwards, up to the end of the conclusions section, seem too vague. Labeling in Fig. 4 caption is not correct.

Responses to the reviewer's comments

Reviewer #1 (Remarks to the Author)

In this work, the strain in OER catalyst was produced by a thermophysical effect, anisotropic thermal expansion. Totally different from the previous strategies to create constant strains in the catalysts, the thermal strain is real-time tuned by the temperature variation. Given that downshifting the d-band center of transition metal oxides by compressive strain is an efficient route to accelerate the sluggish OER kinetics, the temperature-adaptive compressive strain is really a big leap for development of high-activity OER materials with energy coupling properties. The authors have provided various solid experimental evidences, such as temperature-dependent XRD, XPS, Raman, and XAFS, to confirm the generation and effects of thermal strains on OER performances. In addition, the nonlinear Arrhenius relationship was proposed to be the criterion to judge the thermal contribution in controlling electronic states of materials, which is able to guide the development of materials with thermophysical effects. This study opens a new avenue to achieve water splitting by multienergy coupling and it could be recommended to publish after the authors considering the following points.

1. It is well-known that heating stimulates lattice vibration and causes thermal expansion in most materials, which may also cause lattice strain. Why did you choose this unusual material, Sr_2IrO_4 , with anisotropic thermal expansion effect?

Reply: Thanks for your question. Generally, rising temperature, most materials exhibit volume expansion with the positive coefficient of thermal expansion (CTE). The volume expansion is a result of inharmonious longitudinal vibrations of lattice atoms under thermal stimulation. In this case, the thermal expansion occurs via inharmonious vibrations along the chemical bond, which leads to that this thermal effect cannot produce a compressive strain in the MO_6 units. However, when there are inharmonious transverse vibrations in the lattice of materials, the M-O-M chemical bonds can bend or rotate. Such motions will lead to the anisotropic thermal expansion, thus providing a possibility to generate compressive strains in crystals. In our case, the Sr_2IrO_4 is a layered Ruddelsden-Popper compound, in which lattice the transverse vibration of bridging oxygen between IrO_6 units induced that the lattice contracts along c-axis and the lattice expands along a,b-axes. The anisotropic thermal expansion is sensitive to the temperature changes and results in a compressive strain in Sr_2IrO_4 . Accordingly, as a concept demonstration, we adopted the Sr_2IrO_4 as the model material to verify the anisotropic thermal expansion effects on the electronic states of OER catalysts.

2. The novelty of this work must be further emphasized, highlighting the roles of thermophysical effect. In addition, the “high-activity OER” of Sr_2IrO_4 is the least important aspect of the paper, and should be put into better context by comparing to known reported systems.

Reply: Thanks for your suggestions. As pointed out by the Reviewer, our study focused on the thermophysical effect of catalyst and its contributions to the OER performance. So, we further emphasized the roles of thermophysical effect in the Abstract and Conclusion parts in the revised manuscript.

Indeed, the OER performances of all the known Ir-based catalysts were reported at room temperature. Therefore, we listed the room-temperature OER performances in **Supplementary Table 6** to show that the room-temperature OER performances of the Sr_2IrO_4 prepared in our

study are comparable with the reported results. To underline the thermophysical effects, we compared heating OER on the SrIrO₃ without thermal compressive strain and the Sr₂IrO₄ with thermal compressive strain.

We added the following sentences into the main text:

The OER performances of the Sr₂IrO₄ under room temperature is comparable with the known reported Ir-based catalysts (Supplementary Table 6). Notably, under heating, the OER performances of the Sr₂IrO₄ increase significantly.

Supplementary Table 6. OER performances for Ir-based materials reported

Electrodes	Current density (mA cm ⁻²)	Overpotential (mV)	Electrolyte	Operation temperature	Ref.
Sr ₂ IrO ₄	10	312	1 M KOH	Room temperature	This work
		235	1 M KOH	90 °C	This work
SrIrO ₃	0.1	270	0.1 M KOH		4
Sr ₄ IrO ₆		287	0.1 M HClO ₄		
Sr ₂ IrO ₄		286	0.1 M HClO ₄		5
SrIrO ₃		353	0.1 M HClO ₄		
Sr ₂ IrO ₄	10	300	0.1 M HClO ₄	Room temperature	7
		280	0.5 M H ₂ SO ₄		
SrIrO ₃		290	1 M KOH		9
SrIrO ₃		300	1 M KOH		10
IrO ₂		295	0.1 M HClO ₄		11

3. The XRD pattern of Sr₂IrO₄ in Supplementary Fig. 1 is different from JCPDS Card. Why is there an extra peak at about 25°.

Reply: Thanks for your question. The XRD peak at 26.38° is attributed to the (002) plane of graphic carbon. In addition, the XRD peak at 54.54° is attributed to the (004) plane of graphic carbon. This because we dropped the Sr₂IrO₄ ink onto the carbon paper to evaluate its OER performances. The XRD analysis was carried out to check the structures and compositions of Sr₂IrO₄/carbon paper electrode, thus the peak of carbon paper at 26.38° and 54.54°, corresponding to the (002) and (004) crystal plane, respectively, was detected. To confirm this fact, we added the XRD pattern of carbon paper into **Supplementary Fig. 1b**. Clearly, the strong XRD peak of (002) and (004) facet was observed.

Supplementary Fig. 1 | a, XRD patterns for carbon paper and Sr_2IrO_4 coated on the carbon paper before and after electrochemical activation by i-t test at 1.45 V at 90 °C for 1 h in 1 M KOH. The XRD peak at 26.38° and 54.54° are assigned to the (002) and (004) crystal plane of graphitic carbon, respectively.

4. The Tafel slope at each temperature point was reported in this manuscript. We recommend that the authors show the Tafel slope as a function of temperature, which is a direct comparison with the non-linear $\ln k-T^{-1}$ relationship for the readers to understand the effects of thermal strains on OER kinetics.

Reply: Thanks for your suggestion. According to the Reviewer's suggestions, we show the Tafel slope as a function of temperature in **Fig. 3b** for Sr_2IrO_4 and in **Supplementary Fig. 9** for SrIrO_3 . Obviously, for Sr_2IrO_4 , as rising temperatures, the Tafel slope exhibited a similar variation tend to the $\ln k-T^{-1}$ relationship, suggesting that the OER kinetics of Sr_2IrO_4 are temperature-dependent. However, the Tafel slope of SrIrO_3 is nearly independent of temperature, meaning that the thermally-promoted OER kinetics of SrIrO_3 are totally originated from the thermal diffusion.

We added the following sentences into the main text:

The significantly reduced Tafel slopes under higher temperatures indicate that the thermal strains accelerated OER kinetics beyond the thermally induced mass transfer.

Fig. 3 | OER performances of Sr₂IrO₄ with thermal strains. **b**, Temperature-dependent Tafel slope of Sr₂IrO₄.

Supplementary Fig. 9 | **c**, Temperature-dependent Tafel slope of SrIrO₃.

5. As an initial demonstration, it is expected to deeply discuss the availability and advantages of the heat-electricity coupling water splitting.

Reply: Thanks for your deeply thinking. According to the mechanism of water splitting, the materials with thermophysical effects are able to create the efficient energy conversion with simultaneous input of electricity and heat. The advantages of electricity-heat coupling water electrolysis are as follows: 1) Heat partially replaces electricity to compensate the inherent heat requirement of water splitting; 2) Heat reduces the thermodynamic potential of water electrolysis; 3) Heating changes the electronic states of materials, greatly lowering the electron transfer barriers during OER. More importantly, the heat can be easily accessed from geothermy by heat pumps, solar heat by concentrating sunlight, or industrial waste heat. Therefore, the various heat-electricity coupling routes can be created to split water, such as the combination of photovoltaic and solar thermal (**Figure 1 for review**).

It is worth pointing out that the design concept of electrode materials for heat-electricity

coupling water splitting is totally different from the single electricity driven water splitting. For single electricity driven water splitting, the strategies to tune electronic states of electrode materials mainly include substituting with foreign elements, generating defects, and inducing lattice distortion, named direct modifications. In this route, the goal of regulating properties of materials is to efficiently response the electricity input. However, adjusting electronic states of materials by energy stimulation requires that materials have the physical effects. A few strategies, such as the light coupling with electricity to create photoelectrochemical catalysis, magnetism coupling with light or electricity to achieve spin catalysis, were developed to achieve a multienergy coupling water splitting. In these systems, the required electrical energy input was partially replaced by light or magnetism. Accordingly, it can be predicted that the various energy fields, including heat, light, or magnetism, are available to couple with the electricity though they trigger the thermo/photo/magnetophysical effects of catalytic materials.

Figure 1 for review. A typical heat-electricity coupling water splitting device driven by photovoltaic and solar thermal.

We added the following sentences into the main text:

It is worth pointing out that adjusting electronic states of electrode materials by heating is different from the previous strategies, such as electronic modifications by doping, defects, or distortions. In this case, the thermal field is utilized to tune the electronic states of materials via thermal strain effect. Our results imply that the materials with thermophysical effects can be applied to create the efficient water splitting with the simultaneous input of heat and electricity.

6. The intrinsic activity should play an important role in electrocatalysis, so it is more valuable to take this into account, in-depth understanding the real electrocatalytic process in this work.

Reply: In our case, the temperature-dependent OER activities of Sr_2IrO_4 were normalized with electrochemically active surface area (ECSA), a standard methodology to show the electrochemical activity of catalysts if the entire surface area of the electrode is electrocatalytically active (*ACS Nano* 2018, 12, 9635-9638; *ACS Catal.* 2018, 8, 6560-6570).

The CV scans were carried out in a non-faradaic potential region of different electrodes. The current density at middle potential was used for the calculation of the ECSA. The electrical double-layer capacitance (C_{dl}) was calculated by the equation $C_{dl} = (j_a - j_c)/2v$, where j_a and j_c are

the anodic current density and cathodic current density, respectively, and v is the scan rate. Thus, C_{dl} is the slope of the linear relationship between $(j_a - j_c)/2$ and scan rates. The ECSA can be calculated by $ECSA = C_{dl}/C_{dl, Ref}$, where $C_{dl, Ref}$ is the specific capacitance of an ideal flat surface of the catalyst. Here, we use the average value of 0.04 mF cm^{-2} for $C_{dl, Ref}$ in alkaline solutions without taking the used material and measurement conditions into account. The calculated ECSA is $2.29\text{-}2.68 \text{ mF cm}^{-2}$ when rising temperature from 30 to $90 \text{ }^\circ\text{C}$, indicating that the number of active species is nearly independent on temperature variation. Therefore, as shown in **Fig. 3a**, the normalizing the OER activities with ECSA is a reasonable method to show the intrinsic activity of Sr_2IrO_4 .

Fig. 3a, LSV of Sr_2IrO_4 in 1.0 M KOH under different temperatures normalized with ECSA.

Reviewer #2 (Remarks to the Author):

This manuscript demonstrate that the d-band center of Sr_2IrO_4 can be easily tuned by heating to trigger compressive strains in its IrO_6 units, thus optimizing the OER kinetics by adjusting the binding strength between OER intermediates and Ir active species. Furthermore, this manuscript indicated that thermal contributions to OER kinetics are not limited to a mass transfer effect by thermal diffusion and the heating is an effective route to continuously and real-timely tune the electronic states of materials. The manuscript is well written and the results are very useful for the scientific community. Hence, this manuscript can be suitable for publication in Nature Communications after major revision.

Additional comments:

1. In the three-electrode configuration (1 M KOH), is the reference electrode still the Hg/HgO electrode? As far as I know, the Hg/HgO reference electrode can only stably work at a temperature below 40 °C. How did the author consider the instability of the Hg/HgO electrode at 90 °C?

Reply: Thanks for the comment. In the three-electrode configuration (1 M KOH), the Hg/HgO electrode is the reference electrode. Indeed, for the commercial Hg/HgO electrode, the suggested operation temperature is below 40 °C. This because the higher temperature will increase the risk of the electrode damage, mainly resulting from the damage of porous ceramic tip, as shown in **Figure 2 for review**. That is, the tip of Hg/HgO electrode is composed of a porous ceramic plug that is inserted into the glass tube. At $T > 40$ °C, the big thermal expansion difference between porous ceramic plug and glass tube will increase the risk of porous ceramic plug to be broken. Indeed, the service life of Hg/HgO electrode is reduced to a great extent due to the damage of porous ceramic plug after multiple repeating (it would be damaged within several months).

Figure 2 for review. **a**, The optical photograph of Hg/HgO reference electrode. **b**, The optical photograph of the broken Hg/HgO reference electrode. The insets show the broken porous ceramic tip.

In the 20-100 °C, the Hg is thermally stable with a low vapor pressure (0.16 Pa at 20 °C, 36.38 Pa at 100 °C) due to its high boiling point at 356.6 °C (J. Chem. thermodynamics 1972, 4,603-620). Therefore, the operation temperature below 40 °C is just for protecting the

electrode. In principle, the Hg/HgO electrode can be used at higher temperatures (*J. Appl. Electrochem.* **1987**, *17*, 505-514). The working principle of Hg/HgO electrode is based on the equilibrium reaction between mercuric oxide layer formed on mercury and hydroxide anion in alkaline solution as internal solution, that is, a electrode reaction of $\text{HgO} + 2\text{e}^- + \text{H}_2\text{O} \rightleftharpoons \text{Hg} + 2\text{OH}^-$. The electrode potential is given by equation of $E = 0.116 - (RT/F) \ln a_{\text{OH}^-}$. Obviously, rising temperature will change the potential of Hg/HgO electrode. All measured potentials in this work were converted to RHE according to the Nernst equation of $E_{\text{RHE}} = E_{\text{Hg/HgO}} + (2.303RT/nF)\text{pH} + \Delta E_{\text{Hg/HgO},T}$, where $E_{\text{Hg/HgO}}$ is the measured potential, R is the molar gas constant, T is the absolute temperature, pH is the pH value of the electrolyte, F is the Faraday constant (96485 C mol^{-1}), n is the number of electrons transferred ($n = 1$ for hydrogen), and $\Delta E_{\text{Hg/HgO},T}$ is the temperature-dependent potential deviation of the Hg/HgO electrode. Considering the $\Delta E_{\text{Hg/HgO},T}$ and pH values being reversibly temperature-dependent, they can be experimentally corrected by detecting the open circuit voltage in two-electrode system comprising of a Hg/HgO electrode and a standard RHE at the given temperatures under different temperatures.

To assure that the Hg/HgO reference electrode works well during rising temperature test, we measure the open-circuit voltage of the Hg/HgO reference electrode after each temperature point test. As shown in **Supplementary Fig. 17**, we confirmed the stability of reference electrode in the different-temperature LSV tests. The open-circuit voltage of the Hg/HgO reference electrode before and after heating was measured to further verify the reliable measurement of OER activity. The Hg/HgO electrode before and after high-temperature test was used as the reference electrode and the standard Hg/HgO electrode was used as the working electrode to form a circuit to detect the open-circuit voltage. The open-circuit potential difference between the Hg/HgO electrode before and after heating is within 2 mV, confirming that the Hg/HgO electrode is well-being.

Supplementary Fig. 17. The open-circuit voltage testing of the reference electrode before and after the different temperature LSV with the standard Hg/HgO as reference electrode at room temperature.

2. It would be better to provide the parameters used to fit the XPS spectra (Figure 2e,f) to increase the reliability of deconvolution results.

Reply: Thanks a lot for your suggestions. We have provided the fitting parameters of the XPS as shown in **Supplementary Tables 3 and 4**.

Supplementary Table 3. The fitting parameters of the Ir 4f XPS spectra.

Temperature	Element/ Transition	Binding Energy /eV	Peak Width FWHM /eV	Peak Area /eV counts
-------------	------------------------	-----------------------	------------------------	-------------------------

25 °C	Ir 4f _{5/2}	61.81	2.33	71308
	Ir 4f _{5/2} satellite	63.15	2.78	40123
	Ir 4f _{7/2}	64.61	2.33	53481
	Ir 4f _{7/2} satellite	65.95	2.78	30092
90 °C	Ir 4f _{5/2}	61.98	2.26	66216
	Ir 4f _{5/2} satellite	63.41	2.63	30909
	Ir 4f _{7/2}	64.78	2.26	49662
	Ir 4f _{7/2} satellite	66.21	2.63	23182

Supplementary Table 4. The fitting parameters of the valence band spectra.

Temperature	Element / Transition	Binding Energy / eV	Peak Width FWHM / eV	Peak Area / eV counts
25 °C	Ir 5d t _{2g} J _{eff} = 1/2	0.99	1.23	428.95
	Ir 5d t _{2g} J _{eff} = 3/2	3.76	2.30	8581.4
	Ir-O π band	8.17	5.86	14237
90 °C	Ir 5d t _{2g} J _{eff} = 1/2	1.03	1.29	383.53
	Ir 5d t _{2g} J _{eff} = 3/2	3.85	2.21	7208.9
	Ir-O π band	8.17	5.74	11552

3. The lack of error bars is mind-boggling; especially when a nonlinear $\ln J_0 - T^{-1}$ relationship is claimed in Figure 3C. The activity values should be based on at least three separate measurements with associated standard deviations.

Reply: Thanks a lot for your suggestions. We have provided the error bars from three separate data in Fig. 3c and supplementary Fig. 9d.

Fig. 3 | OER performances of Sr₂IrO₄ with thermal strains. c, Calculated $\ln J_0 - T^{-1}$ plot with nonlinear Arrhenius relationship.

Supplementary Fig. 9 | d, Calculated $\ln j_0-1/T$ plot with linear Arrhenius relationship.

4. The authors state that the activity was measured with LSV at a scan rate of 5 mV/s with 90% iR-drop compensation. However, information on potential range, sweep direction, rotations rate, number of scans etc. are not available. In addition, it should be stated how the iR-drop compensation was performed.

Reply: Thanks a lot for your suggestions. We have provided the detail information regarding the LSV test. Here, we recorded the electrochemical performances on a working electrode prepared by dropping Sr_2IrO_4 ink onto carbon paper, not the rotating disc electrode. In our case, the electrochemical measurements were not involved with the rotation rate and number of scans.

The IR compensation was performed on the CHI 660E electrochemical workstation (Shanghai Chenhua Science Technology Corp., Ltd., Shanghai, China) via on-the-fly correction, an automatic correction mode based on the positive feedback. The IR compensation mechanism was well described in *Anal. Chem.* 1986, 58, 517-523; *ACS Energy Lett.* 2023, 8, 1952-1958. In brief, during IR compensation, the system works about a test potential at which no faradaic reaction occurs. By examining the current response to a $\Delta E = 50$ mV potential increment step across this test point, the instrument first determines the amount of uncompensated resistance. Signal-averaged currents at 54 and 72 μs after the step edge are extrapolated backward to provide a zero-time current, which is $\Delta E/R_u$, where R_u is solution resistance. The system then stages the application of positive feedback by using a multiplying digital-to-analog converter as a digitally controlled potentiometer. A test for potentiostatic stability is made after each stage. This test is carried out by applying a 50-mV step across the test potential and watching for ringing in the output of the I/V converter. When this ringing reaches a defined threshold, the system attempts to stabilize the network capacitively. When full compensation is reached or when the system can take no further action without crossing the allowed ringing threshold, the process ceases.

We provided the detail information as follows:

Linear sweep voltammetry (LSV) curves were recorded by positive-going potential scan from 1.20 to 1.56 V vs. RHE, with 90% iR-drop compensation at a scan rate of 5 mV/s. The IR compensation was performed by on-the-fly correction, an automatic correction mode based on positive feedback (Anal. Chem. 1986, 58, 517-523).

5. The crystal structure of as synthesized SrIrO_3 should be claimed, given that SrIrO_3 contains multiple possible crystal phases (e.g., 3C phase, 6H phase).

Reply: Thanks a lot for your suggestions. We have provided the XRD pattern of SrIrO_3 . According to the XRD analysis, the SrIrO_3 is a typical 6H phase, which is in tune with the standard pattern of JCPDS-25-0897. The SrIrO_3 with 6H phase structure exhibits negligible anisotropic thermal expansion in 0-100°C.

Supplementary Fig. 7 | a, XRD patterns for 6H-phase SrIrO_3 under different temperatures and the standard pattern of JCPDS-25-0897.

Figure 3 for review. A typical structure for 6H-phase SrIrO_3 .

Reviewer #3 (Remarks to the Author):

In the reviewed manuscript, the authors propose to use thermal energy to continuously tune internal strains in catalysts that exhibit anisotropic thermal expansion, as a route to modulate their electronic states, with the aim of optimizing their catalytic performance. The approach is exemplified investigating the oxygen evolution reaction (OER) on the layered iridate oxide Sr_2IrO_4 . It is shown that compression strain on the IrO_6 octahedron of this material can be thermally induced, which modify the electronic state of the active iridium (downshifting the d-band center), modulating (weakening) its interaction with the involved intermediates ($^*\text{OH}$, $^*\text{O}$ and $^*\text{OOH}$), favoring the OER reaction (reducing the rate determining step height, and thus the overpotential) beyond simply favoring thermal diffusion. It is found that, at 90°C , the Ruddelsden-Popper phase of Sr_2IrO_4 , as catalyst for the OER, outperforms (in terms of overpotential) most of the known iridate catalysts for this reaction (although the optimal operation temperatures for these ones have not been explored). The reported results and mechanisms being interesting, I have no doubts about the relevance of the proposal. I am concerned about the modeling. At least the presentation of the introductory part of the manuscript and the conclusions should be improved.

Reply: We appreciate the time and effort that you dedicated to providing feedback on our manuscript and are grateful for the insightful comments on and valuable improvements to our paper.

According to the Reviewer's suggestions, we have revised the Introduction and the Conclusions as follows:

Abstract: Compressive strain, downshifting the d-band center of transition metal oxides, is an efficient way to accelerate the sluggish kinetics of oxygen evolution reaction (OER) for water electrolysis. Here, we find that anisotropic thermal expansion can induce compressive strains of the IrO_6 octahedron in Sr_2IrO_4 catalyst, thus downshifting its d-band center. Different from the previous strategies to create constant strains in the crystals, the thermal-induced compressive strains can be real-time tuned by varying temperature. As a result of the thermal strain accelerating OER kinetics, the Sr_2IrO_4 exhibits the nonlinear $\ln J_0 - T^{-1}$ (J_0 , exchange current density; T , absolute temperature) Arrhenius relationship, resulting from the thermally induced low-barrier electron transfer. Our results verify that the thermal field can be utilized to manipulate the electronic states of materials via thermal strain effect, significantly accelerating the OER kinetics beyond the traditional thermal diffusion effects.

Conclusions: In summary, we successfully demonstrate that the d-band center of Sr_2IrO_4 can be easily tuned by heating to trigger compressive strains in its IrO_6 units, thus optimizing the OER kinetics by adjusting the binding strength between OER intermediates and Ir active species. Completely different from the previous routes to produce the constant strains in crystals, the thermally induced strains are easily tuned by varying temperature. Heating to produce the compressive strains in crystals will effectively regulate the electronic energetics at electrode-electrolyte interface (Fig. 5), for example, downshifting the d-band center to accelerate the OER kinetics and breaking the linear Arrhenius relationship to achieve higher energy conversion efficiency. Our findings indicated that heating is an effective route to tune the electronic states of materials, thus significantly accelerating OER kinetics, beyond the traditional thermal diffusion effect.

A) Understanding that mechanical strain has the potential of modifying electronic states in materials, and that temperature in materials exhibiting anisotropic thermal expansion can induce internal mechanical strain, using thermal energy to optimize the catalytic performance of materials having this property does not seem a particularly original proposal. In fact, knowing the potential relationships between mechanical strain and electronic state modulation, and between electronic states of catalysts and catalytic performance, even improving performance beyond simply favoring thermal diffusion cannot be considered as a surprise.

Reply: In our study, the main goal is to find a new method to generate variable lattice strain in catalysts. That is, here, the new idea for the strain generated by thermal stimulation was proposed based on the known relationships between mechanical strain and catalytic activity. As is well known, modifying electronic states in materials is crucial to optimize the catalytic performance. Indeed, as pointed out by the Reviewer, lots of methods to generate mechanical strain, such as lattice mismatch, doping heteroatoms, morphology controlling, and introducing defects, were developed to modify the electronic states, thus tuning the OER performances. In these cases, the electronic states of materials are determined by the compositions, microstructures and synthesis conditions of materials, without consideration of the thermal stimulation. The known relationships, between mechanical strain and electronic states, and between electronic states of catalysts and catalytic performance, guide us to design a novel method to induce internal mechanical strain, which could tune the strain more conveniently and more flexibly.

In addition, our results verify that the thermal field can be utilized to tune the electronic states of materials via thermal strain effect. This fact is beneficial for strengthening the knowledge of effects of heat field on catalytic reaction. In fact, in previous reports, the understanding of thermal effect in chemical reaction usually focuses on Arrhenius relationship, where heating accelerates the kinetics with a constant activation energy. That is, it is assumed that the electronic states of catalysts are independent of temperature. However, if the catalysts have the thermophysical effect, the electronic states will vary with temperature, thus changing the catalytic performances. Accordingly, our results are a visible leap to show that the thermal strains accelerated OER kinetics beyond the thermally induced mass transfer.

B) On the one hand, given the enormous complexity and subtlety of the chain of effects involved, it does not seem that the proposed optimization approach can be easily designed from mechanisms. Moreover, in general, detailed knowledge of thermal induced effects and mechanisms on materials and reactions can hardly be transferred to other materials and reactions, which would be particularly true for complex reactions involving multidentate intermediates.

Reply: Indeed, as described by Reviewer, the most heterogeneous catalytic reactions are complex processes that involved the enormous factors. So far, lots of famous theories, such as d-band center theory, charge-transfer energy (covalency), and e_g occupancy, were developed to guide the material design and understand their catalytic mechanism (*J. Phys. Chem. C* 2016, 120, 78-86). As pointed out by the Reviewer, these models are not a general method to all materials because the main control factors of reaction kinetics are catalyst-dependent. However, a specific theory is able to effectively explain the catalytic mechanism and guides the design of a certain material. In our work, the thermal field modulating electronic states is proposed based on the d-band center theory, a well-known mechanism to guide the design of transition metal compounds. Based on this model, the design concept of materials is clear: tensile strain upshifts d-band center, whereas compressive lattice strain downshifts the d-band center. Therefore, design of materials with thermal strain effects would follow the same rules in controlling d-band center: thermal strain upshifts or downshifts the d band center. This means that the d-band center theory is still effective to guide the design of materials with

thermal strain effects.

Indeed, although it is a big challenge to transfer a specific catalytic mechanism to other materials and reactions, the more materials are hopeful to be found based on the various mechanism to generate thermal strain effects. For example, materials having negative thermal expansion effect (NTE, a typical anisotropic thermal expansion) are a huge material family, which exhibit the thermal strain effects (*Front. Phys.* **2021**, *16*, 53302). In particular, the framework structure NTE materials contain divers oxides, including the family with AM_2O_7 , $A_2M_3O_{12}$ and $A_2M_2O_7$ formula, as well as fluorides represented by ScF_3 , and molecular framework materials, which have organic chain linkages, such as Prussian blue analogues (PBAs), and metal-organic frameworks (MOFs) (*Coord. Chem. Rev.* **2021**, *449*, 214204). It is found that the NTE in these framework structure materials is closely related to low frequency phonon, so they are called phonon-driven NTE. For these materials, strain effects can be introduced by heating to optimize *d*-band center position.

C) On the other hand, thermally induced effects, even on complex catalysts and reactions, can be easily exploited, in practice, even without knowing them, by simply exploring experimentally the degree of freedom operation temperature of the catalytic configuration. Thus, perhaps the most valuable contribution of the manuscript would be that it provides evidence that the performance of a catalytic process involving a catalyst exhibiting anisotropic thermal expansion can be significantly increased, even beyond simply favoring thermal diffusion, by operating at a higher temperature. Therefore, the main conclusion seems to be that “it is worthwhile to explore the degree of freedom operation temperature of a catalytic configuration”, which seems obvious. This said, the reported results and mechanisms are interesting.

Reply: Thank you for your recognition of the scientific contribution of our study. We totally agree with the Reviewer’s opinion that the thermally induced effects can be exploited by introducing heat source into the catalytic configuration, even without knowing them. The main contribution of this study is to disclose a thermophysical effect, the thermal-induced strain, can accelerate OER kinetics beyond the thermally induced mass transfer. Given that catalytic activity of materials depends closely on their electronic states, it would seem to be a general truth that not all thermophysical effects are beneficial for accelerating the rate of chemical reactions. In our case, we just found that the thermal-induced compressive strain is a positive thermal contribution to OER performance. Various thermophysical effects, such as thermal expansion, thermal diffusion, and thermal phase transition, can tune the electronic states of materials. Although we have verified that thermally induced magnetic transition (*Adv. Funct. Mater.* **2022**, *32*, 2111234) and thermally manipulated spin changes (*Nano Lett.* **2022**, *22*, 9131) are able to improve OER performances, it is possible that there is a given themophysical effect that does not exhibit a positive contribution to OER activity. For example, a thermal-induced structural phase transition of martensite to austenite may exhibit a negative contribution to OER activity because the austenite generally exhibits the low ability in electron transport. Therefore, we agree with the reviewer’s standpoint that it is worthwhile to explore the effects of operation temperature on a catalytic configuration, thus disclosing the OER mechanism with a given thermophysical effect beyond the thermal diffusion.

According to the Reviewer’s suggestions, we have revised the Introduction and the Conclusions as follows:

Abstract: Compressive strain, downshifting the *d*-band center of transition metal oxides, is an efficient way to accelerate the sluggish kinetics of oxygen evolution reaction (OER) for water electrolysis. Here, we find that anisotropic thermal expansion can induce compressive strains of

the IrO_6 octahedron in Sr_2IrO_4 catalyst, thus downshifting its d -band center. Different from the previous strategies to create constant strains in the crystals, the thermal-induced compressive strains can be real-time tuned by varying temperature. As a result of the thermal strain accelerating OER kinetics, the Sr_2IrO_4 exhibits the nonlinear $\ln J_0 - T^{-1}$ (J_0 , exchange current density; T , absolute temperature) Arrhenius relationship, resulting from the thermally induced low-barrier electron transfer. Our results verify that, beyond the thermal diffusion, the thermal field can be used to manipulate the electronic states of materials via thermal strain effect, leading to a significant acceleration of the OER kinetics.

Conclusions: In summary, we successfully demonstrate that the d -band center of Sr_2IrO_4 can be easily tuned by heating to trigger compressive strains in its IrO_6 units, thus optimizing the OER kinetics by adjusting the binding strength between OER intermediates and Ir active species. Completely different from the previous routes to produce the constant strains in crystals, the thermally induced strains are easily tuned by varying temperature. Heating to produce the compressive strains in crystals will effectively regulate the electronic energetics at electrode-electrolyte interface (Fig. 5), for example, downshifting the d -band center to accelerate the OER kinetics and breaking the linear Arrhenius relationship to achieve higher energy conversion efficiency. Our findings indicated that heating is an effective route to tune the electronic states of materials, thus significantly accelerating OER kinetics, beyond the traditional thermal diffusion effect.

D) By means of XRD, SEM, HRTEM, and HAADF-TEM, it is concluded that single crystal nanoparticles from 200 to 300 nm of Sr_2IrO_4 were obtained. However, the shape of the nanocrystals is not identified. Therefore, the dominant crystalline plane exposed by the catalyst cannot be established, reason why the experimental and computational results cannot be reliably correlated. Citing Chem. Eng. J. 423 (2021) 130185 (Ref 41), OER on the Sr_2IrO_4 (001) surface is computationally investigated. However, the (001) surface focus in such a reference is not clearly supported. The shape of the nanoparticles in Ref. 41 is not identified. Moreover, from the comparison between Fig. S2c of Ref. 41 and Fig. S1a of the revised manuscript, similar shapes cannot be concluded.

Reply: Thanks for your valuable comments. Indeed, the Sr_2IrO_4 particles are not a perfect single crystal, but a stunted crystal (**Supplementary Fig.1b**). The crystal growth is checked by careful SEM and TEM observations on a relatively perfect single-crystal particle. As shown in **Supplementary Fig.1c**, the Sr_2IrO_4 particle was formed by stacking a given crystallographic facet along a specific crystallographic direction. The selected area electron diffraction (SAED) was performed on a cuboid-like particle and exhibited clear diffraction pattern of the single crystal, illustrating that the particle is a well-grown single crystal (**Supplementary Fig. 1e**). The high-resolution TEM image confirmed that two intersectional lattice fringes with an angle of 90° were detected. Both the longitude and the latitude lattice spacing are 0.274 nm and are assigned to the (110) and (1 $\bar{1}$ 0) facets of Sr_2IrO_4 , respectively. Therefore, the incoming electron beam was along the [001] axis. According to the tetragonal crystallographic structure of Sr_2IrO_4 , we can conclude that the exposed facet of Sr_2IrO_4 is {001} facet and which tends to stack along the [001] crystallographic direction. The SAED is well coincident with the HRTEM results and further confirmed that it is a thermodynamically dominated process for the Sr_2IrO_4 particle with

exposure of the {001} facet stacking along [001] axis.

The crystal growth of stunted Sr_2IrO_4 particles would follow the same growth mechanism to the relatively perfect ones. Therefore, we believe that the stunted Sr_2IrO_4 particles are a result of stacking (001) facets. Indeed, although the particles are not a perfect crystal with regular shape, we can trace the profile of the single crystal-like particles by careful observations on SEM image. As shown in **Supplementary Fig.1b**, the stunted particles exhibit the plate-like shape.

Supplementary Fig. 1 | a, The XRD patterns for Sr_2IrO_4 before/after OER and carbon paper substrate. The XRD peak at 26.38° and 54.54° are assigned to the (002) and (004) crystal plane of graphic carbon, respectively. **b**, SEM image for Sr_2IrO_4 after electrochemical activation by i-t test at 1.45 V at 90°C for 1 h in 1 M KOH. The dotted lines trace the profile of the stunted Sr_2IrO_4 particles. **c**, SEM image for Sr_2IrO_4 particle formed by stacking a given crystallographic facet along a specific crystallographic direction. **d**, HRTEM lattice image for a stunted Sr_2IrO_4 single crystal with irregular shape. **e**, TEM image, HRTEM lattice image, and SAED pattern for a relatively perfect Sr_2IrO_4 single crystal to show the possible growth mechanism and exposed facet. The HRTEM lattice image and SAED pattern were collected on the yellow box area of this crystal.

E) The assumed adsorbates scenery (species, coverages, sites ...), under the different operation conditions (pH, potential, temperature...), should be discussed. Solvation effects should be included. The approach of using experimental lattice parameters to accurately simulate thermally induced strain under a specific model chemistry should be validated.

Reply: Thanks for your valuable suggestions. The Rietveld refinements of XRD patterns were carried out to get the temperature-dependent lattice parameters of the Sr₂IrO₄. The XRD refined results indicated that lattice parameters of Sr₂IrO₄ under 25 °C are $a = b = 5.50 \text{ \AA}$, $c = 25.958 \text{ \AA}$, and $a = b = 5.51 \text{ \AA}$, $c = 25.723 \text{ \AA}$ for Sr₂IrO₄ under 90 °C. The thermal train in IrO₆ was determined by fitting EXAFS results. Under 25 °C, the Ir-O_{ab-plane} and Ir-O_{c-axis} bond lengths of IrO₆ unit are 1.989 Å and 1.993 Å, respectively. Under 90 °C, the Ir-O_{ab-plane} and Ir-O_{c-axis} bond lengths are 1.938 Å and 1.937 Å, respectively. According to the XRD refinement and EXAFS fitting result, the (001) facet slab model was accordingly created to accurately check the effects of thermally induced strain on OER.

As suggested by the Reviewer, here the OER intermediate surface coverage (θ) was calculated from the capacitance of the intermediate adsorption (C_{ads}). Since the C_{ads} was normalized by ECSA, the result shows the intrinsic ability of Sr₂IrO₄ for adsorbing the intermediates under different temperatures. Then we recalculated the standard Gibbs free energy for *OH, *O, and *OOH adsorbed on Sr₂IrO₄ by considering the active sites, pH, potential, temperature, and solvation effects.

Surface coverage of OER intermediates at different temperatures was calculated by fitting the EIS plots recorded at different potentials. As shown in Supplementary Fig. 13a, the Randle's equivalent circuit is composed of the series resistance in the circuit (R_s), the capacitance from double layer (C_{dl}) and intermediates adsorption (C_{ads}), the charge-transfer resistance (R_1 and R_2 , $R_{\text{ct}} = R_1 + R_2$) (J. Am. Chem. Soc. 2016, 138, 31, 9978–9985). In the Nyquist plot, the C_{ads} and C_{dl} were obtained by fitting the plots from low-frequency regions (Supplementary Fig. 13b,c).

Harrington and Conway gave the definition of potential-dependent $C_{\text{ads}}(E)$ as: (J. Electroanal. Chem. Interfacial Electrochem. 1987, 221, 1.)

$$C_{\text{ads}}(E) = q[d\theta(E)/dE] \quad (1)$$

where $C_{\text{ads}}(E)$ is the capacitance of intermediates adsorption at applied potential E , q is the charge density for a monolayer coverage, which can be assumed as constant because the adsorption sites of Sr₂IrO₄ are stable during OER, $\theta(E)$ is the surface coverage of OER intermediates at applied potential E .

By integrating Eq. 2, we can get the $\theta(E)$:

$$\theta(E) = [\int_{E_0}^E C_{\text{ads}}(E) dE]/\sigma \quad (2)$$

where the E_0 refers to the potential under which no obvious adsorption occurs, the E is the potential at arbitrary coverage, σ is the number of adsorbates per surface area for monolayer coverage. In our case, the E_0 is 1.45 V vs. RHE under 20 °C and 1.3 V vs. RHE under 90 °C, respectively.

As shown in Supplementary Fig. 13d, the Sr₂IrO₄ exhibited the similar coverage of OER intermediates at the same current density, nearly independent of the temperatures, suggesting that the coverage of OER intermediates is independent of the temperatures.

Supplementary Fig. 13 | Electrochemical characterization of oxygenated intermediates adsorption. **a**, Electrical equivalent circuit for adsorption. R_s refers to the series resistance in the circuit. C_{dl} and C_{ads} are the capacitance from double layer and intermediates adsorption, respectively, whereas the physical significance of R_1 and R_2 are still under debate but with a general viewpoint that overall charge transfer resistance $R_{ct} = R_1 + R_2$. (*J. Electroanal. Chem.* 2009, 631, 62) **b,c** Trend of C_{ads} decoupled from EIS results (Fig. 3d,e) for Sr_2IrO_4 measured under 20 (b) and 90 °C (c). **d**, Variation in the surface coverage of intermediates.

We recalculated the standard Gibbs free energy for *OH , *O , and *OOH adsorbed on Sr_2IrO_4 by considering the active sites, pH, potential, temperature, and solvation effects as follows:

Effects of temperature. As described in reference of *ACS Catal.* 2021, 11, 11305-11319, Rajan et al. have developed a method to describe standard Gibbs free energies of OER intermediates at arbitrary temperatures below the water boiling point. We adopted the same method to generalize the standard Gibbs free energies of molecular (H_2 , O_2 , and H_2O), ionic ($H^+ + e^-$) species, and the OER intermediates (*OH , *O , *OOH) at any temperature T lower than the boiling point of water.

In our case, the OER thermodynamics at elevated temperature was calculated based on the computational hydrogen electrode (CHE) framework advanced by Nørskov and colleagues (*J. Phys. Chem. C* 2008, 112, 9872-9879). Given that the CHE is based on the standard hydrogen electrode (SHE), whose potential at all temperatures is zero by convention, the standard potential of the hydrogen evolution reaction is zero in this framework, so that:

$$G_{(H^++e^-)}^0(T) = 0.5G_{H_2(g)}^0(T) \quad (3)$$

Where $G_{(H^++e^-)}^0(T)$ and $G_{H_2(g)}^0(T)$ are the standard Gibbs free energies of the proton-electron couple and gaseous hydrogen at temperature T .

The standard Gibbs free energies of molecular H_2 and H_2O at arbitrary temperatures below the water boiling point were calculated by the Eq. 4:

$$G_i^{\circ}(T) = E_i^{\text{DFT}} + ZPE_i^{\text{DFT}} + [H_i^{\circ}(T) - H_i^{\circ}(0 \text{ K}) - TS_i^{\circ}(T)] \quad (4)$$

where i denotes either $\text{H}_2(\text{g})$ or $\text{H}_2\text{O}(\text{g})$, E_i^{DFT} the species' DFT total energy at zero Kelvin, ZPE_i^{DFT} its zero-point vibrational energy from DFT. $H_i^{\circ}(T)$ and $S_i^{\circ}(T)$ its standard enthalpy and entropy at T , respectively, and can be calculated through Shomate equations from the National Institute of Standards and Technology (NIST) database.

The standard Gibbs free energy of liquid water at temperature T was calculated by

$$G^{\circ}_{\text{H}_2\text{O}(\text{l})}(T) = G^{\circ}_{\text{H}_2\text{O}(\text{g})}(T) + \Delta G^{\circ}_{\text{c,H}_2\text{O}(\text{g})}(T) \quad (5)$$

where $\Delta G^{\circ}_{\text{c,H}_2\text{O}(\text{g})}(T)$ is the standard Gibbs free energy of the condensation of water vapor.

The standard Gibbs free energy of molecular O_2 was calculated by the following equation:

$$G^{\circ}_{\text{O}_2(\text{g})}(T) = \Delta G^{\circ}_{\text{r,H}_2\text{O split}}(T) + [2G^{\circ}_{\text{H}_2\text{O}(\text{l})}(T) - 2G^{\circ}_{\text{H}_2(\text{g})}(T)] \quad (6)$$

where $\Delta G^{\circ}_{\text{r,H}_2\text{O split}}(T)$ is the standard Gibbs free energy of the reaction for water splitting.

The free energies of adsorbed OER intermediates ($^*\text{OH}$, $^*\text{O}$, $^*\text{OOH}$) as a function of temperature using the harmonic oscillator model for solids as:

$$G_i^{\circ}(T) = E_i^{\text{DFT}} + E_{\text{vib},i} - TS_{\text{vib},i} \quad (7)$$

Where G_i° is the standard Gibbs free energy of the i th OER intermediates, E_i^{DFT} is the total energy of the i th OER intermediates, $E_{\text{vib},i}$ is vibrational internal energy, $TS_{\text{vib},i}$ is vibrational entropy.

Effects of pH. In our case, the pH changes of the electrolyte are originated from the thermally adjusted ability of water to ionise. When heating, pH of the electrolyte decreases from 13.7 at room temperature to 11.9 at 90 °C with no change in KOH concentration to be 1 M. Therefore, in our case, the pH decrease mainly originates from the water dissociation of $\text{H}_2\text{O} = \text{H}^+ + \text{OH}^-$ driven by absorbing extra heat. The water ionisation constant increases with increasing the temperatures, thus decreasing the pH due to that the pH is defined as the negative log of the hydrogen ion concentration expressed in mol/L. We add the $\text{pH} \times k_{\text{B}}T \ln 10$ into the Gibbs free energy expression of elementary reaction steps to reflect the effects of pH.

Solvation effects. We utilize an implicit solvation model to describe the solvation effects under thermal stimulation through solving the linearized Poisson-Boltzmann equation by VASP_{sol} (*J. Chem. Phys.* **2014**, *140*, 084106; *J. Chem. Phys.* **2019**, *151*, 234101). The relative permittivity of water was set as 78.4 under 25 °C and 58.13 under 90 °C (*J. Phys. Chem. Ref. Data* **1997**, *26*(4), 1125-1166). The Debye length of the 1.0 M electrolyte was set to 3.04 Å under 25 °C and 3.36 Å under 90 °C (*J. Chem. Phys.* **2014**, *140*, 084106; *J. Chem. Phys.* **2019**, *151*, 234101).

Considering the pH, potential, temperature, and solvation effects, the standard Gibbs free energy for elementary reaction steps at arbitrary temperatures below the water boiling point were calculated by Eqs. 8-10:

$$\Delta G^{\circ}_{\text{OH}}(T) = G^{\circ}_{\text{OH}}(T) + 0.5 \times G^{\circ}_{\text{H}_2}(T) - G^{\circ}_{\text{H}_2\text{O}}(T) - G^{\circ}_{\text{O}}(T) - eU + \text{pH} \times k_{\text{B}}T \ln 10 \quad (8)$$

$$\Delta G^{\circ}_{\text{O}}(T) = G^{\circ}_{\text{O}}(T) + G^{\circ}_{\text{H}_2}(T) - G^{\circ}_{\text{H}_2\text{O}}(T) - G^{\circ}_{\text{O}}(T) - eU + \text{pH} \times k_{\text{B}}T \ln 10 \quad (9)$$

$$\Delta G^{*OOH}(T) = G^{*OOH}(T) + 1.5 \times G_{H_2}(T) - 2 \times G_{H_2O}(T) - G^*(T) - eU + pH \times k_B T \ln 10 \quad (10)$$

where, $-eU$ stands for the free energy changes for one electron transfer, U is electrode potential respect to the standard hydrogen electrode. pH effects on free energy can be defined as $pH \times k_B T \ln 10$, where k_B is Boltzmann constant (*J. Catal.* **2018**, *358*, 100–107).

Considering the pH, potential, temperature, and solvation effects (**Supplementary Fig. 14**), the Gibbs free energy profile on the Ir sites of the Sr_2IrO_4 (001) surface still follows the same trend as the energy calculations with sole consideration of the strain effect (**Fig. 4a,b**). The rate-determining step is still *OH deprotonation step with a smaller free energy difference between ΔG^{*OH} and ΔG^{*O} to be 1.22 eV under 25 °C and 1.18 eV under 90 °C.

Active sites on Sr_2IrO_4 with consideration of operation conditions. We calculated the Gibbs free energy diagrams for OER intermediates (*OH , *O , *OOH) adsorbed on the Sr or Ir sites on the Sr_2IrO_4 surface. The results were shown in **Supplementary Figs. 14 and 15** and indicated that the Ir sites are more likely to be the active OER catalytic center due to the lower energy requirement in rate-determining step. The RDS on Sr sites are estimated to be 1.86 and 1.69 eV under 25 and 90 °C, respectively, which both are much larger than on Ir sites.

To visualize the effects of internal strains on OER, we show the Gibbs free energy diagrams for OER intermediates (*OH , *O , *OOH) adsorbed on the Ir sites of the Sr_2IrO_4 with sole consideration of thermal strain effect in the main text. And we add the Gibbs free energy diagrams for OER intermediates (*OH , *O , *OOH) adsorbed on Ir sites of Sr_2IrO_4 with consideration of operation conditions into the Supplementary materials.

Supplementary Fig. 14 | a-d, Structures of the Ir sites of the Sr_2IrO_4 (001) surface (a) and *OH (b), *O (c), *OOH (d) adsorptions on Ir sites. e,f, Gibbs free energy diagrams for OER intermediates (*OH , *O , *OOH) adsorbed on the Ir sites of the Sr_2IrO_4 (001) surface with thermal

strains under 25 °C (e) and 90 °C (f). Solvation, temperature, and pH effects were included in the calculations.

Supplementary Fig. 15 | a-d, Structures of the Sr sites of the Sr₂IrO₄ (001) surface (a) and *OH (b), *O (c), *OOH (d) adsorptions on Sr sites. e,f, Gibbs free energy diagrams for OER intermediates (*OH, *O, *OOH) adsorbed on the Sr sites of the Sr₂IrO₄ (001) surface with thermal strains under 25 °C (e) and 90 °C (f). Solvation, temperature, and pH effects were included in the calculations.

F) The applied numerical treatment (DFT + U under the PBE functional) should be validated for the application (energy profile and DOS of the OER on bimetallic earth-alkaline/transition metal oxides). Better figures displaying the investigated configurations should be provided.

Reply: Thanks for your valuable reminding. The U value is crucial for the accuracy of DFT calculations including DOS and energy profile of the OER on Sr₂IrO₄. It has been well studied that the U value of ~2 eV is now recognized as a validated value both based on experimental (*Phys. Rev. Lett.* 2008, 110, 226402) and theoretical results (*Phys. Rev. Lett.* 2012, 108, 086403; *JPS Conf. Proc.* 2014, 3, 013023; *Phys. Rev. Mater.* 2018, 2, 075003; *Phys. Rev. Mater.* 2019, 3, 063607). With a U value of 2.0 eV for Sr₂IrO₄, the calculated electronic structures are consistent with the joint density of states described by the experiential result of optical conductivity spectrum $\sigma(\omega)$ (*Phys. Rev. Lett.* 2008, 101, 226402). Using *ab initio* by the constrained random phase approximation (cRPA), U value of Sr₂IrO₄ also can be calculated as ~2 eV (*Phys. Rev. Lett.* 2012, 108, 086403). Therefore, the U value of 2 eV was used in our calculations to describe the Coulomb interaction.

We updated the Fig.5 to display our investigated configurations.

Fig. 5 | Energetics at catalyst-electrolyte interface with thermal strain effect. Effects of heating on the OER. The improved OER performances originate from that heating accelerates mass transfer and the thermal strains in IrO_6 units downshift the d -band center. The strong spin-orbital coupling in the low-spin Ir^{4+} ($5d^5$, $t_{2g}^5e_g^0$) $5d$ orbital in Sr_2IrO_4 splits the five electrons in the t_{2g} band into four electrons occupied as the $J_{\text{eff}} = 3/2$ sub-band and to one electron occupied as the $J_{\text{eff}} = 1/2$ sub-band. The thermal strains downshifting d -band center is real-time sensitive to the temperature. The thermal diffusion accelerating OER follows a linear $\ln k - T^{-1}$ Arrhenius relationship and the thermal strain accelerating OER follows the nonlinear $\ln k - T^{-1}$ Arrhenius relationship due to that downshifting d -band center will adjust the Ir-O bonding (σ) and antibonding (σ^*) states to meet the Sabatier principle.

G) The second sentence in the abstract section (lines 14-17) is too long, making it difficult to understand.

Reply: Abstract was revised as: *Compressive strain, downshifting the d -band center of transition metal oxides, is an efficient way to accelerate the sluggish kinetics of oxygen evolution reaction (OER) for water electrolysis. Here, we find that anisotropic thermal expansion can induce compressive strains of the IrO_6 octahedron in Sr_2IrO_4 catalyst, thus downshifting its d -band center. Different from the previous strategies to create constant strains in the crystals, the thermal-induced compressive strains can be real-time tuned by varying temperature. As a result of the thermal strain accelerating OER kinetics, the Sr_2IrO_4 exhibits the nonlinear $\ln J_0 - T^{-1}$ (J_0 , exchange current density; T , absolute temperature) Arrhenius relationship, resulting from the thermally induced low-barrier electron transfer. Our results verify that the thermal field can be utilized to*

manipulate the electronic states of materials via thermal strain effect, significantly accelerating the OER kinetics beyond the traditional thermal diffusion effects.

Assuming that the strain is sensitive to the temperature, which can vary continuously, the constructions “...thermal-triggered continuous...” in line 19 (and 49), or “...sensitive to the temperature variation...” in line 48 (also 155 and 269), are confusing to me.

Reply: Corresponding sentences was revised as:

1. As a result of the thermal strain accelerating OER kinetics, the Sr_2IrO_4 exhibits the nonlinear $\ln J_0 - T^{-1}$ (J_0 , exchange current density; T , absolute temperature) Arrhenius relationship, resulting from the thermally induced low-barrier electron transfer.
2. As a result, the binding strength between OER intermediates and the Ir active species was optimized to meet the low-barrier catalytic reaction described by Sabatier principle,²⁷ thus exhibiting the nonlinear Arrhenius relationship.
3. Under thermal stimulation, the compressive strains of the IrO_6 octahedra in Sr_2IrO_4 catalyst can be created, consequently downshifting the d-band center.
4. During heating and cooling electrolyte, the completely reversible LSV polarization curves witnessed that the OER performances of Sr_2IrO_4 are highly sensitive to the temperature.
5. Completely different from the previous routes to produce the constant strains in crystals, the thermally induced strains are easily tuned by varying temperature.

“identifying” in line 20 could be not the best choice. It should be explained the sense in which strain engineering is “efficient” in optimizing bands (lines 28-30).

Reply: The effectiveness for optimizing bands by strain engineering was clearly described as: Strain engineering is especially effective in optimizing the d-band center of transition metal-based catalysts via compressive or tensile lattice strain.¹⁵ Generally, tensile lattice strain reduces the overlap of the wavefunctions and thus gives rise to the narrowing of the metal d band and an upshift of d-band center.⁶ In contrast, compressive lattice strain has the opposite effect, causing a broadening of the d band and a downshift of the d-band center. For OER, the d-band center is closely related to the interaction energy between adsorbate states and the metal d states, thus determining the OER kinetics.

The content from line 44 to 47 should be presented in a more concise form. I think that the last sentence in the last paragraph of the introductory section of the manuscript (lines 51 to 53) is a bit triumphalist.

Reply: This sentence was revised as: Our findings prove that producing thermal strains by anisotropic thermal expansion is an effective route to tune the electronic states of materials, significantly increasing OER performances beyond the traditional thermal diffusion effects.

Significant thermally induced improvements could be limited to only a few reactions on some catalysts from among those exhibiting anisotropic thermal expansion. Sentences from line 271 onwards, up to the end of the conclusions section, seem too vague. Labeling in Fig. 4 caption is not correct.

Reply: Thanks for your suggestions. The conclusion was revised as: In summary, we successfully demonstrate that the d-band center of Sr_2IrO_4 can be easily tuned by heating to

trigger compressive strains in its IrO_6 units, thus optimizing the OER kinetics by adjusting the binding strength between OER intermediates and Ir active species. Completely different from the previous routes to produce the constant strains in crystals, the thermally induced strains are easily tuned by varying temperature. Heating to produce the compressive strains in crystals will effectively regulate the electronic energetics at electrode-electrolyte interface (Fig. 5), for example, downshifting the d-band center to accelerate the OER kinetics and breaking the linear Arrhenius relationship to achieve higher energy conversion efficiency. Our findings indicated that heating is an effective route to tune the electronic states of materials, thus significantly accelerating OER kinetics, beyond the traditional thermal diffusion effect.

The labeling in Fig. 4 caption was revised.

In addition, we have carefully revised all the text to avoid the possible confusion and highlighted the revisions by yellow color.

REVIEWER COMMENTS

Reviewer #1 (Remarks to the Author):

After the revision, this work could be accepted.

Reviewer #2 (Remarks to the Author):

I recommend publishing the current version.

Reviewer #3 (Remarks to the Author):

Although the authors have addressed all my comments, my doubts about the relevance of the proposal, and my concerns about the modeling, still persist.

In their "Responses to the reviewers' comments", when the authors try to reproduce the first paragraph of my comments, they write "...I have no doubts about the relevance of the proposal.", where "...I have doubts about the relevance of the proposal." was in fact written. Unfortunately, those doubts still persist. I will not insist on them here.

Even accepting that single crystal nanoparticles were produced, I think that, from the performed characterization effort, that {001} is the dominant exposed facet cannot be concluded. Continuous lattice fringes along the entire length of the particle confirm crystallinity. Characteristic distance in the lattice fringes allows identifying the orientation of the observed crystal with respect to the observer. From this, the plausibility of certain facets being exposed can be argued. However, since the dominant shape of the nanoparticles has not been identified, I think that it is not possible to conclude which is the dominant exposed facet. Thus, I think that the obtained experimental and computational results cannot be reliably correlated. In the current version of the manuscript (line 71), that there exist "...a big possibility to expose {001} facet." is written. However, a big possibility of having exposed {001} facets does not implicate that this is the dominant exposed facet. In the current version of the manuscript (lines 67 and 68), the nanoparticles are described as having "...irregular shape..." being "...stunted single crystal". In the presence of nanoparticles having irregular shape, calculations on a single crystalline plane and

experimental results cannot be reliably correlated. I do not understand why Reference 41 is maintained in the current version of the manuscript, to support the {001} facet focus, despite my comments, which have not been responded to by the authors.

Overall, the presentation of the characterization of the synthesized material seems confusing to me. In the original version of the manuscript (lines 58 to 61), that “The high-resolution transmission electron microscope (HRTEM) lattice image confirmed that the Sr₂IrO₄ is highly crystalline and exhibits the continuous lattice fringes of 0.32 nm for (004) facet extending through the whole particle, evidencing that the particle is single crystal, as further confirmed by the fast Fourier transform image (Fig. 1a).” was written. Instead, in the current version of the manuscript (lines 65 to 68), the authors write that “The high-resolution transmission electron microscope (HRTEM) lattice image confirmed that the stunted Sr₂IrO₄ is highly crystalline and exhibits the continuous lattice fringes of 0.32 nm for (004) facet extending through the whole particle, evidencing that the particle with irregular shape is a stunted single crystal (Fig. 1a).” The fast Fourier transform image, that in the previous version of the manuscript was presented as an essential validation test, has been removed, and essential shape descriptions are now based on the relatively unspecific adjective “stunted”. In only four lines (66 to 70), “stunted” is used, up to four times, to qualify oxide (line 66), crystallinity (line 68), and particle (line 70). Whatever “stunted” means for the authors, I sincerely think that it cannot be appreciated from Fig. 1a. To support the {001} facet focus, the authors (starting at line 68) attempt “To disclose the growth mechanism of the stunted Sr₂IrO₄ single crystals...”, with explanations that, in my opinion, are somewhat speculative, rather irrelevant, and fail to achieve their purpose. The figure referred to in line 70 is not correct. The characterization of the synthesized material should be much more carefully presented.

Regarding the assumed adsorbates scenery (species, coverages, sites ...), under the different operation conditions (pH, potential, temperature...), I do not understand how many molecules of each intermediate adsorbate, including water, are involved in each computational model and why. I do not understand a.u. as a unit of surface coverage in Supplementary Fig. 13d. I do not understand why ticks, allowing values reading, are not shown in Supplementary Fig. 13d. The apparent presence of a different number of adsorbed molecules in the model series should be clarified (Fig. 4a and b, Supplementary Fig. 14). The geometric boundaries of the unit cells should be delineated in the figures. The presence of visually unbounded (from the surface) molecules in the models is confusing to me (Fig. 4b or Supplementary Fig. 14c). Better figures, showing the investigated configurations, should be provided. I do not understand why pre-adsorbed molecules are not included in the models displayed in Supplementary Fig. 15 a-d. pH effects and/or potential could give rise to charged electrode, which should be considered, making it difficult the energies alignment corresponding to models having different composition.

To validate the applied numerical treatment (DFT[GGA] + U under the PBE functional), the authors focus their effort on justifying the value adopted for the U parameter, as if an accurate value for U were in itself a sufficient (and not only necessary) condition for accurate calculations. Supporting the adopted value, Phys. Rev. Lett. 2008, 101, 226402 (DFT[LDA] + U under unspecified functional); Phys. Rev. Lett. 2012, 108, 086403 (DFT[GGA] + SOC + U under the PBE functional, in a multi-energy-scale scheme); Phys.

Rev. Mater. 2018, 2, 075003 (BSE on full relativistic DFT + SOC + U under unspecified exchange-correlation treatment, not accurately reproducing GW + BSE); or Phys. Rev. Mater. 2019, 3, 063607 (DFT[GGA] + SOC + U under the PBE functional), are cited. Although DFT calculations are often referred to as “ab initio” or “first principles” calculations, parameters are actually embedded in the approximation to the essentially unknown exchange-correlation functional, for whose formulation there exist several approaches: LDA, GGA, hybrid, Meta... (AIP Conference Proceedings 577, 1 (2001); <https://doi.org/10.1063/1.1390175>). An inadequate numerical treatment of the self-interaction, correlation, spin-orbit coupling, relativistic effects, or spin multiplicity can compromise the validity of calculations on metal oxides involving 5d electrons. The variety of numerical treatments adopted by the authors of the cited references, which include components not considered by the authors, exemplify the suggested complexity. Furthermore, not only properties of the materials have to be captured (as it is the case of the cited references), but also interaction with adsorbates. For instance, in a recent OER research on transition metal oxides (Nature Energy 2022, 7, 765), the hybrid exchange-correlation functional PBE0 is considered. The accuracy of each numerical treatment depends on the system and property investigated: structure, energy, density, spectroscopy... (Science 355 (2017) 49). The accuracy of the adopted functional in this case should be established for DOSs and free energy profiles involved in the OER on bimetallic earth-alkaline/transition metal oxides.

It appears that the authors understand that, to accurately model the strain under different temperature conditions, experimentally derived lattice parameters must be used. However, on the one hand, the free energy profiles of reactions at a given temperature are usually derived, from calculations performed using the lattice parameters corresponding to the ground state, correcting for zero-point energies and temperature-dependent entropies. On the other hand, since any adopted functional is only an approximation, the experimentally derived lattice parameters rarely match the corresponding ones in model chemistry. Most likely, the experimentally derived lattice parameters give rise to uncontrolled strain in the model chemistry. For this reason, experimentally derived lattice parameters are rarely used to establish computational models, with optimized lattice parameters under model chemistry being preferable. $a = b = 5.50$ at 25 °C versus $a = b = 5.51$ at 90 °C are in the order of the typical differences observed between experimentally derived and computationally optimized lattice parameters. The desirability of using experimentally derived lattice parameters does not seem obvious to me. The thermally induced strain simulation should be validated with care.

I believe that the presentation of the manuscript would still benefit from careful revision. For instance, I suggest “effective” by “efficient” in line 13. From line 15 to 20, “induce/d” is used up to three times with two different “inducing agents”: a material property “anisotropic thermal expansion”, and an energy form “heat”. A more rigorous and concise presentation of the abstract section would be desirable. In line 28, I assume that authors wanted to write “product” instead of “production”. I suggest reconsidering the construction “We are accordingly inspired by the thermal lattice strains...” (line 49). In the caption of Fig. 4, the investigated crystalline plane should be specified. The sentence “Our results imply that the materials with thermophysical effects can be applied to create the efficient water splitting with the simultaneous input of heat and electricity” (lines 283 to 285) seems to me unfortunate. All the materials experiment thermophysical effects. Not all the catalysts exhibiting anisotropic thermal expansion would

necessarily favor the water splitting when heated. The sentence “Heating to produce the compressive strains in crystals will effectively regulate the electronic energetics at electrode-electrolyte interface...” (lines 301 and 302) is not accurate enough. Heating would not produce compressive strain in all crystalline materials. The sentence “Our findings indicated that heating is an effective route to tune the electronic states of materials, thus significantly accelerating OER kinetics, beyond the traditional thermal diffusion effect.” (lines 304 and 305) seems to me unfortunate. Heating as a route to tune the electronic states of materials exhibiting isotropic thermal expansion has not been discussed. Heating would not necessarily accelerate the OER on all catalyst exhibiting anisotropic thermal expansion. A more rigorous and concise presentation of conclusions would be desirable. Given that the atom positions are relaxed, I suggest to remove the experimentally derived bond lengths (lines 365 and 366) from the methods section. In the caption of Supplementary Fig.1, I assume that authors wanted to write “graphitic” instead of “graphic”.

Responses to the Reviewer's comments

Reviewer #3 (Remarks to the Author):

Although the authors have addressed all my comments, my doubts about the relevance of the proposal, and my concerns about the modeling, still persist. In their "Responses to the reviewers' comments", when the authors try to reproduce the first paragraph of my comments, they write "...I have no doubts about the relevance of the proposal.", where "...I have doubts about the relevance of the proposal." was in fact written. Unfortunately, those doubts still persist. I will not insist on them here.

Reply: We appreciate the time and efforts again for your insightful comments. Firstly, we apologize for this mistake with an addition of "no" word during reproducing the first paragraph of Reviewer's comments. This was an accident and we feel so sorry to make this mistake. Here, we response the Reviewer's doubts about the relevance of the proposal. In the manuscript, the aim is to find a new method to generate variable lattice strain in catalysts, thus tuning the OER performances. Here, as described in the Introduction, we find that the anisotropic thermal expansion can induce the thermal strains in lattice, which is timely responsive to the temperature to tune the lattice strain. We carried out the in situ XRD, in situ XAFS, in situ XPS to confirm the generation of thermal strains in IrO₆ units of Sr₂IrO₄. While, the DFT calculations were performed to support the experimental standpoint that the thermal compressive strains in IrO₆ units downshift the d band center, thus enhancing the OER performances. Accordingly, our results provide a method to produce variable strains in catalysts with anisotropic thermal expansion.

Even accepting that single crystal nanoparticles were produced, I think that, from the performed characterization effort, that {001} is the dominant exposed facet cannot be concluded. Continuous lattice fringes along the entire length of the particle confirm crystallinity. Characteristic distance in the lattice fringes allows identifying the orientation of the observed crystal with respect to the observer. From this, the plausibility of certain facets being exposed can be argued. However, since the dominant shape of the nanoparticles has not been identified, I think that it is not possible to conclude which is the dominant exposed facet. Thus, I think that the obtained experimental and computational results cannot be reliably correlated. In the current version of the manuscript (line 71), that there exist "...a big possibility to expose {001} facet." is written. However, a big possibility of having exposed {001} facets does not implicate that this is the dominant exposed facet. In the current version of the manuscript (lines 67 and 68), the nanoparticles are described as having "...irregular shape..." being "...stunted single crystal". In the presence of nanoparticles having irregular shape, calculations on a single crystalline plane and experimental results cannot be reliably correlated. I do not understand why Reference 41 is maintained in the current version of the manuscript, to support the {001} facet focus, despite my comments, which have not been responded to by the authors.

Overall, the presentation of the characterization of the synthesized material seems confusing to me. In the original version of the manuscript (lines 58 to 61), that "The high-

resolution transmission electron microscope (HRTEM) lattice image confirmed that the Sr_2IrO_4 is highly crystalline and exhibits the continuous lattice fringes of 0.32 nm for (004) facet extending through the whole particle, evidencing that the particle is single crystal, as further confirmed by the fast Fourier transform image (Fig. 1a).” was written. Instead, in the current version of the manuscript (lines 65 to 68), the authors write that “The high-resolution transmission electron microscope (HRTEM) lattice image confirmed that the stunted Sr_2IrO_4 is highly crystalline and exhibits the continuous lattice fringes of 0.32 nm for (004) facet extending through the whole particle, evidencing that the particle with irregular shape is a stunted single crystal (Fig. 1a).” The fast Fourier transform image, that in the previous version of the manuscript was presented as an essential validation test, has been removed, and essential shape descriptions are now based on the relatively unspecific adjective “stunted”. In only four lines (66 to 70), “stunted” is used, up to four times, to qualify oxide (line 66), crystallinity (line 68), and particle (line 70). Whatever “stunted” means for the authors, I sincerely think that it cannot be appreciated from Fig. 1a. To support the {001} facet focus, the authors (starting at line 68) attempt “To disclose the growth mechanism of the stunted Sr_2IrO_4 single crystals...”, with explanations that, in my opinion, are somewhat speculative, rather irrelevant, and fail to achieve their purpose. The figure referred to in line 70 is not correct. The characterization of the synthesized material should be much more carefully presented.

Reply: Thanks for your questions. As pointed out by the Reviewer, in our case, the Sr_2IrO_4 particle is not a perfect single crystal with regular profile. Therefore, it is difficult to assign the dominant facet on the undeveloped crystal. Here, we check the relatively perfect crystal (**Supplementary Fig. 1c**) to suppose the possible growth mechanism of the Sr_2IrO_4 crystals, and thus assuming the possible exposed facets on the Sr_2IrO_4 particle. The SEAD and HRTEM suggested that the Sr_2IrO_4 particle tends to stack the {001} facet along [001] axis, thus exhibiting a possibility to expose {001} facet. Here, we assume that the Sr_2IrO_4 particles with irregular shape follow the similar crystal growth to the perfect ones due to that they grow under the same conditions. According to the Reviewer’s suggestions, we removed Reference 41 reported the undeveloped Sr_2IrO_4 particles from the manuscript.

And we also revised the corresponding descriptions. We described the Sr_2IrO_4 particles with irregular profile as “undeveloped particle” and the (001) facet is a possible exposed facet according to the crystal growth mechanism. Indeed, as pointed out by the Reviewer, due to the dominant shape of the nanoparticles has not been identified, calculations on a single Sr_2IrO_4 (001) surface and experimental results cannot be reliably correlated. Here, we calculated the free energy profiles on the Sr_2IrO_4 (110) surface, a possible exposed side facet according to the perfect Sr_2IrO_4 cuboid (**Figure 1 for review**), which exhibit the similar OER tendency with thermally induced strain effect (**Supplementary Fig. 16**). The rate-determining step is still *OH deprotonation step with a free energy difference between ΔG_{*OH} and ΔG_{*O} to be 1.80 eV under 25 °C and 1.12 eV under 90 °C.

Supplementary Fig. 1 | a, The XRD patterns for Sr_2IrO_4 before/after OER and carbon paper substrate. The XRD peak at 26.38° and 54.54° are assigned to the (002) and (004) crystal plane of graphitic carbon, respectively. **b**, SEM image for undeveloped Sr_2IrO_4 particles. The dotted lines trace the profile of the undeveloped Sr_2IrO_4 particles. **c**, SEM image for the relatively perfect Sr_2IrO_4 crystals formed by stacking (001) facet along the c-axis crystallographic direction. **d**, HRTEM lattice image for an undeveloped Sr_2IrO_4 single crystal with irregular shape. Inset shows the fast Fourier transform image. **e**, TEM image, HRTEM lattice image, and SAED pattern for a relatively perfect Sr_2IrO_4 single crystal to show the possible growth direction and possible exposed facet. The HRTEM lattice image and SAED pattern were collected on the yellow box area.

Figure 1 for review. Crystal profile of the Sr_2IrO_4 crystal.

Supplementary Fig. 16 | a-d, Structures of the Ir sites of the Sr₂IrO₄ (110) surface (a) and *OH (b), *O (c), *OOH (d) adsorptions on Sr sites. e,f, Gibbs free energy diagrams for OER intermediates (*OH, *O, *OOH) adsorbed on the Ir sites of the Sr₂IrO₄ (110) surface with thermal strains under 25 °C (e) and 90 °C (f). Solvation, temperature, and pH effects were included in the calculations.

Regarding the assumed adsorbates scenery (species, coverages, sites ...), under the different operation conditions (pH, potential, temperature...), I do not understand how many molecules of each intermediate adsorbate, including water, are involved in each computational model and why. I do not understand a.u. as a unit of surface coverage in Supplementary Fig. 13d. I do not understand why ticks, allowing values reading, are not shown in Supplementary Fig. 13d. The apparent presence of a different number of adsorbed molecules in the model series should be clarified (Fig. 4a and b, Supplementary Fig. 14).

Reply: Thanks for your questions. We calculated the OER activity on a single active site model, where the coverage for an intermediate adsorbate on the active site is 100%. The intermediate adsorbates, *OH, *O, and *OOH, were highlighted on each computational model. When we treat the solvent effects, the two fundamentally different approaches to modeling this aqueous solution environment are implicit (or continuum) and explicit (or atomistic) solvation models. Here, we utilize an implicit solvation model to describe the solvation effects through solving the linearized Poisson-Boltzmann equation by VASP_{sol}. In the implicit solvation model, we do not need to add the water molecules onto the computational model, but need to set the relative permittivity of water and the Debye length of the 1.0 M electrolyte. In this model, the solute density responds self-consistently to a surrounding dielectric continuum parametrized according to the identity of the solvent. This is a mean-field treatment of solvation in which solute flexibility can be described, in principle, through conformational sampling but is often only accounted for within the harmonic approximation. Therefore, we need to set the relative permittivity of water as 78.4 under 25 °C and 58.13 under 90 °C, but also, the Debye length of the 1.0 M electrolyte as 3.04 Å under 25 °C and 3.36 Å under 90 °C.

The surface coverage can be calculated by the $\theta(E) = [\int_{E_0}^E C_{\text{ads}}(E) dE] / \sigma$, where σ is the charge density for a monolayer coverage. Here, the σ was taken as 0.21 mC cm⁻², a typical

value for a monolayer coverage on Pt plate electrode (*J. Electroanal. Chem. Interfacial Electrochem.* 1987, 221, 1-21). Thus, the surface coverage is proportional to $[\int_{E_0}^E C_{\text{ads}}(E) dE]$. Here, we calculated the surface coverage by $\theta(E) = [\int_{E_0}^E C_{\text{ads}}(E) dE]/\sigma$ under different temperatures. The unit of surface coverage is a percentage.

Supplementary Fig. 13 | Electrochemical characterizations of OER intermediates adsorption. **a**, The equivalent circuit used in EIS fitting. R_s refers to the series resistance in the circuit. C_{dl} and C_{ads} are the capacitance from double layer and intermediates adsorption, respectively, the charge-transfer resistance, R_{ct} , is composed of R_1 and R_2 ($R_{ct} = R_1 + R_2$). **b,c** The C_{ads} from fitting EIS data (Fig. 3d,e) for Sr_2IrO_4 under 20 (**b**) and 90 °C (**c**), as a function of applied potentials. **d**, The calculated surface coverage for OER intermediates on Sr_2IrO_4 at around onset potential under 20 and 90 °C.

The geometric boundaries of the unit cells should be delineated in the figures.

Reply: Thanks for your valuable suggestions. We updated the slab models with geometric boundaries in **Supplementary Figs. 14-16**.

Supplementary Fig. 14 | a-d, Structures of the Ir sites of the Sr₂IrO₄ (001) surface (a) and *OH (b), *O (c), *OOH (d) adsorptions on Ir sites. e,f, Gibbs free energy diagrams for OER intermediates (*OH, *O, *OOH) adsorbed on the Sr sites of the Sr₂IrO₄ (001) surface with thermal strains under 25 °C (e) and 90 °C (f). Solvation, temperature, and pH effects were included in the calculations.

Supplementary Fig. 16 | a-d, Structures of the Ir sites of the Sr₂IrO₄ (110) surface (a) and *OH (b), *O (c), *OOH (d) adsorptions on Ir sites. e,f, Gibbs free energy diagrams for OER intermediates (*OH, *O, *OOH) adsorbed on the Ir sites of the Sr₂IrO₄ (110) surface with thermal strains under 25 °C (e) and 90 °C (f). Solvation, temperature, and pH effects were included in the calculations.

The presence of visually unbounded (from the surface) molecules in the models is confusing to me (Fig. 4b or Supplementary Fig. 14c).

Reply: Thanks for your question. In DFT calculations for OER in an alkaline media, in order to obtain a stable structure of surface slab model, we usually passivate the unsaturated terminal sites by hydroxyl group. Therefore, in our case, the visually unbounded molecules are hydroxyls to passivate the unsaturated Ir sites on the long-range-ordered surface except the one as active site. At the edge of the slab models, some Ir sites are not displayed completely. Thus, the hydroxyls look unbounded. To avoid misunderstanding, we remove the incompletely displayed information in Fig. 4b. The corresponding models with the geometric boundaries of the unit cells are clearly shown in **Supplementary Figs. 14-16**.

Better figures, showing the investigated configurations, should be provided.

Reply: Thanks for your valuable suggestion. We updated Fig. 5 as follows.

Fig. 5 | Energetics at catalyst-electrolyte interface with thermal strain effect. **a**, A scheme to describe the effect of compressive strain on the d band of transition metals and our thermal stimulation strategy to generate variable lattice strain. **b**, Effects of heating on the OER. The improved OER performances originate from that heating accelerates mass transfer and the thermal strains in IrO_6 units downshift the d-band center. The strong spin-orbital coupling in the low-spin Ir^{4+} ($5d^5$, $t_{2g}^5e_g^0$) 5d orbital in Sr_2IrO_4 splits the five electrons in the t_{2g} band into four electrons occupied as the $J_{\text{eff}} = 3/2$ sub-band and to one electron occupied as the $J_{\text{eff}} = 1/2$ sub-band. The thermal strains downshifting d-band center is real-time sensitive to the temperature. The thermal diffusion accelerating OER follows a linear $\ln k - T^{-1}$ Arrhenius relationship and the thermal strain accelerating OER follows the nonlinear $\ln k - T^{-1}$ Arrhenius relationship due to that downshifting d-band center will adjust the Ir-O bonding (σ) and antibonding (σ^*) states to meet the Sabatier principle.

I do not understand why pre-adsorbed molecules are not included in the models displayed in Supplementary Fig. 15 a-d. pH effects and/or potential could give rise to charged electrode, which should be considered, making it difficult the energies alignment corresponding to models having different composition.

Reply: Thanks for your question. We performed the DFT calculations on the Sr_2IrO_4 (001) surface with exposed Sr sites. To obtain the stable structure, the unsaturated Sr sites except the Sr active site were passivated by hydroxyls. The calculated results were updated in the Supplementary Fig. 15.

Supplementary Fig. 15 | a-d, Structures of the Sr sites of the Sr₂IrO₄ (001) surface (a) and *OH (b), *O (c), *OOH (d) adsorptions on Sr sites. **e,f**, Gibbs free energy diagrams for OER intermediates (*OH, *O, *OOH) adsorbed on the Sr sites of the Sr₂IrO₄ (001) surface with thermal strains under 25 °C (e) and 90 °C (f). Solvation, temperature, and pH effects were included in the calculations.

As pointed out by the Reviewer, pH effects and/or potential could give rise to charged electrode, which can be described by Eqs. 8-10:

$$\Delta G^{*OH}(T) = G^{*OH}(T) + 0.5 \times G_{H_2}(T) - G_{H_2O}(T) - G^*(T) - eU + pH \times k_B T \ln 10 \quad (8)$$

$$\Delta G^{*O}(T) = G^{*O}(T) + G_{H_2}(T) - G_{H_2O}(T) - G^*(T) - eU + pH \times k_B T \ln 10 \quad (9)$$

$$\Delta G^{*OOH}(T) = G^{*OOH}(T) + 1.5 \times G_{H_2}(T) - 2 \times G_{H_2O}(T) - G^*(T) - eU + pH \times k_B T \ln 10 \quad (10)$$

where, $-eU$ and $pH \times k_B T \ln 10$ are correction terms for charged electrode caused by external potential and pH effects, respectively. Indeed, these equations are based on the hypothesis that the composition of the catalyst have no change under different potential and pH. In our case, actually, the HRTEM observations have shown that the Sr₂IrO₄ is electrochemically stable without the visible surface reconstruction on catalyst after OER tests. Accordingly, we considered that the pH effects and/or potential could give rise to charged electrode and did not change the surface compositions.

To validate the applied numerical treatment (DFT[GGA] + U under the PBE functional),

the authors focus their effort on justifying the value adopted for the U parameter, as if an accurate value for U were in itself a sufficient (and not only necessary) condition for accurate calculations. Supporting the adopted value, Phys. Rev. Lett. 2008, 101, 226402 (DFT[LDA] + U under unspecified functional); Phys. Rev. Lett. 2012, 108, 086403 (DFT[GGA] + SOC + U under the PBE functional, in a multi-energy-scale scheme); Phys. Rev. Mater. 2018, 2, 075003 (BSE on full relativistic DFT + SOC + U under unspecified exchange-correlation treatment, not accurately reproducing GW + BSE); or Phys. Rev. Mater. 2019, 3, 063607 (DFT[GGA] + SOC + U under the PBE functional), are cited. Although DFT calculations are often referred to as “ab initio” or “first principles” calculations, parameters are actually embedded in the approximation to the essentially unknown exchange-correlation functional, for whose formulation there exist several approaches: LDA, GGA, hybrid, Meta... (AIP Conference Proceedings 577, 1 (2001); <https://doi.org/10.1063/1.1390175>). An inadequate numerical treatment of the self-interaction, correlation, spin-orbit coupling, relativistic effects, or spin multiplicity can compromise the validity of calculations on metal oxides involving 5d electrons. The variety of numerical treatments adopted by the authors of the cited references, which include components not considered by the authors, exemplify the suggested complexity.

Reply: According to the Reviewer’s suggestions, spin-orbit coupling effect cannot be neglected for 5d electrons. In order to simulate DOS and free energy profiles accurately, we performed the generalized gradient approximation with the Perdew-Burke-Ernzerh (GGA-PBE) functional and including spin-orbit coupling and electron-electron repulsion (PBE + SOC + U).

We performed all the calculations by PBE + SOC + U and updated the calculated methods described in the Methods: *To describe the electron correlation of Ir 5d states, DFT plus on-site repulsion U method and spin-orbital coupling effects (DFT + U + SOC) were used.*

Furthermore, not only properties of the materials have to be captured (as it is the case of the cited references), but also interaction with adsorbates. For instance, in a recent OER research on transition metal oxides (Nature Energy 2022, 7, 765), the hybrid exchange-correlation functional PBEo is considered. The accuracy of each numerical treatment depends on the system and property investigated: structure, energy, density, spectroscopy... (Science 355 (2017) 49). The accuracy of the adopted functional in this case should be established for DOSs and free energy profiles involved in the OER on bimetallic earth-alkaline/transition metal oxides.

Reply: Thanks for your valuable suggestions. Indeed, as pointed out by the Reviewer, the PBEo has higher accuracy to describe the electronic structure with strong correlation, such as 3d metal oxides (Nature Energy 2022, 7, 765). However, in recent report (Adv. Theory Simul. 2022, 2200513, <https://doi.org/10.1002/adts.202200513>), Ilaria Barlocco et al. have demonstrated that the reaction energies and electronic structures obtained with PBE have a substantial deviation from PBEo for first row TMs, i.e., 3d systems, while smaller deviations are found for the 4d and 5d series. Their results clearly clarify that the use of PBE appears acceptable for 4d and 5d metals with relatively weak localized d electrons, while in the case of 3d systems with strong localized d electrons PBE + U or PBEo approaches are recommended, in particular for magnetic ground states. Considering that the time cost of PBEo is about more than one thousand times higher than that of PBE, we used the PBE + U + SOC to calculate the

reaction energies and electronic structures of Ir 5d of Sr_2IrO_4 . We calculated the electronic structure after adsorption of OER intermediates onto (001) and (110) facets by PBE + U + SOC. Indeed, as shown in **Supplementary Figs. 18 and 19**, the calculated results indicated that the adsorption of $^*\text{OH}$, $^*\text{O}$, or $^*\text{OOH}$ will slightly affect the projected DOS of Ir 5d of Sr_2IrO_4 due to their interactions. However, after the interactions between OER intermediates and Ir active sites, the downshift of d band center is still dominated by the thermal strain.

Supplementary Fig. 18 | The projected DOS of Ir 5d for Sr_2IrO_4 (001) facet after adsorption of $^*\text{OH}$, $^*\text{O}$, or $^*\text{OOH}$ onto Ir active site. **a**, Ir active site without adsorbates. **b**, $^*\text{OH}$ adsorption. **c**, $^*\text{O}$ adsorption. **d**, $^*\text{OOH}$ adsorption.

Supplementary Fig. 19 | The projected DOS of Ir 5d for Sr_2IrO_4 (110) facet after adsorption of $^*\text{OH}$, $^*\text{O}$, or $^*\text{OOH}$ onto Ir active site. **a**, Ir active site without adsorbates. **b**, $^*\text{OH}$ adsorption. **c**, $^*\text{O}$ adsorption. **d**, $^*\text{OOH}$ adsorption.

It appears that the authors understand that, to accurately model the strain under different temperature conditions, experimentally derived lattice parameters must be used.

However, on the one hand, the free energy profiles of reactions at a given temperature are usually derived, from calculations performed using the lattice parameters corresponding to the ground state, correcting for zero-point energies and temperature-dependent entropies. On the other hand, since any adopted functional is only an approximation, the experimentally derived lattice parameters rarely match the corresponding ones in model chemistry. Most likely, the experimentally derived lattice parameters give rise to uncontrolled strain in the model chemistry. For this reason, experimentally derived lattice parameters are rarely used to establish computational models, with optimized lattice parameters under model chemistry being preferable. $a = b = 5.50$ at $25\text{ }^\circ\text{C}$ versus $a = b = 5.51$ at $90\text{ }^\circ\text{C}$ are in the order of the typical differences observed between experimentally derived and computationally optimized lattice parameters. The desirability of using experimentally derived lattice parameters does not seem obvious to me. The thermally induced strain simulation should be validated with care.

Reply: Thanks for your valuable suggestions. As pointed out by the Reviewer, we need to optimize the atom positions before calculating the free energy profiles. However, the structure optimization process is difficult to directly impose the temperature field. Indeed, a reliable route is that the free energy profiles of reactions at a given temperature are usually derived, from calculations performed using the lattice parameters corresponding to the ground state, correcting for zero-point energies and temperature-dependent entropies. However, in our case, the compressive strain generation in IrO_6 units is originated from the anisotropic thermal expansion. Here, in Sr_2IrO_4 , the anisotropic thermal expansion is typically originated from the thermally induced elementary microstructural deformation caused by transverse vibrations of the bridging atoms, thus producing lattice strains. In principle, the correcting for zero-point energies and temperature-dependent entropies is difficult to simulate the anisotropic thermal expansion. In order to simulate thermally induced strains to a great extent, we optimized the atom positions at the surface, while the atom positions in the subsurface layer are fixed according to the experimentally derived lattice parameters. In this situation, we can simulate the electronic states in the presence of the lattice strain and the free energy profiles of reactions at a given strain.

I believe that the presentation of the manuscript would still benefit from careful revision. For instance, I suggest “effective” by “efficient” in line 13. From line 15 to 20, “induce/d” is used up to three times with two different “inducing agents”: a material property “anisotropic thermal expansion”, and an energy form “heat”. A more rigorous and concise presentation of the abstract section would be desirable.

Reply: Thanks for your valuable suggestions. The corresponding sentences were revised as:

1. *Compressive strain, downshifting the d-band center of transition metal oxides, is an effective way to accelerate the sluggish kinetics of oxygen evolution reaction (OER) for water electrolysis.*
2. *Here, we find that anisotropic thermal expansion can produce compressive strains of the IrO_6 octahedron in Sr_2IrO_4 catalyst, thus downshifting its d-band center.*
3. *Different from the previous strategies to create constant strains in the crystals, the thermal-triggered compressive strains can be real-time tuned by varying temperature.*

In line 28, I assume that authors wanted to write “product” instead of “production”.

Reply: Thanks for your valuable suggestions. We use “product” instead of “production”.

I suggest reconsidering the construction “We are accordingly inspired by the thermal lattice strains...” (line 49).

Reply: This sentence was revised as “Accordingly, it is possible to utilize the thermal expansion effects as a convenient route to generate variable strains in OER catalysts.”

In the caption of Fig. 4, the investigated crystalline plane should be specified.

Reply: Thanks for your valuable suggestions. We write the investigated crystalline plane of (001) in the caption of Fig. 4.

The sentence “Our results imply that the materials with thermophysical effects can be applied to create the efficient water splitting with the simultaneous input of heat and electricity” (lines 283 to 285) seems to me unfortunate.

Reply: This sentence was revised as “Our results imply that the materials with thermal strain effects can be applied to create the efficient water splitting with the simultaneous input of heat and electricity.”

All the materials experiment thermophysical effects. Not all the catalysts exhibiting anisotropic thermal expansion would necessarily favor the water splitting when heated. The sentence “Heating to produce the compressive strains in crystals will effectively regulate the electronic energetics at electrode-electrolyte interface...” (lines 301 and 302) is not accurate enough.

Reply: This sentence was revised as “The thermal compressive strains in crystals with anisotropic thermal expansion will effectively regulate the electronic energetics at electrode-electrolyte interface.”

Heating would not produce compressive strain in all crystalline materials. The sentence “Our findings indicated that heating is an effective route to tune the electronic states of materials, thus significantly accelerating OER kinetics, beyond the traditional thermal diffusion effect.” (lines 304 and 305) seems to me unfortunate.

Reply: Our findings indicated that heating is an effective route to tune the electronic states of materials with anisotropic thermal expansion, thus significantly accelerating OER kinetics, beyond the traditional thermal diffusion effect.

Heating as a route to tune the electronic states of materials exhibiting isotropic thermal expansion has not been discussed. Heating would not necessarily accelerate the OER on all catalyst exhibiting anisotropic thermal expansion. A more rigorous and concise presentation of conclusions would be desirable.

Reply: Conclusions was revised as: *In summary, we successfully demonstrate that the d-band center of Sr₂IrO₄ can be easily tuned by heating to trigger compressive strains in its IrO₆ units, thus optimizing the OER kinetics by adjusting the binding strength between OER intermediates and Ir active species. Completely different from the previous routes to produce the constant strains in crystals (Fig. 5a), the thermally induced strains are easily tuned by varying*

temperature. The thermal compressive strains in crystals with anisotropic thermal expansion will effectively regulate the electronic energetics at electrode-electrolyte interface (Fig. 5b), for example, downshifting the d-band center to accelerate the OER kinetics and breaking the linear Arrhenius relationship to achieve higher energy conversion efficiency. Our findings indicated that heating is an effective route to tune the electronic states of materials with anisotropic thermal expansion, thus significantly accelerating OER kinetics, beyond the traditional thermal diffusion effect.

Given that the atom positions are relaxed, I suggest to remove the experimentally derived bond lengths (lines 365 and 366) from the methods section.

Reply: Thanks for your valuable suggestions. In our case, the compressive strain generation in IrO_6 units is originated from the anisotropic thermal expansion. Here, in Sr_2IrO_4 , the anisotropic thermal expansion is typically originated from the thermally induced elementary microstructural deformation caused by transverse vibrations of the bridging atoms, thus producing lattice strains. In principle, the correcting for zero-point energies and temperature-dependent entropies is difficult to simulate the anisotropic thermal expansion.

In order to simulate thermally induced strains to a great extent, we optimized the atom positions at the surface layer of slab model, while the atom positions in the subsurface are fixed according to the experimentally derived lattice parameters. In this situation, we can simulate the electronic states in the presence of the lattice strain and the free energy profiles of reactions at a given strain.

In the caption of Supplementary Fig.1, I assume that authors wanted to write “graphitic” instead of “graphic”.

Reply: Thanks for your valuable suggestions. We correct the word in the caption of Supplementary Fig.1.

Reviewers' comments:

Reviewer #3 (Remarks to the Author):

All my concerns have been addressed again. However, most of them still remain.

My doubts about the relevance of the contribution were already expressed in my first report. I will not insist on them. I assume that such relevance is clearer to the editors and other reviewers.

Recognizing the difficulties in reliably correlating experimental and computational results, the OER is now computationally investigated not only on the Sr₂IrO₄ (001) surface, but also on the Sr₂IrO₄ (110) one. The authors seem to understand that calculations on two likely exposed facets, instead of only one, increases the reliability of the correlation between experimental and computational results. However, in my opinion, in the presence of irregular particles, such a correlation cannot be reliably established. Perhaps, the goal of “accurately modeling an entire experimental process” should become the less compromised goal of “establishing the plausibility of the mechanism on a likely exposed facet”, modifying the presentation accordingly.

With adsorbed OH being an essential intermediate in this reaction, its consideration as a passivating agent, regardless of the adsorbates scenery discussion, is confusing to me. In fact, the use of the passivating agent concept in this context is confusing to me. Also, the idea of that the OER activity is investigated on “...a single active site model, where the coverage for an intermediate adsorbate on the active site is 100%.” is very confusing to me. In any case, the authors describe what they do, but not why. Water as a solvent is modeled by an implicit solvation model. However, since water is the oxidized species in this reaction, its explicit consideration as an adsorbate may be required. For each modeled experiment (concentrations, pH, potential, temperature...), the likely adsorbates scenery (species, coverages, sites...) should be identified (number of molecules of each relevant adsorbate per surface unit, preferred sites...) and modeled. Again, perhaps, the goal of “accurately modeling the entire experimental process under a broad variety of conditions” should become the less compromised goal of “establishing the plausibility of the mechanism on a likely exposed facet under a plausible adsorbates scenery”, modifying the presentation accordingly. In any case, each adsorbates scenery modeled should not only be described, but also justified and properly referenced in the manuscript. In addition, each assumed adsorbates scenery should be noted in the captions of the involved figures.

Acknowledging that the applied numerical treatment was not validated for the application, and after considering my comments about the numerical treatments applied in the references provided by the authors in their rebuttal letter [Phys. Rev. Lett. 2008, 101, 226402 (DFT[LDA] + U under unspecified functional); Phys. Rev. Lett. 2012, 108, 086403 (DFT[GGA] + SOC + U under the PBE functional, in a multi-

energy-scale scheme); Phys. Rev. Mater. 2018, 2, 075003 (BSE on full relativistic DFT + SOC + U under unspecified exchange-correlation treatment, not accurately reproducing GW + BSE); and Phys. Rev. Mater. 2019, 3, 063607 (DFT[GGA] + SOC + U under the PBE functional)], spin orbit effects (DFT[GGA] + SOC + U under the PBE functional), and not just DFT[GGA] + U under the PBE functional, as numerical treatment, are now included in the calculations. However, not only the material properties have to be captured (as it is the case of the cited references), but also interaction with adsorbates. After considering Nature Energy 2022, 7, 765 (OER on cobalt oxide, using the PBE0 functional as a reference), and supported by the conclusions reached in Adv. Theory Simul. 2022, 2200513 (HER and OER on SACs based on TM + N-doped graphene), arguing that PBE0 is three orders of magnitude more computationally expensive than PBE, the latter is retained as the exchange correlation functional in the applied numerical treatment. However, the catalysts investigated in the latter reference are different from transition metal oxides. Thus, although the applied numerical treatment is now much better supported, I consider that it is not yet rigorously validated for the application. In any case, after validation, the applied numerical treatment should not only be described, but also justified and properly referenced in the manuscript.

I understand the difficulties of modeling thermal induced strains. However, I consider that the approach of directly using experimentally obtained lattice parameters is a very risky procedure, which has not been reasonably validated. I believe that the lattice parameters optimized under the model chemistry should always be considered in any reasonable approach. Perhaps, applying a thermal expansion coefficient. Once again, perhaps, the goal of “accurately modeling this thermal effect” should become the less compromised goal of “establishing the plausibility of the mechanism as a thermally induced effect”, modifying the presentation accordingly.

The presentation of the manuscript still requires careful revision. In my last report, I provided examples of that. Not only the noted examples should have been carefully revised. For instance, the idea “Oxygen evolution reaction (OER), the bottleneck reaction for hydrogen production by water splitting, consumes excessive energy due to its sluggish kinetics.”, (lines 24 to 25, in the three submitted versions of the manuscript), which obviously depend on the catalyst, should be better introduced. In addition, I believe that some of the provided examples were not carefully revised, revealing haste and lack of consideration for the referee. For instance, in my last report, I pointed out that the sentence “Our results imply that the materials with thermophysical effects can be applied to create the efficient water splitting with the simultaneous input of heat and electricity” (lines 283 to 285, in the previous version of the manuscript) seemed unfortunate to me. Supporting such consideration, I wrote that “All the materials experiment thermophysical effects.”, and that “Not all the catalysts exhibiting anisotropic thermal expansion would necessarily favor the water splitting when heated.”. In the current version of the manuscript, the problematic sentence has been substituted by “Our results imply that the materials with thermal strain effects can be applied to create the efficient water splitting with the simultaneous input of heat and electricity.”. However, as already was explained, I believe that not all the materials with thermal strain effects would necessarily favor the water splitting with the simultaneous input of heat and electricity. For instance, in my last report, I pointed out that the sentence “Our findings indicated that heating is an effective route to tune the electronic states of materials, thus significantly accelerating OER kinetics, beyond the traditional thermal diffusion effect.” (lines 304 and 305, in the previous version of the

manuscript) seemed unfortunate to me. Supporting such consideration, I wrote that “Heating as a route to tune the electronic states of materials exhibiting isotropic thermal expansion has not been discussed.”, and that “Heating would not necessarily accelerate the OER on all catalyst exhibiting anisotropic thermal expansion.” In the current version of the manuscript, the problematic sentence has been substituted by “Our findings indicated that heating is an effective route to tune the electronic states of materials with anisotropic thermal expansion, thus significantly accelerating OER kinetics, beyond the traditional thermal diffusion effect”. However, as already was explained, I believe that heating would not necessarily accelerate the OER on all catalyst exhibiting anisotropic thermal expansion.

Response to the Reviewers' comments

Reviewer #3 (Remarks to the Author):

Many thanks for your hard work in helping us improve the quality of our research. Your valuable suggestions deepened our understanding of these problems you have pointed out. In our work, as an experimental research, we provide various experimental evidences, including in-situ XRD to confirm the anisotropic thermal expansion in Sr_2IrO_4 , the Raman spectra and the temperature-dependent EXAFS to confirm the thermal strains in IrO_6 units, and the X-ray photoelectron spectroscopy (XPS) valence band spectra to show the downshifting of d band center in the presence of thermal strains. To deepen the understanding of effects of thermal strains on OER, we also carried out the DFT calculations as the additional results to help understanding the experimental results.

As pointed out by the Reviewer, there are the recognized difficulties in reliably correlating experimental and computational results. Indeed, in principle, the theoretical calculation is infinitely to approach the truth, but never the same. Accordingly, usually, the DFT results are an additional evidence to provide a possible understanding of the experimental conclusions. However, to carry out the theoretical calculations, we should consider a compromise between the time cost and the computational accuracy during the modeling and choosing the calculation method. As suggested by the Reviewer, we carried out the DFT calculations to consider as many factors as possible. Indeed, in the previous submissions, there are many imprecise expressions in describing the theoretical results. We totally agree with the Reviewer's suggestions that we should describe the details of theoretical calculations and provide the scientific basis for the application of numerical treatment options, which can help the readers understand the correlations between experimental and computational results.

We sincerely thanks the Reviewer's help in precise expression of our theoretical results. And we have revised these presentations you pointed out in whole text. We hope these revisions will meet the high standard in the scientific expression.

All my concerns have been addressed again. However, most of them still remain. My doubts about the relevance of the contribution were already expressed in my first report. I will not insist on them. I assume that such relevance is clearer to the editors and other reviewers.

Reply: Thanks for your time. In the revised version, we make every effort to clarify your concerns and we hope you find our reply satisfactory.

Recognizing the difficulties in reliably correlating experimental and computational results, the OER is now computationally investigated not only on the Sr_2IrO_4 (001) surface, but also on the Sr_2IrO_4 (110) one. The authors seem to understand that calculations on two likely exposed facets, instead of only one, increases the reliability of the correlation between experimental and computational results. However, in my opinion, in the presence of irregular particles, such a correlation cannot be reliably established. Perhaps, the goal of "accurately modeling an entire experimental process" should become the less compromised goal of "establishing the plausibility of the mechanism on a likely exposed facet", modifying the presentation accordingly.

Reply: Thanks. We totally agree with the Reviewer's opinion that for the nearly irregular particles the DFT results on the (001) and (110) facets are the representative cases to verify the plausibility of the mechanism of thermal strains promoted OER. Here, we give comprehensive consideration in selecting the likely exposed (001) facet to carry out the DFT calculations. Firstly, the (001) facet is a thermodynamically stable plane in the Sr_2IrO_4 (*Sci. Rep.*

2013, 3, 3073), which is able to reflect at least partially the bulk properties of Sr_2IrO_4 . In addition, the layered Sr_2IrO_4 is easy to cleave along the (001) facet due to the weak interlayer interactions (Sci. Rep. 2013, 3, 3073), increasing the probability to expose it. In addition, as shown in **Supplementary Fig. 1b**, although the single-crystal Sr_2IrO_4 particles are nearly irregular, the stacking growth of (001) facets is visible as traced by dotted lines. Therefore, we carried out the DFT calculations on the likely exposed facets to establish the plausibility of the mechanism of thermal strains promoted OER.

We have revised the corresponding presentations as:

To elucidate effects of the thermal strains on OER kinetics, we established the plausibility of the mechanism of thermal strains promoted OER on the likely exposed facets by theoretical calculations.

Supplementary Fig. 1 | b, SEM image for undeveloped Sr_2IrO_4 particles. The dotted lines trace the profile of the undeveloped Sr_2IrO_4 particles.

With adsorbed OH being an essential intermediate in this reaction, its consideration as a passivating agent, regardless of the adsorbates scenery discussion, is confusing to me. In fact, the use of the passivating agent concept in this context is confusing to me. Also, the idea of that the OER activity is investigated on “... a single active site model, where the coverage for an intermediate adsorbate on the active site is 100%.” is very confusing to me. In any case, the authors describe what they do, but not why. Water as a solvent is modeled by an implicit solvation model. However, since water is the oxidized species in this reaction, its explicit consideration as an adsorbate may be required. For each modeled experiment (concentrations, pH, potential, temperature...), the likely adsorbates scenery (species, coverages, sites...) should be identified (number of molecules of each relevant adsorbate per surface unit, preferred sites...) and modeled. Again, perhaps, the goal of “accurately

modeling the entire experimental process under a broad variety of conditions” should become the less compromised goal of “establishing the plausibility of the mechanism on a likely exposed facet under a plausible adsorbates scenery”, modifying the presentation accordingly. In any case, each adsorbates scenery modeled should not only be described, but also justified and properly referenced in the manuscript. In addition, each assumed adsorbates scenery should be noted in the captions of the involved figures.

Reply: We will response your questions in following three parts.

1) With adsorbed OH being an essential intermediate in this reaction, its consideration as a passivating agent, regardless of the adsorbates scenery discussion, is confusing to me. In fact, the use of the passivating agent concept in this context is confusing to me. Also, the idea of that the OER activity is investigated on “...a single active site model, where the coverage for an intermediate adsorbate on the active site is 100%.” is very confusing to me. In any case, the authors describe what they do, but not why.

Reply: In DFT calculations, to passivate the unsaturated sites of the surface of oxide catalysts is a common method for obtaining a stable surface slab model without severe reconstruction during structure optimization. Usually, the OH passivating agent is used to passivate the unsaturated sites of oxides in the alkaline water electrolysis due to that surface hydroxylation of oxides is a thermodynamically favorite process in the alkaline environment (*J. Phys. Chem. C* 2011, 115, 26, 12901-12907; *Angew. Chem.* 2014, 126, 13622-13626). In our case of alkaline water electrolysis on Sr_2IrO_4 , the surface hydroxylation of Sr_2IrO_4 is a thermodynamically favorite process due to the abundant OH^- in the 1 M KOH electrolyte. Accordingly, we adopt the OH^- to passivate the coordinately unsaturated sites of surface slab model of Sr_2IrO_4 , thus stabilizing the surface structure and avoiding the undesired surface reconstruction during structure optimization. If there are various coordination environments for unsaturated sites on the studied terminations, we should firstly optimize to find out which sites are the possible active site for OER (*Adv. Mater.* 2019, 31, 1804769). In the (001) and (110) facets of Sr_2IrO_4 , the coordinately unsaturated Ir atoms share the same 5-coordinated environment (**Supplementary Fig. 13**), that is, the Ir sites on two facets of Sr_2IrO_4 are equivalent, thus we can preset any one of the unsaturated Ir sites as the active site. Correspondingly, although the $^*\text{OH}$ is an essential intermediate of OER reaction, the adsorbed energy is the energy difference between the slab model (OH passivating (001) facet or (110) facet) before and after $^*\text{OH}$ adsorption (**Response Fig. 1**). That is, during the calculations of the adsorbed energy, the DFT calculations are equivalently to treat these passivated OH at unsaturated Ir sites and the adsorbed OH at the active site. This means that all the Ir sites on the surface have the same thermodynamic energy requirement for adsorption of OER intermediates, thus we can preset an active site to adsorb the $^*\text{OH}$ to calculate the adsorbed energy by an energy difference before and after $^*\text{OH}$ adsorption. That is, for thermodynamic calculations, the adsorbed energy is the energy difference between the slab model with and without adsorbates at each elementary step. In addition, considering that the experimental evidence have exhibited the satisfied stability of Sr_2IrO_4 during OER (No surface reconstruction after OER i-t test at 1.45 V at 90 °C for 1 h is observed by the HRTEM in Fig.1a), we use the single-site adsorbate evolution mechanism (AEM) model to calculate the Gibbs free energy change. In this model, there is no evolution of adsorbates between the adjacent sites or the simultaneous evolution of adsorbates on all active sites at the same time. If there is a simultaneous evolution of adsorbates on all active sites at the same time, this would mean that the adsorbed species at adjacent sites have strong interactions, leading to that the evolution of adsorbates deviates from the single-site AEM model to follow the multi-site AEM model (*Energy Environ. Sci.*, 2015, 8, 1404-1427).

The adsorbed energy of $^*\text{OH}$: $\Delta E_{^*\text{OH}} = E(\text{slab-OH adsorbate}) - [E(\text{active OH}^-) + E(\text{slab})]$

The adsorbed energy of $^*\text{O}$: $\Delta E_{^*\text{O}} = E(\text{slab-O adsorbate}) - [E(\text{active O}) + E(\text{slab})]$

The adsorbed energy of $^*\text{OOH}$: $\Delta E_{^*\text{OOH}} = E(\text{slab-OOH adsorbate}) - [E(\text{active OOH}) + E(\text{slab})]$

Response Fig. 1 A scheme to show the calculation method of adsorbed energy of OER intermediates.

As for the coverage of adsorbates on the active sites, the active sites are occupied by adsorbates entirely defined as 100% coverage, that is, one monolayer (1 ML) (*Nat. Commun.* 2022, 13, 5788; *Phys. Chem. Chem. Phys.* 2015, 17, 21643). In our case, the single-site adsorption model with 100% coverage of adsorbates on Ir active site is beneficial to reflect the intrinsic catalytic activity of Sr_2IrO_4 , avoiding the possible interactions between adjacent sites. All Gibbs free energy changes for the OER intermediates, $^*\text{OH}$, $^*\text{O}$, and $^*\text{OOH}$ were calculated on the surface slab model of Sr_2IrO_4 with OH^- passivation except the active site. Therefore, in our case, the 0 and 100% coverage is for the active site occupied without and with the adsorbate, respectively.

2) Water as a solvent is modeled by an implicit solvation model. However, since water is the oxidized species in this reaction, its explicit consideration as an adsorbate may be required. For each modeled experiment (concentrations, pH, potential, temperature...), the likely adsorbates scenery (species, coverages, sites...) should be identified (number of molecules of each relevant adsorbate per surface unit, preferred sites...) and modeled.

Reply: According to the Reviewer's suggestions in the previous comments, the various factors including the concentrations, pH, potential, and temperature were needed to be considered during DFT calculations. When we treat the solvent effects, the two fundamentally different approaches to modeling this aqueous solution environment are implicit and explicit solvation models (*J. Phys. Chem. C* 2019, 123, 18467-18474). Implicit solvation methods approximate the solvent through a continuum model based on electrostatics. Although this approach works well for long-range effects, the method cannot account for the immediate vicinity of an adsorbate in contact with a protic solvent such as liquid water, which interacts significantly with solutes through hydrogen bonds and van der Waals interactions. Explicit solvation methods can employ one or several layers of water molecules in a DFT calculation to compute the binding energies of adsorbates. These water layers are placed above the surface and can be ice-like or undergo structural relaxation. If the water layer is allowed to relax, a doubt remains whether the optimized water structure is an accurate representation of the liquid at the solvated interface. In any case, DFT structural relaxations are performed at 0 K, and therefore such schemes fail in describing the finite temperature behavior of liquid solvents. Furthermore, relaxation of the water layer near the adsorbate can lead to an overestimate of the stabilization due to hydrogen bonds, some of which can be disrupted in finite temperature dynamics where disorder is taken into account. This means that the explicit solvation model with relatively high uncertainty is more complex than the implicit solvation model, in particular, to consider the various factors including the concentrations, pH, potential, and temperature may increase the uncertainty of this approach.

In addition, usually, for alkaline water electrolysis on surface of oxides, the water dissociation is a spontaneous process to form OH^- on the unsaturated sites. As demonstrated on the surface of IrO_2 (*J. Am. Chem. Soc.* 2017, 139, 149-155). For the stoichiometric IrO_2 (110) surface, half of the surface Ir atoms are 5 coordinated. The H_2O binds strongly (by ~ 1.7 eV/ H_2O in liquid water) at this surface and the H_2O molecules spontaneously dissociate to form ^-OH at the unsaturated Ir (Ir-5c) and ^-OH at the bridging O, independent of the starting surface water configurations. Indeed, in the alkaline water electrolysis, oxides or hydroxides have been demonstrated to be the efficient catalysts to accelerate water dissociation. For example, the alkaline water electrolysis on Pt electrode is effectively promoted by the Fe, Co, Ni, Mn containing hydr(oxy)oxides (*Science* 2011, 334, 1256-1260; *Nat. Mater.* 2012, 11, 550-557). This means that the initial water dissociation on surface of oxides is commonly spontaneous process to occur, thus the implicit solvation model is able to describe the solvation environment for checking the evolution of OER intermediates ($^*\text{OH}$, $^*\text{O}$, $^*\text{OOH}$).

Accordingly, after we give comprehensive consideration to the pros and cons of explicit solvation model and implicit solvation model and the spontaneous H_2O dissociation on the surface of oxides, the implicit solvation model was adopted to consider the solvation effect involved with the effects of temperature, pH, and concentrations. That is, considering the

spontaneous H₂O dissociation on the surface of most oxides, we adopted the implicit solvation model based on its advantage in describing the temperature effect, pH, and concentrations. In the implicit solvation model, we set the relative permittivity of water and the Debye length of the 1.0 M electrolyte. In this model, the solute density responds self-consistently to a surrounding dielectric continuum parametrized according to the identity of the solvent. This is a mean-field treatment of solvation in which solute flexibility can be described, in principle, through conformational sampling but is often only accounted for within the harmonic approximation. Therefore, the effects of temperature, pH, and concentrations on solvation effect can be considered by setting the relative permittivity of water as 78.4 under 25 °C and 58.13 under 90 °C, but also, the Debye length of the 1.0 M electrolyte as 3.04 Å under 25 °C and 3.36 Å under 90 °C.

All in all, implicit solvation model was used after considering the spontaneous H₂O dissociation on the surface of most oxides and the performances of the solvation model in describing the effects of the concentrations, pH, and temperature.

3) Again, perhaps, the goal of “accurately modeling the entire experimental process under a broad variety of conditions” should become the less compromised goal of “establishing the plausibility of the mechanism on a likely exposed facet under a plausible adsorbates scenery”, modifying the presentation accordingly. In any case, each adsorbates scenery modeled should not only be described, but also justified and properly referenced in the manuscript. In addition, each assumed adsorbates scenery should be noted in the captions of the involved figures.

Reply: We have revised the corresponding expression as “establishing the plausibility of the mechanism on a likely exposed facet under a plausible adsorbates scenery”. We have noted the justification and the reference in the manuscript. The each assumed adsorbates scenery was noted in the captions of the involved figures.

Acknowledging that the applied numerical treatment was not validated for the application, and after considering my comments about the numerical treatments applied in the references provided by the authors in their rebuttal letter [Phys. Rev. Lett. 2008, 101, 226402 (DFT[LDA] + U under unspecified functional); Phys. Rev. Lett. 2012, 108, 086403 (DFT[GGA] + SOC + U under the PBE functional, in a multi-energy-scale scheme); Phys. Rev. Mater. 2018, 2, 075003 (BSE on full relativistic DFT + SOC + U under unspecified exchange-correlation treatment, not accurately reproducing GW + BSE); and Phys. Rev. Mater. 2019, 3, 063607 (DFT[GGA] + SOC + U under the PBE functional)], spin orbit effects (DFT[GGA] + SOC + U under the PBE functional), and not just DFT[GGA] + U under the PBE functional, as numerical treatment, are now included in the calculations. However, not only the material properties have to be captured (as it is the case of the cited references), but also interaction with adsorbates. After considering Nature Energy 2022, 7, 765 (OER on cobalt oxide, using the PBEo functional as a reference), and supported by the conclusions reached in Adv. Theory Simul. 2022, 2200513 (HER and OER on SACs based on TM + N-doped graphene), arguing that PBEo is three orders of magnitude more computationally expensive than PBE, the latter is retained as the exchange correlation functional in the applied numerical treatment. However, the catalysts investigated in the latter reference are different from transition metal oxides. Thus, although the applied numerical treatment is now much better supported, I consider that it is not yet rigorously validated for the application. In any case, after validation, the applied numerical treatment should not only be described, but also justified and properly referenced in the manuscript.

Reply: Thanks for your time. Here, after giving comprehensive consideration to various factors including the Ir⁴⁺ with 5d⁵ electron configuration, exchange-correlation functionals under DFT infrastructure, and the validated availability of theoretical methods used in the Ir⁴⁺-containing oxides, we used the GGA-PBE + SOC + U to calculate the d band center and the adsorbed Gibbs free energy on Sr₂IrO₄ with Ir 5d⁵ electron configuration.

The reason for the selection of the theoretical method is as follows:

To clearly understand the plausibility of the theoretical methods we used, we would

need to look back on the DFT infrastructure for helping to select the available exchange-correlation functional for theoretical calculations of Sr₂IrO₄.

Density-functional theory (DFT) is an approach to the many-electron problem in which the electron density, rather than the many-electron wave function, plays the central role (Rev. Mod. Phys. 2008, 80, 3-59). The central tenet of DFT is the Hohenberg-Kohn theorem (Phys. Rev. 1964, 136, B864), an important variational principle, which tells us that the ground-state density can be found by minimizing a total energy expression of the form of $E_{\text{tot}} = \int v_{\text{ext}}(\mathbf{r})n(\mathbf{r})d\mathbf{r} + F[n]$, where $n(\mathbf{r})$ is ground-state density of a system of interacting electrons in some external potential $v_{\text{ext}}(\mathbf{r})$, $F(n)$ is a universal functional of the density. However, a challenge is that the form of $F(n)$ is not known, and sufficiently accurate explicit approximations have not yet emerged. Fortunately, Kohn and Sham (Phys. Rev. 1965, 140, A1133) used equation of $E_{\text{tot}} = \int v_{\text{ext}}(\mathbf{r})n(\mathbf{r})d\mathbf{r} + F[n]$ to construct a more practical set of equations. They considered a fictitious system of noninteracting electrons and postulated the existence of an external potential $v_{\text{KS}}(\mathbf{r})$, such that the ground-state density obtained from the fictitious, noninteracting electron system is equal to the ground-state density of the given real, interacting-electron system. If such a potential exists, then solving a set of one-electron Schrödinger equations of the form of $[(\hbar^2\nabla^2)/2m + v_{\text{KS}}(\mathbf{r})]\phi_i(\mathbf{r}) = \varepsilon_i \phi_i(\mathbf{r})$, where \hbar is the reduced Planck constant, m is the electron mass, and ε_i and ϕ_i are known as the Kohn-Sham energies and orbitals, respectively. The density $n(\mathbf{r})$ was calculated as a sum over filled orbitals, $n(\mathbf{r}) = \sum_{i, \text{occupied}} |\phi_i(\mathbf{r})|^2$, and $v_{\text{KS}}(\mathbf{r})$ was described by $v_{\text{KS}}(\mathbf{r}) = v_{\text{ext}}(\mathbf{r}) + \int n(\mathbf{r}')/|\mathbf{r}-\mathbf{r}'|d^3\mathbf{r}' + v_{\text{xc}}[n];\mathbf{r}$ where v_{xc} is an additional potential to consider many-body physics $v_{\text{xc}}[n];\mathbf{r} = \delta E_{\text{xc}}/\delta n(\mathbf{r})$. E_{xc} is the exchange-correlation energy, which must include all nonclassical electron interactions, namely, Pauli exchange, electron correlation, as well as the difference between the kinetic energy of the interacting- and noninteracting-electron systems. Despite the elegance of the Kohn-Sham equation, it would remain useless in practice without adequate approximations for the exchange-correlation functional E_{xc} . Numerous approximations, including local density approximation (LDA), or its spin-dependent version, the local spin density approximation (LSD), generalized gradient approximation (GGA), meta-GGA, and hybrid-GGA, were developed to describe the exchange-correlation energy. The local density approximation (LDA) is the first generation of functionals, which postulates the electron density of the system as uniform electron gas. However, it sometimes overestimates the chemical bonds and underestimates the barrier heights (Phys. Chem. Chem. Phys. 2019, 21, 23782-23802). Subsequently, other functionals were developed to obtain more accurate calculation results, such as generalized gradient approximation (GGA). The impact of GGA has been quite dramatic, especially in quantum chemistry where DFT is now competitive in accuracy with more traditional methods while being computationally less expensive.

Noting that in addition to electron density, the exchange energy and correlation energy of the system also depend on the density gradient, the GGA for the exchange-correlation energy includes information on deviations from homogeneity by considering the gradients of the spin-polarized charge densities, as described by: $E_{\text{xc}}^{\text{GGA}}[n_{\uparrow}(\mathbf{r}), n_{\downarrow}(\mathbf{r})] = \int f(n_{\uparrow}(\mathbf{r}), n_{\downarrow}(\mathbf{r}), |\nabla n_{\uparrow}(\mathbf{r})|, |\nabla n_{\downarrow}(\mathbf{r})|)d^3\mathbf{r}$. In GGA, $f(n)$ is constructed to reproduce the exact result in certain limits, e.g., the slowly varying and rapidly varying limits, and obeys many important properties of the exact functional. PBE is an empirical avenue for determining $f(n)$ (Phys. Rev. Lett. 1996, 77, 3865; J. Chem. Phys., 1999, 110, 5029-5036), is the most commonly used in solid-state physics and small molecules. The PBE functional is constructed in such a way that all the essential conditions for the reliability of the local spin-density approximation are preserved. Further physical constraints are satisfied using the dependence on the reduced density gradient $s = |\nabla n|/(2k_F n)$, with $k_F = (3\pi^2 n)^{1/3}$. The PBE exchange-correlation energy depends on n , s , and ζ and is expressed as: $E_{\text{xc}}^{\text{PBE}} = \int n(\mathbf{r})\varepsilon_{\text{xc}}^{\text{PBE}}(r_s(\mathbf{r}), s(\mathbf{r}), \zeta(\mathbf{r})) d^3\mathbf{r}$, where $\zeta = (n_{\uparrow} - n_{\downarrow})/n$ is the spin-polarization and $r_s = (4\pi n/3)^{-1/3}$ is the Wigner-Seitz radius.

Compared to the empirical PBE functional, hybrid functionals make up a special category of exchange-correlation functionals that mix some fraction of Fock exchange into a semilocal functional. This approach is inspired by the well-known Hartree-Fock (HF) method, which is the exact solution for uncorrelated systems. The nonlocal Fock potential reads $v_x^{\text{HF}}(\mathbf{r}, \mathbf{r}') =$

$(-e^2/2)\sum_{nk}f_{nk}\phi_{nk}^*(\mathbf{r}')\phi_{nk}(\mathbf{r})/|\mathbf{r}-\mathbf{r}'|$, where f_{nk} is the occupation of band n and at a specific k point, \mathbf{r} is the position and e the charge of an electron. This equation depends on the KS orbitals instead of the density and thus, hybrid functionals are not density functionals. PBE0 is one of the hybrid functionals and has the following composition (*J. Chem. Phys.* 1996,105, 9982; *J. Chem. Phys.* 1999, 110, 6158-6170): $E_{xc}^{PBE0} = 1/4E_x^{HF} + 3/4E_x^{PBE} + E_c^{PBE}$. Here, E_x^{HF} is the Fock exchange energy, while E_x^{PBE} and E_c^{PBE} are the PBE exchange and correlation energies, respectively. Compared to many other functionals, PBE0 is usually well fitted to experimental data with relatively less errors but computationally time-consuming. Unfortunately, Hybrid functionals also introduce some degree of empiricism, this is, it need to be validated how much percentage of Fock exchange energy added to the E_{xc} (*J. Phys. Chem. C* 2011, 115, 3716-3721).

Through a review and summary on the main characteristic of these functionals, we can know that for a given system we need to select the appropriate functionals to present the material characteristics we care about, while considering a compromise between computational cost and physical accuracy. Indeed, for Ir-based oxides, in the previous reports, the electronic properties were investigated by the different functionals. For example, the DFT[LDA] + U+ SOC was used to describe the d band electronic structures of Sr₂IrO₄, in good accord with the spectroscopic results (*Phys. Rev. Lett.* 2008, 101, 226402). Another example is that the electronic structure of Sr₂IrO₄ was studied by DFT[GGA] + SOC + U under the PBE functional, in a multi-energy-scale scheme (*Phys. Rev. Lett.* 2012, 108, 086403), in describing the magnetic states with a good agreement with the experimental results; BSE on full relativistic DFT + SOC + U under unspecified exchange-correlation treatment was carried out to check the the electronic and optical properties of Sr₂IrO₄ (*Phys. Rev. Mater.* 2018, 2, 075003), and found that mBSE qualitatively captures the characteristic two-peak structure but the overall spectra deviates substantially from those obtained from the full GW+BSE procedure; and DFT[GGA] + SOC + U under the PBE functional was carried out to explain an exotic low spin state in tetrahedral coordination of Sr₉Ir₃O₁₇ (*Phys. Rev. Mater.* 2019, 3, 063607).

In describing the transition metal oxides, the PBE with inclusion of a Hubbard-type effective on-site Coulomb term (U), which is computationally inexpensive, often provides a computationally tractable and physically reasonable approach for transition metal oxides (*Phys. Rev. B*, 2018, 97, 035117). In our case, we mainly focus on the adsorption energy and the d band center change. Considering that the GGA + U under the PBE functional has demonstrated to be effective in solving d band center position and OER energy profiles for Ir⁴⁺(5d⁵)-based oxides to understand their electrocatalytic properties (*Nat. Energy* 2016, 2, 16189, GGA-PBE+U+SOC for OER on La₂LirO₆; *J. Phys. Chem. C* 2015, 119, 11570-11577 GGA-PBE+U+SOC for OER on IrO₂).

Usually, for transition-metal oxides, the localized d states yield strongly correlated bands with a on-site Coulomb repulsion U . (*Phys. Rev. B* 2005, 71, 035105). In our case, the U value of 2 eV was adopted to consider the on-site d -electron Coulomb interaction effect for partly overcoming the DFT (GGA under the PBE functional) delocalization error (*J. Phys. Chem. C* 2015, 119, 11570-11577). In addition, the Sr₂IrO₄ with an electron configuration of 5d⁵ exhibits the strong spin-orbit coupling (SOC) effects. At this point, it is natural to consider the SOC effects in Sr₂IrO₄ since the SOC effects is much larger than that in 3d and 4d oxides (*Phys. Rev. Lett.* 2008, 101, 076402; *Phys. Rev. Lett.* 2008, 101, 226402).

As for PBE0 used for calculations of OER Gibbs free energy on CoO_x(OH)_y with coexistence of Co²⁺ and Co³⁺ in *Nature Energy* 2022, 7, 765, as stated by the authors, the PBE+U is incapable of correcting the delocalization error of pure PBE for the catalysts with multiple metal oxidation states (*Phys. Rev. B* 2011, 83, 245204). So, the PBE0 is necessary to meet the accuracy requirement for the CoO_x(OH)_y with coexistence of multiple metal oxidation states because a single value of U is incapable of correcting the delocalization error of pure PBE. However, in our case of Sr₂IrO₄, all of Ir sites are Ir⁴⁺(5d⁵) configuration, and the PBE+U is capable of describing the properties of Sr₂IrO₄, as demonstrated by the consistency of the

experimental and theoretical results. Here, as shown in **Response Fig. 2** (also shown in **Supplementary Fig. 19**), the density of states computed within PBE+U+SOC is comparable with the experimental XPS valence band spectrum, suggesting that the PBE+U+SOC is acceptable for the theoretical calculations of Sr_2IrO_4 . Indeed, the metal-like electronic states of Sr_2IrO_4 with thermal strain computed within PBE+U is in good agreement with an experimental fact that the Sr_2IrO_4 at 20 - 90 °C is paramagnetic state (Néel temperature at 240 K) (*Adv. Mater.* 2020, 32, 1904508). This conclusion is also confirmed in the density of states of IrO_2 computed within PBE+U+SOC (*J. Phys. Chem. C* 2015, 119, 11570-11577).

Response Fig. 2 a, The valence band spectra of Sr_2IrO_4 under 25 °C and 90 °C. **b**, The projected DOS of Ir 5d of Sr_2IrO_4 under 25 °C and 90 °C.

As pointed out by the Reviewer, not only the material properties have to be captured (as it is the case of the cited references), but also interaction with adsorbates. We provided the Bader charge analysis to check the interactions between Ir active site and OER intermediates. As shown in **Supplementary Fig. 15**, Bader charge analysis revealed that about 0.38-0.5 electrons were transferred from OER intermediates into Ir active site during the adsorption to proceed, confirming that the Ir active site is able to extract electrons from OER intermediates via their interactions. The higher electron transfer numbers for $^*\text{OH}$ than $^*\text{OOH}$ under 25-90 °C agree well with the Sabatier principle, that is, a good OER site should adsorb $^*\text{OOH}$ weaker than $^*\text{OH}$ for initial reactant capture and product desorption. This conclusion also supports the standpoint that the PBE+U is acceptable for computing the interactions between Ir active site and OER intermediates. Indeed, the PBE exchange-correlation functional is widely used to describe the bulks and surfaces of solid-state transition metals compounds and also the interaction of adsorbates with them (*J. Chem. Theory Comput.* 2014, 10, 9, 3832-3839; *J. Phys. Chem. C* 2011, 115, 3716-3721; *ACS Catal.* 2022, 12, 15, 9256-9269).

Supplementary Fig. 15 | The electron transfer number (Δp) was obtained by Bader charge analysis during OER intermediates (*OH, *O, *OOH) adsorbing onto Ir active site on different facets. **a**, (001) and (110) facets under 25 °C. **b**, (001) and (110) facets under 90 °C. The single-site adsorbate evolution model, achieving by OH⁻ passivating the coordinately unsaturated Ir sites on the terminal facet except the Ir active site, was used in the DFT calculations by setting the 100% coverage of all the OER intermediates adsorbing onto single Ir active site. The black ball is adsorbed oxygen atom. The blue ball is the Ir active site. The brown ball is the hydrogen atom.

I understand the difficulties of modeling thermal induced strains. However, I consider that the approach of directly using experimentally obtained lattice parameters is a very risky procedure, which has not been reasonably validated. I believe that the lattice parameters optimized under the model chemistry should always be considered in any reasonable approach. Perhaps, applying a thermal expansion coefficient. Once again, perhaps, the goal of “accurately modeling this thermal effect” should become the less compromised goal of “establishing the plausibility of the mechanism as a thermally induced effect”, modifying the presentation accordingly.

Reply: Thanks for understanding the difficulties in modeling the thermal strains and your valuable suggestions. In the initial stage of building structure model, we have considered the route suggested by the Reviewer to apply a thermal expansion coefficient to simulate the thermal-induced lattice strains. In this case of modeling thermal-induced strains, we should ensure that the compressive strains was produced in the IrO_6 units of Sr_2IrO_4 . Therefore, we firstly compared the modeling methods of the crystal structure of the Sr_2IrO_4 with thermal strains. A route is directly to create the structural model of Sr_2IrO_4 with $I4_1/acd$ symmetry by setting the cell parameters from XRD refinement at 20 and 90 °C and the atomic fractional coordinates from Ir-O bonding length in IrO_6 units from EXAFS analysis at 20 and 90 °C. The another route is to create the structural model of Sr_2IrO_4 (the crystal structure of Sr_2IrO_4 was created according to the JCPDS No. 09-0099 and firstly was optimized by structure relaxing both of the lattice parameters and the atomic fractional coordinates) with the thermal strains at 20 °C and 90 °C by relaxing the atomic positions in the situation of fixed lattice parameters with a thermal expansion coefficient of $2.8 \times 10^{-5} \text{ K}^{-1}$ for a- and b-axes and $-1.2 \times 10^{-5} \text{ K}^{-1}$ for c-axis, which is obtained by fitting XRD data. As shown in **Supplementary Fig. 23**, the $\text{Ir-O}_{ab\text{-plane}}$ and $\text{Ir-O}_{c\text{-axis}}$ bonding lengths in the Sr_2IrO_4 structure obtained by setting the thermal expansion coefficient are obviously longer compared to the experimental values (**Supplementary Figs. 23a and b**). In addition, the temperature dependence of Ir-O bonding length is not obvious in the Sr_2IrO_4 structure obtained by setting the thermal expansion coefficient. This is because the thermal expansion coefficient is able to obviously affect the unit cell parameters, but

slightly affect the strains of IrO₆ units. In particular, the heating slightly enlarges the bonding length of Ir-O_{ab-plane}, which severely deviates the experimental result of thermal compressive strain generation in IrO₆ units. The theoretical results have demonstrated that the physical properties of Sr₂IrO₄ are dependent highly on the structure distortion (*Phys. Rev. B* 2011, 84, 100402(R)). Correspondingly, the theoretical calculations based on the Sr₂IrO₄ model created by considering thermal expansion coefficient indicated the slightly thermally-induced changes in the d band center and the Gibbs free energy for adsorbing OER intermediates onto the Ir active site of (001) facet (**Supplementary Figs. 23c-e**). This would be attributed to that the structural model of Sr₂IrO₄ created by setting the thermal expansion coefficient is underperformance in describing the compressive strains in IrO₆ units. Here, in Sr₂IrO₄, the anisotropic thermal expansion is typically originated from the thermally induced elementary microstructural deformation caused by transverse vibrations of the bridging atoms, thus producing compressive strains in IrO₆ units. During structure optimization, the numerical treatment fails to consider the transverse vibrations of the bridging atoms, thus failing to simulate the strains in IrO₆ units.

We totally agree with the Reviewer's opinion that the approach of directly using experimentally obtained lattice parameters is a very risked procedure, which may not follow the energy minimization. However, as pointed out by the Reviewer, there is a big difficulty in modeling thermal strain. In the absence of a better method to model the thermal strain in IrO₆ units, as a compromise, we use the experimental lattice parameters and Ir-O bonding length to model the thermal strains in IrO₆ units. Indeed, in this situation, the DFT calculations just provide the information of electronic states of the Sr₂IrO₄ structure with strains, which do not reflect the ground-state properties of Sr₂IrO₄. And also the Gibbs free energy is a result of the interactions between the valence states of OER intermediates and the electronic states of the Sr₂IrO₄ structure with strains. Although there are a few successful examples to investigate electronic states of topological insulators by carrying out DFT calculations on a structure model from experimental data (*Phys. Rev. Lett.* 2016, 117, 056805; *Nat. Commun.* 2014, 6, 7373), we still agree with the Reviewer's opinion that this is a very risked procedure and needs carefully to validate. In our case, fortunately, the calculated electronic states on the Sr₂IrO₄ structure created by experimental data are comparable with the experimentally obtained d band structure. We have revised the corresponding expressions to describe the compromised DFT calculation goal of "establishing the plausibility of the mechanism as a thermally induced effect".

Supplementary Fig. 23. Modeling the thermal strains for IrO_6 units in Sr_2IrO_4 . **a**, To model the thermal strains in IrO_6 units, the crystal structure of Sr_2IrO_4 with $I4_1/acd$ symmetry was created by setting the cell parameters from XRD refinement and the atomic fractional coordinates from Ir-O bonding length in IrO_6 units from EXAFS analysis. **b**, The crystal structure of Sr_2IrO_4 was created according to the JCPDS No. 09-0099. Firstly, the structure is acquired through structure relaxing both of the lattice parameters and the atomic fractional coordinates. Then, the thermal strains at 20 °C and 90 °C were considered by relaxing the atomic positions in the situation of fixed lattice parameters with a thermal expansion coefficient of $2.8 \times 10^{-5} \text{ K}^{-1}$ for a - and b -axes and $-1.2 \times 10^{-5} \text{ K}^{-1}$ for c -axis, which is obtained by fitting XRD data. **c**, The projected DOS of Ir 5d for the Sr_2IrO_4 with a crystal structure created by setting thermal expansion coefficient to model the thermal strains under 25 °C and 90 °C (Structural model in **Supplementary Fig. 23b**). The ϵ_d indicates the d-band center calculated as the first statistical moment of the d-projected DOS. **d**, **e**, Gibbs free energy diagrams for adsorbed the OER intermediates ($^*\text{OH}$, $^*\text{O}$, $^*\text{OOH}$) onto Ir active site of (001) facet of Sr_2IrO_4 with a crystal structure created by setting thermal expansion coefficient to model the thermal strains under 25 °C (**d**) and 90 °C (**e**). The (001) slab model with six atomic layers in a 1×1 unit is created according to the Sr_2IrO_4 with a crystal structure created by setting thermal expansion coefficient to model the thermal strains under 25 °C and 90 °C. The vacuum regions of 15 Å were used to avoid periodic image interactions, that is, there is enough vacuum space

so that the electron density of the material tails off to zero in the vacuum and the top of one slab has essentially no effect on the bottom of the next. We performed the hydroxyl group passivating the coordinately unsaturated Ir sites except the Ir active site on the slab model to stabilize the surface structure, thus avoiding the undesired surface reconstruction during structure optimization. The structure optimization was performed by fixing the bottom four layers and allowing the top two layers to relax. The single-site AEM model with 100% coverage of adsorbates on Ir active site was used to check the intrinsic catalytic activity (100% coverage is one adsorbate per one active site, *Phys. Chem. Chem. Phys.* 2015, 17, 21643).

The presentation of the manuscript still requires careful revision. In my last report, I provided examples of that. Not only the noted examples should have been carefully revised. For instance, the idea “Oxygen evolution reaction (OER), the bottleneck reaction for hydrogen production by water splitting, consumes excessive energy due to its sluggish kinetics.”, (lines 24 to 25, in the three submitted versions of the manuscript), which obviously depend on the catalyst, should be better introduced. In addition, I believe that some of the provided examples were not carefully revised, revealing haste and lack of consideration for the referee. For instance, in my last report, I pointed out that the sentence “Our results imply that the materials with thermophysical effects can be applied to create the efficient water splitting with the simultaneous input of heat and electricity” (lines 283 to 285, in the previous version of the manuscript) seemed unfortunate to me. Supporting such consideration, I wrote that “All the materials experiment thermophysical effects.”, and that “Not all the catalysts exhibiting anisotropic thermal expansion would necessarily favor the water splitting when heated.”. In the current version of the manuscript, the problematic sentence has been substituted by “Our results imply that the materials with thermal strain effects can be applied to create the efficient water splitting with the simultaneous input of heat and electricity.”. However, as already was explained, I believe that not all the materials with thermal strain effects would necessarily favor the water splitting with the simultaneous input of heat and electricity. For instance, in my last report, I pointed out that the sentence “Our findings indicated that heating is an effective route to tune the electronic states of materials, thus significantly accelerating OER kinetics, beyond the traditional thermal diffusion effect.” (lines 304 and 305, in the previous version of the manuscript) seemed unfortunate to me. Supporting such consideration, I wrote that “Heating as a route to tune the electronic states of materials exhibiting isotropic thermal expansion has not been discussed.”, and that “Heating would not necessarily accelerate the OER on all catalyst exhibiting anisotropic thermal expansion.” In the current version of the manuscript, the problematic sentence has been substituted by “Our findings indicated that heating is an effective route to tune the electronic states of materials with anisotropic thermal expansion, thus significantly accelerating OER kinetics, beyond the traditional thermal diffusion effect”. However, as already was explained, I believe that heating would not necessarily accelerate the OER on all catalyst exhibiting anisotropic thermal expansion.

Reply: Thanks for your carefully reading and valuable suggestions. The detailed point-by-point responses are listed below:

1) “Oxygen evolution reaction (OER), the bottleneck reaction for hydrogen production by water splitting, consumes excessive energy due to its sluggish kinetics.”, (lines 24 to 25, in the three submitted versions of the manuscript), which obviously depend on the catalyst, should be better introduced.

Reply : This sentence was revised as: “*However, the high overpotentials required for the oxygen evolution reaction (OER) in the water electrolysis pose a bottleneck for large-scale hydrogen production. Therefore, there is an urgent need to develop efficient and stable OER catalysts to overcome this challenge.* “

2) “Our results imply that the materials with thermophysical effects can be applied to create the efficient water splitting with the simultaneous input of heat and electricity” (lines 283 to 285, in the previous version of the manuscript) seemed unfortunate to me. Supporting such

consideration, I wrote that “All the materials experiment thermophysical effects.”, and that “Not all the catalysts exhibiting anisotropic thermal expansion would necessarily favor the water splitting when heated.”. In the current version of the manuscript, the problematic sentence has been substituted by “Our results imply that the materials with thermal strain effects can be applied to create the efficient water splitting with the simultaneous input of heat and electricity.”. However, as already was explained, I believe that not all the materials with thermal strain effects would necessarily favor the water splitting with the simultaneous input of heat and electricity.

Reply: This sentence was revised as “Our results imply that it is possible to apply the materials with a positive catalytic contribution of thermal strains to OER activity for creating the efficient water splitting with the simultaneous input of heat and electricity”.

3) For instance, in my last report, I pointed out that the sentence “Our findings indicated that heating is an effective route to tune the electronic states of materials, thus significantly accelerating OER kinetics, beyond the traditional thermal diffusion effect.” (lines 304 and 305, in the previous version of the manuscript) seemed unfortunate to me. Supporting such consideration, I wrote that “Heating as a route to tune the electronic states of materials exhibiting isotropic thermal expansion has not been discussed.”, and that “Heating would not necessarily accelerate the OER on all catalyst exhibiting anisotropic thermal expansion.” In the current version of the manuscript, the problematic sentence has been substituted by “Our findings indicated that heating is an effective route to tune the electronic states of materials with anisotropic thermal expansion, thus significantly accelerating OER kinetics, beyond the traditional thermal diffusion effect”. However, as already was explained, I believe that heating would not necessarily accelerate the OER on all catalyst exhibiting anisotropic thermal expansion.

Reply: This sentence was revised as “Our findings indicated that heating is a possible route to tune the electronic states of materials, which may exhibit a positive catalytic contribution to OER kinetics, beyond the traditional thermal diffusion effect.”

In addition, as for the effect of isotropic thermal expansion on OER, we have provided a control sample, SrIrO₃ with isotropic thermal expansion, in the main text. At the same heating procedure, the SrIrO₃ exhibited the thermal diffusion promoted OER because the small effect of isotropic thermal expansion on electronic states of SrIrO₃. However, the single example from SrIrO₃ is not able to reflect the properties of all materials with isotropic thermal expansion. Therefore, we totally agree with the Reviewer’s suggestions. We have revised these presentations.

We have carefully read the whole text and the similar imprecise expressions in the main text have been revised as highlighted by yellow color.

All in all, we sincerely thanks the Reviewer’s kind and warm help in improving the quality of our study. Your deep thinking deepened our understanding and improved the logic and scientific rigor of our presentations.

REVIEWER COMMENTS

Reviewer #3 (Remarks to the Author):

All my concerns have been addressed once again, although some of them still remain. I consider that the review process is progressing, albeit slowly. However, I would like to say right now that, mainly because I have doubts about the relevance of the contribution, and because the overall presentation does not seem very effective to me, it is unlikely that I could feel comfortable recommending publication of this work. This said, assuming that the relevance of the contribution becomes clearer for the editors and other referees, that the experimental procedures and results are validated by experimentalists (I am not one of them), that the computational results are presented as having only a complementary role, and that the overall presentation is further improved, I believe that the current version of the manuscript has the potential to reach a point acceptable for publication.

Recognizing the difficulties in reliably correlating experimental and computational results, the authors now choose “establishing the plausibility of the mechanism on likely exposed facets”. The arguments and the reference reinforcing the (001) facet focus provided in their last Response to the Reviewers’ comments should be included in the manuscript.

It is not entirely evident to me that, when adsorbed OH evolves and is finally desorbed, the exposed active site cannot be occupied by water, as adsorbate (instead of hydroxyl), with a significant lifetime. If this possibility cannot be ruled out, the relationship between adsorbed OH and adsorbed water should be determined and modeled under the simulated conditions. Water and hydroxyl would interact very differently with the surface. A significant amount of adsorbed water rather than adsorbed hydroxyl on the surface could potentially result in a very different model of the active site. Using the term “passivating agent” to refer to a adsorbed species that dynamically evolves until it is finally desorbed, is confusing to me. The way in which coverages are described is also confusing to me.

On line 424 of the current version of the manuscript, the authors write that “...the PBE exchange-correlation functional is widely used to describe the bulks and surfaces of solid-state transition metals compounds and also the interaction of adsorbates with them.”, citing J. Chem. Theory Comput. 10 (2014) 3832 (on bulk properties of transition metals), J. Phys. Chem. C 115 (2011) 3716 (on the stability of CeO₂ surfaces), and ACS Catal. 12 (2022) 9256 (on the interaction of atomic carbon on transition metal surfaces). Thus, only ACS Catal. 12 (2022) 9256 deals with adsorbates. However, a very different adsorbate (the carbon atom) from the OER involved intermediates, and very different surfaces from transition metal oxides (transition metal only) are investigated in such a reference. It has been some time since the PBE functional was revised by Norskov to more accurately describe interactions between adsorbates and transition metal surfaces, giving rise to the RPBE functional (Phys. Rev. B 59 (1999) 7413), not to be confused with the revPBE functional (Phys. Rev. Lett. 80 (1998) 890). Currently, RPBE is much

more widely used in electro-catalysis on transition metals than PBE. Assuming that PBE accurately describe the bulk properties of transition metals, widespread adoption of the RPBE functional in electro-catalysis on transition metals would provide evidence that accurately capturing the bulk properties of catalysts does not guarantee accurately capturing their interaction with adsorbates. These edges, in the context of transition metal oxides, should be polished.

The difficulties in accurately modeling the strain effects by DFT should be introduced in the manuscript, to give the reader the opportunity to assess for himself the risk of the adopted procedure.

Response to the Reviewers' comments

Reviewer #3 (Remarks to the Author):

All my concerns have been addressed once again, although some of them still remain. I consider that the review process is progressing, albeit slowly. However, I would like to say right now that, mainly because I have doubts about the relevance of the contribution, and because the overall presentation does not seem very effective to me, it is unlikely that I could feel comfortable recommending publication of this work. This said, assuming that the relevance of the contribution becomes clearer for the editors and other referees, that the experimental procedures and results are validated by experimentalists (I am not one of them), that the computational results are presented as having only a complementary role, and that the overall presentation is further improved, I believe that the current version of the manuscript has the potential to reach a point acceptable for publication.

Reply: Thanks for your time to discuss with us during the forth-round communication. Two experts in the electrochemical field have validated our experimental procedures and results and are willing to recommend publication of our work. Although the experimental chains are enough to verify our conclusions, we want to provide a complementary understanding by computational results. Your doubts about the relevance of the contribution make us think how to give a presentation that is effective to the experts from both theoretical and experimental fields. As is well-known, for the catalytic reactions, the electronic structures of catalysts are directly related to the catalytic performances. It is possible to obtain an excellently active catalyst by regulating the electronic structure to optimize OER performances. So, in the previous reports, the strategies to regulate electronic states of materials mainly include substituting with foreign elements, generating vacancies, tuning the strain, and engineering the interface (*Chem. Soc. Rev.* 2020, 49, 2196). I think, as a DFT expert, these strategies may also be used in your research to regulate properties of a given material. In principle, similarly, the purpose for our thermal strain route is to regulate the electronic structure for optimizing catalytic performances. The novelty of our work is that our proposed strategy to produce strain in Sr_2IrO_4 by anisotropic thermal expansion is inherently different from the external-force induced and architecture induced routes (*Nature Reviews Materials* 2017, 2, 17059). In our route, the strain generation in IrO_6 units can be adjusted by changing the temperature, a convenient route to optimize the catalytic performances of Sr_2IrO_4 .

In Introduction, we provide these description to tell the readers why we carried out this research. For example, in Introduction, we described “*Electronic states of materials, determining the energetics during the reagent adsorption, the evolution of intermediates, and the product desorption, are directly related to the OER kinetics*”. And we described “*Strain engineering is especially effective in optimizing the d-band center of transition metal-based catalysts via compressive or tensile strain in crystals. Generally, tensile strain reduces the overlap of the wavefunctions and thus gives rise to the narrowing of the metal d band and an upshift of d-band center. In contrast, compressive strain has the opposite effect, causing a broadening of the d band and a downshift of the d-band center. For OER, the d-band center is closely related to the interaction energy between adsorbate states and the metal d states, thus determining the OER kinetics. Conventionally, various straining methods, such as lattice mismatch, doping heteroatoms, morphology controlling, and introducing defects, were proposed to produce the constant strains in the crystals. In these methods, the constant strain is mainly determined by the compositions, microstructures and synthesis conditions of materials. Additionally, the variable strains can be in situ generated during the service of materials, in which the materials directly grow or are coated onto the flexible substrates to form a flexible film which can be subjected to tensile or compressive loading from the external forces. Obviously, although the*

variable strain is beneficial to optimize the electronic states, the dependence of complex preparation processes and state-of-the-art equipment limits its practical applications.”

Accordingly, the relevance of our contribution is that in this work we propose the thermal strain route to modify the electronic structures of Sr_2IrO_4 , thus improving its OER performances. The main role of theoretical results is to provide a complementary understanding of the thermal strain induced the downshifting of d-band center, which is in good agreement with the XPS valence band information.

Recognizing the difficulties in reliably correlating experimental and computational results, the authors now choose “establishing the plausibility of the mechanism on likely exposed facets”. The arguments and the reference reinforcing the (001) facet focus provided in their last Response to the Reviewers’ comments should be included in the manuscript.

Reply: We have included the last Response in the manuscript.

Considering that the Sr_2IrO_4 particle is not a perfect crystal with nearly irregular profile, we give comprehensive consideration in selecting the likely exposed (001) and (110) facet to carry out the DFT calculations. Firstly, the (001) facet is a thermodynamically stable plane in the Sr_2IrO_4 (Sci. Rep. 2013, 3, 3073), which is able to reflect at least partially the bulk properties of Sr_2IrO_4 . In addition, the layered Sr_2IrO_4 is easy to cleave along the (001) facet due to the weak interlayer interactions (Sci. Rep. 2013, 3, 3073), increasing the probability to expose it as terminal facet. In addition, as shown in Supplementary Fig. 1b, although the single-crystal Sr_2IrO_4 particles are nearly irregular, the stacking growth of (001) facets is visible as traced by dotted lines. Therefore, the density functional theory (DFT) calculations were conducted to investigate the adsorption properties of oxygen-containing intermediates on Ir sites of Sr_2IrO_4 (001) and (110) facets, two likely exposed facets according to the possible crystal growth mechanism and the crystallographic symmetry of Sr_2IrO_4 .

It is not entirely evident to me that, when adsorbed OH evolves and is finally desorbed, the exposed active site cannot be occupied by water, as adsorbate (instead of hydroxyl), with a significant lifetime. If this possibility cannot be ruled out, the relationship between adsorbed OH and adsorbed water should be determined and modeled under the simulated conditions. Water and hydroxyl would interact very differently with the surface. A significant amount of adsorbed water rather than adsorbed hydroxyl on the surface could potentially result in a very different model of the active site. **Using the term “passivating agent” to refer to a adsorbed species that dynamically evolves until it is finally desorbed**, is confuse to me. The way in which coverages are described is also confuse to me.

Reply: Thanks for your time to think the OER mechanism. Firstly, the water dissociation to OH on the surface defected sites, unsaturated sites such as oxygen vacancies, is a spontaneous process (Nature Materials 2006, 5, 189-192). This means that the stable water adsorption usually occurs on a perfect surface without highly active sites. However, for electrochemical reaction, the active site is usually unsaturated site, which is highly active and able to spontaneously dissociate water into OH as an initial adsorbate. This is why the DFT calculations for OER reaction use the *OH, *O, and *OOH as the elementary reaction step (Journal of Electroanalytical Chemistry 2007, 607, 83-89; J. Phys. Chem. B 2004, 108, 17886-17892).

As described in the last Response, “Usually, for alkaline water electrolysis on surface of oxides, **the water dissociation is a spontaneous process to form OH^- on the unsaturated active sites**. As demonstrated on the surface of IrO_2 (J. Am. Chem. Soc. 2017, 139, 149-155). For the stoichiometric IrO_2 (110) surface, half of the surface Ir atoms are 5 coordinated. The H_2O binds strongly (by ~ 1.7 eV/ H_2O in liquid water) at this surface and the H_2O molecules spontaneously dissociate to form $\cdot\text{OH}$ at the unsaturated Ir (Ir-5c) and $\cdot\text{OH}$ at the bridging O, independent of the starting surface water configurations. Indeed, in the alkaline water electrolysis, oxides or hydroxides have been demonstrated to be the efficient catalysts to accelerate water dissociation. For example, the alkaline water electrolysis on Pt electrode is effectively promoted by the Fe, Co, Ni, Mn containing hydr(oxy)oxides (Science 2011, 334, 1256-1260; Nat. Mater. 2012, 11, 550-557). This

means that the initial water dissociation on surface of oxides is commonly spontaneous process to occur.”

The adsorbate evolution mechanism is typically assumed to involve four concerted proton-electron transfer (CPET) reactions centered on the metal ion, as described in eqn (1)-(4). At each step, a proton is injected into the electrolyte, eventually combining with a transferred electron at the cathode. Accordingly, **OH as the dissolving state of water first adsorbs on the surface active site** (eqn (1)). The adsorbed OH (*OH species) then undergoes subsequent deprotonation to form *O (eqn (2)). The following O-O bond formation step allows *O to react with another OH to form the *OOH intermediate (eqn (3)). In the final step, O₂ is evolved through the deprotonation of *OOH with the regeneration of the active site (eqn (4)).

As is well-known, the adsorbate evolution mechanism is typically assumed to involve four concerted proton-electron transfer (CPET) reactions centered on the metal ion, as described in eqn (1)-(4). That is, the evolution route of OER intermediates is usually preset according to the recognized four-step OER mechanism. The DFT calculations are to calculate the energy difference of the slab model before and after OER intermediates adsorption of each elementary reaction according to the eqn (1)-(4).

The following equations 5 - 7 were employed to calculate the free energy changes:

$$\Delta G_{* \text{OH}} = (E_{* \text{OH}} + 0.5 \times E_{\text{H}_2} - E_{\text{H}_2\text{O}} - E^*) + (ZPE_{* \text{OH}} + 0.5 \times ZPE_{\text{H}_2} - ZPE_{\text{H}_2\text{O}} - ZPE^*) - T \times (S_{* \text{OH}} + 0.5 \times S_{\text{H}_2} - S_{\text{H}_2\text{O}} - S^*) \quad (5)$$

$$\Delta G_{* \text{O}} = (E_{* \text{O}} + E_{\text{H}_2} - E_{\text{H}_2\text{O}} - E^*) + (ZPE_{* \text{O}} + ZPE_{\text{H}_2} - ZPE_{\text{H}_2\text{O}} - ZPE^*) - T \times (S_{* \text{O}} + S_{\text{H}_2} - S_{\text{H}_2\text{O}} - S^*) \quad (6)$$

$$\Delta G_{* \text{OOH}} = (E_{* \text{OOH}} + 1.5 \times E_{\text{H}_2} - 2 \times E_{\text{H}_2\text{O}} - E^*) + (ZPE_{* \text{OOH}} + 1.5 \times ZPE_{\text{H}_2} - 2 \times ZPE_{\text{H}_2\text{O}} - ZPE^*) - T \times (S_{* \text{OOH}} + 1.5 \times S_{\text{H}_2} - 2 \times S_{\text{H}_2\text{O}} - S^*) \quad (7)$$

where E, ZPE, and S are the total energy, zero-point energy, and entropy of intermediates, respectively.

Due to the well-known four-step reaction with *OH, *O, and *OOH intermediates, this means that the DFT calculated results reflect dynamical evolution of the a adsorbed species, but the DFT calculations are to describe the energy difference of the slab model before and after OER intermediates adsorption of each elementary reaction. Therefore, no matter what passivating agent is used, we only need to get the energy difference before and after OER intermediates interacting with the active site. This is totally different from calculating an unknown reaction process, in which we need to search transition state for finding the possible intermediates.

Perhaps, an example to carry out the DFT calculations on the Fe-doped γ -NiOOH can clarify your doubts. On the surface slab of Fe-doped γ -NiOOH with inherent terminal hydroxyl, the adsorbed energy of OER intermediates (*OH, *O, *OOH) on the surface slab model can be calculated on the single Ni active site (*J. Am. Chem. Soc.* 2015, 137, 1305). In this case, the slab model was constructed by removing a OH on Fe-doped γ -NiOOH to generate Ni active site and keeping other OH as terminal OH. This treatment is much similar to construction of our model. The adsorbed energy of *OH was calculated by the energy difference before and after OH interacting with the Ni active site. The coverage on single active site is 100%, only indicating the full adsorbed state on a single active site.

On line 424 of the current version of the manuscript, the authors write that “...the PBE exchange-correlation functional is widely used to describe the bulks and surfaces of solid-state transition metals compounds and also the interaction of adsorbates with them.”, citing J. Chem. Theory Comput. 10 (2014) 3832 (on bulk properties of transition metals), J. Phys. Chem. C 115 (2011) 3716 (on the stability of CeO₂ surfaces), and ACS Catal. 12 (2022) 9256 (on the interaction of atomic carbon on transition metal surfaces). Thus, only ACS Catal. 12 (2022) 9256 deals with adsorbates. However, a very different adsorbate (the carbon atom) from the OER involved intermediates, and very different surfaces from transition metal oxides (transition metal only) are investigated in such a reference. It has been some time since the PBE functional was revised by Norskov to more accurately describe interactions between adsorbates and transition metal surfaces, giving rise to the RPBE functional (Phys. Rev. B 59 (1999) 7413), not to be confused with the revPBE functional (Phys. Rev. Lett. 80 (1998) 890). Currently, RPBE is much more widely used in electro-catalysis on transition metals than PBE. Assuming that PBE accurately describe the bulk properties of transition metals, widespread adoption of the RPBE functional in electro-catalysis on transition metals would provide evidence that accurately capturing the bulk properties of catalysts does not guarantee accurately capturing their interaction with adsorbates. These edges, in the context of transition metal oxides, should be polished.

Reply: Thanks for your suggestions. We have carried out the RPBE calculations and updated the corresponding results and references. We updated the results including the Gibbs free energy diagrams for OER intermediates (*OH, *O, *OOH) adsorbing onto the single Ir active site and the Bader charge analysis. The RPBE calculations present the similar results to the PBE calculations. As is previously studied, the revised RPBE is mainly an advantage for the early and mid transition metals and not very much for the late transition metals (J. Chem. Theory Comput. 2018, 14, 7, 3479–3492).

The difficulties in accurately modeling the strain effects by DFT should be introduced in the manuscript, to give the reader the opportunity to assess for himself the risk of the adopted procedure.

Reply: Thanks for your suggestions. We have added these description into the main text.

In modeling thermal-induced strains, we should ensure that the compressive strains was produced in the IrO₆ units of Sr₂IrO₄. We firstly created the structural model of Sr₂IrO₄ with the thermal strains at 20 °C and 90 °C by setting the unit cell parameters with a thermal expansion coefficient of $2.8 \times 10^{-5} \text{ K}^{-1}$ for a- and b-axes and $-1.2 \times 10^{-5} \text{ K}^{-1}$ for c-axis, which is obtained by fitting XRD data. However, the structural model of Sr₂IrO₄ created by setting the thermal expansion coefficient is underperformance in describing the compressive strains in IrO₆ units (Supplementary Fig. 23). Therefore, as a compromise, the structural model of the Sr₂IrO₄ was built by setting the unit cell parameters from XRD refinement at 20 and 90 °C and the atomic fractional coordinates from Ir-O bonding length in IrO₆ units from EXAFS analysis at 20 and 90 °C. The lattice parameters of Sr₂IrO₄ are $a = b = 5.50 \text{ \AA}$, $c = 25.74 \text{ \AA}$, for Sr₂IrO₄ under 25 °C, and $a = b = 5.51 \text{ \AA}$, $c = 25.72 \text{ \AA}$ for Sr₂IrO₄ under 90 °C. Under 25 °C, the Ir-O_{ab-plane} and Ir-O_{c-axis} bond lengths are 1.989 Å and 1.993 Å, respectively. Under 90 °C, the Ir-O_{ab-plane} and Ir-O_{c-axis} bond lengths are 1.938 Å and 1.937 Å, respectively. It is worth noting that, in this situation, the DFT calculations just provide the information of electronic states of the Sr₂IrO₄ structure with strains, which do not reflect the ground-state properties of Sr₂IrO₄. And also the Gibbs free energy is a result of the interactions between the valence states of OER intermediates and the electronic states of the Sr₂IrO₄ structure with strains.

REVIEWER COMMENTS

Reviewer #3 (Remarks to the Author):

All my concerns have been addressed, although some of them still remains.

To support the assumed adsorbates scenery, it is argued that "...the water dissociation to OH on the surface defected sites, unsaturated sites such as oxygen vacancies, is a spontaneous process (Nature Materials 2006, 5, 189-192). This means that the stable water adsorption usually occurs on a perfect surface without highly active sites. However, for electrochemical reaction, the active site is usually unsaturated site, which is highly active and able to spontaneously dissociate water into OH as an initial adsorbate". However, the catalyst is computationally modeled by a perfect crystalline surface (without defects, without oxygen vacancies). I can assume that adsorbed water on the atom acting as an active site can spontaneously evolve to adsorbed OH. However, is not completely evident to me that all the undercoordinated metal atoms exposed by the surface can simultaneously act as active sites, driving simultaneously the evolution of adsorbed water, under any condition (pH, potential...). In other words, it is not completely evident to me that all exposed uncoordinated metal atoms must necessarily reach the "covered by adsorbed OH" state before a given adsorbed OH has a chance to progress in the oxidation process, under any condition (pH, potential...). In any case, the assumed adsorbates scenery should at least be clearly described and referenced, to give the reader the opportunity to asses for himself the plausibility of the assumption.

To support the applied numerical treatment, it is argued that "...the revised RPBE is mainly an advantage for the early and mid transition metals and not very much for the late transition metals (J. Chem. Theory Comput. 2018, 14, 7, 3479–3492)". However, such a reference deals with adsorption on transition metals, not transition metal oxides, whose electronic structure and reactivity is very different. Despite the reproduced conclusion, which would not be applicable because the substrate would be very different, the authors choose to adopt RPBE as the functional to perform the calculations on adsorption energies. In my previous reports, PBE0 or RPBE were suggested as alternatives to evaluate. RPBE was not pointed out as a recommendation. In fact, "Assuming that PBE accurately describe the bulk properties of transition metals, widespread adoption of the RPBE functional in electro-catalysis on transition metals would provide evidence that accurately capturing the bulk properties of catalysts does not guarantee accurately capturing their interaction with adsorbates. These edges, in the context of transition metal oxides, should be polished" was written. Validation in the context of OER on transition metal oxides was specifically emphasized. I consider that the applied numerical treatment has not yet been rigorously validated for the application. This said, the fact that, after several iterations, it has still not been possible to identify a rigorously validated numerical treatment for the application would provide evidence that the validation of applied numerical treatments is not a common concern. Of course, the applied numerical treatment are always described, but only occasionally arguments supporting their adoption are provided, usually lacking rigorous validations for the application. In this sense, given that the OER on

transition metal oxides is often investigated using PBE, the computational part of this research under PBE would be, at least, “standard”. In the absence of validation, RPBE would be less “standard” for transition metal oxides.

Response to the Reviewers' comments

REVIEWER COMMENTS

Reviewer #3 (Remarks to the Author):

All my concerns have been addressed, although some of them still remains. To support the assumed adsorbates scenery, it is argued that "... the water dissociation to OH on the surface defected sites, unsaturated sites such as oxygen vacancies, is a spontaneous process (Nature Materials 2006, 5, 189-192). This means that the stable water adsorption usually occurs on a perfect surface without highly active sites. However, for electrochemical reaction, the active site is usually unsaturated site, which is highly active and able to spontaneously dissociate water into OH as an initial adsorbate". However, the catalyst is computationally modeled by a perfect crystalline surface (without defects, without oxygen vacancies). I can assume that adsorbed water on the atom acting as an active site can spontaneously evolve to adsorbed OH. However, is not completely evident to me that all the undercoordinated metal atoms exposed by the surface can simultaneously act as active sites, driving simultaneously the evolution of adsorbed water, under any condition (pH, potential...). In other words, it is not completely evident to me that all exposed uncoordinated metal atoms must necessarily reach the "covered by adsorbed OH" state before a given adsorbed OH has a chance to progress in the oxidation process, under any condition (pH, potential...). In any case, the assumed adsorbates scenery should at least be clearly described and referenced, to give the reader the opportunity to asses for himself the plausibility of the assumption.

Reply : Thanks for your thinking on the OER theoretical calculations. Here, as described in the previous Responses and the main text, "Owing to the single Ir 5-coordinated environment on both (001) and (110) facets (Supplementary Fig. 14) and the satisfied stability of Sr₂IrO₄ during OER (No surface reconstruction after OER i-t test at 1.45 V at 90 °C for 1 h is observed by the HRTEM in Fig.1a), we calculated the OER activity on a single active site adsorbate evolution mechanism (AEM) model.⁴⁵ Owing to that the surface hydroxylation of oxides is a thermodynamically favorite process in the alkaline environment, we performed the hydroxyl group passivating the coordinately unsaturated Ir sites except the Ir active site on the slab model to stabilize the surface structure, thus avoiding the undesired surface reconstruction during structure optimization. The single-site AEM model with 100% coverage of adsorbates on Ir active site was used to check the intrinsic catalytic activity (100% coverage is one adsorbate per one active site),⁵⁸ thus establishing the plausibility of the thermal strains enhanced OER on the likely exposed (001) and (110) facets under the plausible adsorbates scenery."

As mentioned-above, the unsaturated Ir active site is really an oxygen vacancy on the slab model, owing to that the unsaturated Ir active site on Sr₂IrO₄ oxide surface can be considered to be formed by removing the lattice oxygen. In addition, we use the single-site adsorbate evolution mechanism (AEM) model to calculate the Gibbs free energy change. In this model, there is no evolution of adsorbates between the adjacent sites or the simultaneous evolution of adsorbates on all active sites at the same time. If there is a simultaneous evolution of adsorbates on all active sites at the same time, this would mean that the adsorbed species at adjacent sites have strong interactions, leading to that the evolution of adsorbates deviates from the single-site AEM model to follow the multi-site AEM model (*Energy Environ. Sci.*, 2015, 8, 1404-1427). Thus, we can know that, in the single-site AEM model, there is no situation that all the undercoordinated metal atoms exposed by the surface can simultaneously act as active sites, driving simultaneously the evolution of adsorbed water, under any condition (pH, potential...). Considering that the OER was

performed under pH = 13.6, we reasonably considered that surface hydroxylation on the surface of Sr_2IrO_4 is a thermodynamically favorite process, which can spontaneously occur without external potential. According to the single-site AEM model, we assumed that there is only one active Ir site in the slab models, this is, other Ir sites with surface hydroxyl is stable during calculations.

An example to carry out the DFT calculations on the Fe-doped $\gamma\text{-NiOOH}$ can clarify your doubts. On the surface slab of Fe-doped $\gamma\text{-NiOOH}$ with inherent terminal hydroxyl, the adsorbed energy of OER intermediates ($^*\text{OH}$, $^*\text{O}$, $^*\text{OOH}$) on the surface slab model can be calculated on the single Ni active site (*J. Am. Chem. Soc.* 2015, 137, 1305). In this case, the slab model was constructed by removing a OH on Fe-doped $\gamma\text{-NiOOH}$ to generate Ni active site and keeping other OH as terminal OH. This treatment is much similar to construction of our model. The adsorbed energy of $^*\text{OH}$ was calculated by the energy difference before and after OH interacting with the Ni active site. The coverage on single active site is 100%, only indicating the full adsorbed state on a single active site.

To support the applied numerical treatment, it is argued that "... the revised RPBE is mainly an advantage for the early and mid transition metals and not very much for the late transition metals (*J. Chem. Theory Comput.* 2018, 14, 7, 3479-3492)". However, such a reference deals with adsorption on transition metals, not transition metal oxides, whose electronic structure and reactivity is very different. Despite the reproduced conclusion, which would not be applicable because the substrate would be very different, the authors choose to adopt RPBE as the functional to perform the calculations on adsorption energies. In my previous reports, PBEo or RPBE were suggested as alternatives to evaluate. RPBE was not pointed out as a recommendation. In fact, "Assuming that PBE accurately describe the bulk properties of transition metals, widespread adoption of the RPBE functional in electro-catalysis on transition metals would provide evidence that accurately capturing the bulk properties of catalysts does not guarantee accurately capturing their interaction with adsorbates. These edges, in the context of transition metal oxides, should be polished" was written. Validation in the context of OER on transition metal oxides was specifically emphasized. I consider that the applied numerical treatment has not yet been rigorously validated for the application. This said, the fact that, after several iterations, it has still not been possible to identify a rigorously validated numerical treatment for the application would provide evidence that the validation of applied numerical treatments is not a common concern. Of course, the applied numerical treatment are always described, but only occasionally arguments supporting their adoption are provided, usually lacking rigorous validations for the application. In this sense, given that the OER on transition metal oxides is often investigated using PBE, the computational part of this research under PBE would be, at least, "standard". In the absence of validation, RPBE would be less "standard" for transition metal oxides.

Reply: Thanks for your thinking on applied numerical treatment. In DFT study for the electronic ground-state properties of the bulk materials and its interaction with adsorbates, We used the PBE functional with corrections of on-site Coulomb U and spin-orbit coupling (SOC) effect to describe the electronic states of Sr_2IrO_4 (details see the Methods). The corrections of U and SOC effect aim at compensating the underperformances in PBE describing localized 5d states. For describing the interactions between catalyst and adsorbates, Nørskov has proposed that the RPBE (a revised PBE functional for accurately describing the chemisorption energies, generating by improving the mathematical form for the exchange energy enhancement factor ⁴⁶) is more accurate than PBE. However, the two functionals, PBE and RPBE, share the same construction logic and therefore contain the same physics and fulfill the same physical criteria, and therefore the most limitations of PBE still remain in RPBE and the RPBE exhibits obvious improvement in describing interactions between catalyst and adsorbates.⁴⁶ It is worth noting that the efficacy of RPBE in describing catalyst/adsorbates interactions depends on transition metals and RPBE usually reduces an

overestimation of PBE describing bond strength (*J. Chem. Theory Comput.* 2018, 14, 3479-3492).

To validate the plausibility of chemisorption energies calculated by PBE functional, we found that the energy profiles obtained with PBE + U + SOC (Fig. 4a,b) and RPBE + U + SOC (Supplementary Fig. 16) are similar, in good agreement with the previous studies (*ChemCatChem* 2017, 9, 1261; *J. Phys. Chem. C* 2018, 122, 29350), suggesting that the PBE + U + SOC approach is acceptable for describing the electronic states of Sr₂IrO₄ and its interactions with the OER adsorbates. The small difference in PBE and RPBE calculations probably stems from that the RPBE is mainly an advantage for the early and mid transition metals and not very much for the late transition metals (*J. Chem. Theory Comput.* 2018, 14, 3479-3492). Although this conclusion was made on the transition metals, we believe that it is suitable for describing the transition metal oxides. This is because the transition metal is usually being the active center for OER reaction. Accordingly, the OER activity is mainly dominated by the inherent physical properties of transition metal. This is why ones like to use the pure metal as an ideal model to discover the underlying OER mechanism, which is beneficial to get the general rules. Here, it is worth noting that, our DFT calculations are based on the assumptions including the direct use of experimental crystal structure, the likely exposed (001) and (110) facets, and the single-site AEM model. In this situation, the density functional theory (DFT) calculations just provide the information of electronic states of the Sr₂IrO₄ structure with strains, which do not reflect the ground-state properties of Sr₂IrO₄. And also the Gibbs free energy is a result of the interactions between the valence states of OER intermediates and the electronic states of the Sr₂IrO₄ structure with strains.